# Single-cell CAR T atlas reveals type 2 function in 8-year leukaemia remission

Zhiliang Bai[1,16], Bing Feng[2,3,16], Susan E. McClory[4,5], Beatriz Coutinho de Oliveira[6], Caroline Diorio[4,5], Céline Gregoire[6], Bo Tao[7], Luojia Yang[8], Ziran Zhao[6], Lei Peng[8], Giacomo Sferruzza[8], Liqun Zhou[8], Xiaolei Zhou[2,3], Jessica Kerr[6], Alev Baysoy[1], Graham Su[1], Mingyu Yang[1], Pablo G. Camara[9], Sidi Chen[8], Li Tang[2,3✉], Carl H. June[10,11,12✉], J. Joseph Melenhorst[6✉], Stephan A. Grupp[4,5✉] & Rong Fan[1,7,13,14,15✉]

Despite a high response rate in chimeric antigen receptor (CAR) T cell therapy for acute lymphocytic leukaemia (ALL)[1–3], approximately 50% of patients relapse within the first year[4–6], representing an urgent question to address in the next stage of cellular immunotherapy. Here, to investigate the molecular determinants of ultralong CAR T cell persistence, we obtained a single-cell multi-omics atlas from 695,819 pre-infusion CAR T cells at the basal level or after CAR-specific stimulation from 82 paediatric patients with ALL enrolled in the first two CAR T ALL clinical trials and 6 healthy donors. We identified that elevated type 2 functionality in CAR T infusion products is significantly associated with patients maintaining a median B cell aplasia duration of 8.4 years. Analysis of ligand–receptor interactions revealed that type 2 cells regulate a dysfunctional subset to maintain whole-population homeostasis, and the addition of IL-4 during antigen-specific activation alleviates CAR T cell dysfunction while enhancing fitness at both transcriptomic and epigenomic levels. Serial proteomic profiling of sera after treatment revealed a higher level of circulating type 2 cytokines in 5-year or 8-year relapse-free responders. In a leukaemic mouse model, type 2$^{high}$ CAR T cell products demonstrated superior expansion and antitumour activity, particularly after leukaemia rechallenge. Restoring antitumour efficacy in type 2$^{low}$ CAR T cells was attainable by enhancing their type 2 functionality, either through incorporating IL-4 into the manufacturing process or by priming manufactured CAR T products with IL-4 before infusion. Our findings provide insights into the mediators of durable CAR T therapy response and suggest potential therapeutic strategies to sustain long-term remission by boosting type 2 functionality in CAR T cells.

CD19-directed CAR T cell therapy has proven to be highly effective in treating relapsed or refractory ALL in paediatric and young adult patients[1–3]. Although early-phase trials have demonstrated an exceptional initial response rate, relapse has been frequently reported with varying patterns and approximately half of the patients do not sustain event-free survival 1 year after CAR T infusion[4–6], highlighting an emerging unmet need to investigate the molecular intricacies that govern long-term durable remission.

Functional persistence of CAR T cells is highly correlated with clinical response[1,7–9]. Advancements in single-cell RNA sequencing (scRNA-seq) technology have revealed the molecular mechanisms that contribute to the enduring persistence of CAR T cells[10]. Despite these insights, most single-cell CAR T profiles stem from patients with less than 2 years of clinical follow-up, and there remains a considerable knowledge gap regarding the characteristic infusion product signatures responsible for mediating ultralong remission exceeding 5 years. Our recent scRNA-seq profiling of pre-infusion CAR T cells revealed a deficiency of T helper 2 (T$_H$2) cell function in CD19$^+$ relapsed patients compared with in durable 5-year relapse-free responders in a small cohort of 12 patients[11], but we were yet to further elucidate the underlying

[1]Department of Biomedical Engineering, Yale University, New Haven, CT, USA. [2]Institute of Bioengineering, École Polytechnique Fédérale de Lausanne (EPFL), Lausanne, Switzerland. [3]Institute of Materials Science & Engineering, EPFL, Lausanne, Switzerland. [4]Division of Oncology, Children's Hospital of Philadelphia, Philadelphia, PA, USA. [5]Department of Pediatrics, Children's Hospital of Philadelphia, Philadelphia, PA, USA. [6]Lerner Research Institute, Cleveland Clinic, Cleveland, OH, USA. [7]Department of Pathology, Yale University School of Medicine, New Haven, CT, USA. [8]Department of Genetics, Yale University School of Medicine, New Haven, CT, USA. [9]Department of Genetics and Institute for Biomedical Informatics, University of Pennsylvania, Philadelphia, PA, USA. [10]Department of Pathology and Laboratory Medicine, Perelman School of Medicine, University of Pennsylvania, Philadelphia, PA, USA. [11]Center for Cellular Immunotherapies, Perelman School of Medicine, University of Pennsylvania, Philadelphia, PA, USA. [12]Parker Institute for Cancer Immunotherapy at University of Pennsylvania, Philadelphia, PA, USA. [13]Yale Stem Cell Center, Yale University School of Medicine, New Haven, CT, USA. [14]Human and Translational Immunology, Yale University School of Medicine, New Haven, CT, USA. [15]Yale Cancer Center, Yale University School of Medicine, New Haven, CT, USA. [16]These authors contributed equally: Zhiliang Bai, Bing Feng. ✉e-mail: li.tang@epfl.ch; cjune@upenn.edu; melenhj@ccf.org; grupp@chop.edu; rong.fan@yale.edu

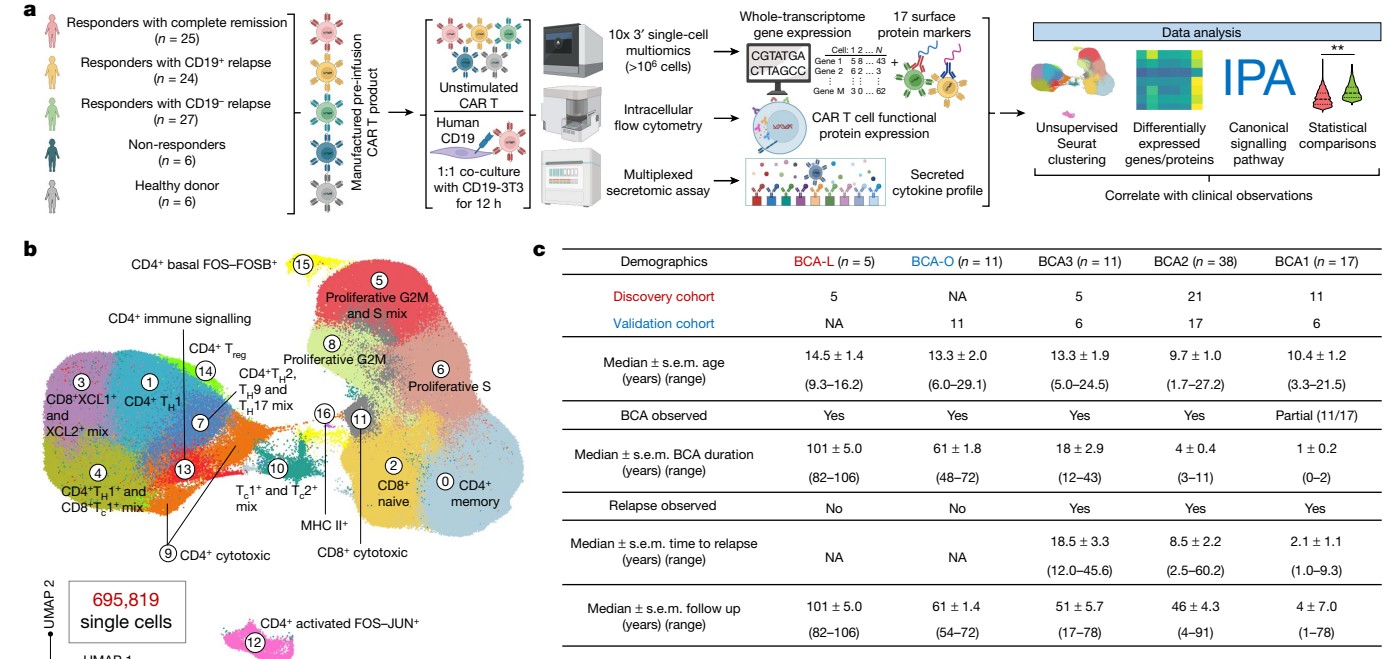

**Fig. 1 | Single-cell atlas of 695,819 CAR T cells from 82 patients and 6 HDs. a**, Schematic of the experimental design. The diagram was created using BioRender. **b**, UMAP visualization of 695,819 high-quality single CAR T cells, filtered from 1,029,340 sequenced cells across all patients and donors. Unsupervised clustering identified 17 distinct clusters. **c**, The patient demographics and clinical documentation. Patients were divided into five persistence groups on the basis of their durations of BCA. The discovery cohort includes 42 patients from clinical trial NCT01626495, and the validation cohort consists of 40 patients from clinical trial NCT02906371. IPA, ingenuity pathway analysis; $T_c$, cytotoxic T cell; $T_{reg}$, regulatory T cell.

biological mechanism and its implications for long-term response over 5 years across a large number of patients.

## Patient cohort and study design

We scrutinized pre-infusion CD19-directed CAR T (CTL019) cell products manufactured from 82 paediatric patients with relapsed and/or refractory ALL, alongside CTL019 cells generated from six healthy donors (HDs) (Supplementary Table 1). Among them, 42 participating patients were enrolled in a single-centre phase I/IIA pilot clinical trial launched in 2012 (ClinicalTrials.gov: NCT01626495), and an additional 40 patients were part of a two-cohort, open-label pilot study (ClinicalTrials.gov: NCT02906371). The CTL019 cells in both trials used a CAR with CD3ζ domain for T cell activation and a CD137 (4-1BB) domain providing costimulatory signal. Patient demographics were collected from September 2012 to July 2022, with a follow-up duration extending up to 106 months at the end point. Within our cohort, six patients exhibited no objective response to the therapy based on morphological assessment (non-responder (NR); $n = 6$). Complete remission (<5% leukaemic blasts) was attained by 73 patients who were minimal residual disease negative, and an additional 3 patients who were minimal residual disease positive. Of these, 24 developed relapse with CD19+ ALL cells (CD19+ relapse, RL+; $n = 24$) at a median of 9.2 months, while 27 patients relapsed with loss of CD19 expression in leukaemia cells at a median of 8.9 months (CD19− relapse, RL−; $n = 27$). The remaining patients demonstrated continuous and robust relapse-free remission (complete responder (CR); $n = 25$) at the last point of contact. To consistently capture authentic CAR-specific immune synapses, we engineered a mouse NIH3T3 cell line with human CD19 expression (CD19-3T3) as antigen-presenting cells (APCs), activating CAR T products exclusively through their CAR. Combining scRNA-seq and CITE-seq[12], we performed multi-omics profiling of both basal unstimulated CAR T cells and the activated CAR+ cells, complemented by independent assessments using intracellular

flow cytometry and a multiplexed secretomic assay (Fig. 1a and Extended Data Fig. 1a).

## Single-cell atlas of pre-infusion CAR T cells

We acquired single-cell transcriptome and surface protein epitope sequencing data from a total of 1,029,340 pre-infusion CAR T cells, achieving an average sequencing depth of 40,497 reads per cell (Supplementary Table 2). After the exclusion of low-quality cells and potential doublets, we analysed the expression profile of 695,819 cells, each with a median of 2,544 genes and a total of 17 surface proteins. Unsupervised clustering and uniform manifold approximation and projection (UMAP) visualization of the integrated dataset identified 17 subpopulations (Fig. 1b, Extended Data Fig. 1b and Supplementary Table 3), separated primarily by stimulation conditions (Extended Data Fig. 1c), exhibiting distinct antibody-derived tag expression patterns for subtype T cell markers CD4 and CD8, the activation T cell marker CD69 and the naive T cell marker CD62L (Extended Data Fig. 1d), with minimum batch effects observed (Extended Data Fig. 1e).

By calculating cell cycle phase scores using canonical markers, we inferred the phase (G1, G2/M or S) of each cell (Extended Data Fig. 1c). The basal state CAR T cells was partitioned into eight subclusters (cluster IDs: 0, 2, 5, 6, 8, 11, 15 and 16), primarily distinguished by their memory or cell cycle state (Extended Data Fig. 1f,l). A minor fraction of unstimulated CD8+ cells in cluster 11 exhibited cytotoxic signatures characterized by high expression of *GZMA* and *GZMB* (Extended Data Fig. 1k), probably indicative of CAR tonic signalling elicited during the manufacturing process[13,14]. After APC stimulation, most cells exhibited CD4+ or CD8+ type 1 ($T_H1$ and $T_c1$) signatures, marked by elevated expression of *Il2*, *IFNG*, *TNF*, *CSF2* and *TBX21* (Extended Data Fig. 1g). Within cluster 7, there was a combination of CD4+ type 2 ($T_H2$), type 9 ($T_H9$) and type 17 ($T_H17$) cells, coupled with heightened expression of *CCL3* and *CCL4* (Extended Data Fig. 1i,j). Cluster 3 exhibited elevated expression of *XCL1* and *XCL2* (Extended Data Fig. 1j), encoding proteins

of the C chemokine subfamily, suggesting immune communication between CAR T cells and conventional type 1 dendritic cells[15]. Expression of the regulatory T cell transcription factor *FOXP3* was identified, particularly within cluster 14 (Extended Data Fig. 1h). In cluster 12, we identified a distinct subgroup of cells marked by the expression of *FOS* and *JUN*, forming an isolated group (Extended Data Fig. 1m). This cluster encompassed both basal and activated CAR T cells. We calculated the average expression level of marker genes and cellular proteins across all the single cells in each cluster, generating a pseudo-bulk heat map that reaffirmed our delineation of the transcriptomic and surface proteomic landscape of CAR T cells (Extended Data Fig. 1n,o).

## Grouping patients based on BCA duration

We proceeded to correlate the heterogeneity of infusion products with clinical observations, aiming to identify mechanisms underlying ultralong CAR T persistence. The presence of B cell aplasia (BCA) in the peripheral blood has served as a surrogate to gauge the functional persistence of CTL019 cells, as patients with early B cell recovery within 6 months almost invariably experience relapse[16]. Patients in our CR cohort have achieved satisfactory long-term remission; however, eight CR patients exhibited detectable BCA lasting less than 6 months, prompting reinfusion of CAR T products within half a year after their initial administration in five of these cases (Supplementary Table 1). In our cohort, five CR patients have demonstrated a median BCA duration of 8.4 years (BCA-L group), maintaining BCA at the time of the last clinical follow-up, representing a unique opportunity to identify molecular determinants of CAR T longevity. Consequently, we classified all patients into five persistence groups based on their BCA duration (Fig. 1c). BCA1 represents the shortest duration, less than 3 months, and includes all of the patients in the NR group. Approximately 86% of patients in BCA2 and BCA3 experienced either CD19$^+$ or CD19$^-$ relapse. Patients in the BCA-O group continued to have BCA at the last data collection, with a median duration of 5.1 years. To mitigate potential confounding variables related to trial design, 42 patients, including the 5 patients in the BCA-L group from NCT01626495, were analysed as the discovery cohort, while the other 40 patients, including the 11 patients in the BCA-O group from NCT02906371, constituted the validation cohort.

## Basal CAR T profiles and persistence

We next examined whether the characteristic profile of basal CAR T cells could differentiate persistence groups. Clustering analysis of CAR T cells at rest resolved 11 transcriptionally distinct subpopulations (Extended Data Fig. 2a–c). The memory score, defined by the expression of *CCR7*, *TCF7*, *Il7R*, *AQP3*, *CD27* and *LTB*, was enriched in clusters 0 and 4, with the BCA-L group showing a significantly higher cell proportion in clusters 0 and 4 compared with BCA2 and BCA1 (Extended Data Fig. 2d). Proliferation markers (*MKI67*, *TYMS*, *TOP2A* and *ASPM*) were enriched in cluster 1 (Extended Data Fig. 2e), while cluster 3 exhibited a high CD8$^+$ cytotoxic score based on *GZMA*, *GZMB*, *GZMH*, *GNLY*, *PRF1* and *NKG7* expression (Extended Data Fig. 2b,f). BCA-L showed a significant decrease in cell proportion in these clusters compared with the other groups. We quantively compared the average expression of marker genes at a pseudo-bulk level, revealing increased memory gene expression and decreased cytotoxicity, proliferation and coinhibitory gene expression in BCA-L and BCA-O CAR T cells (Extended Data Fig. 2g). Surface protein profiling consistently showed elevated expression of CCR7 and CD127 (encoded by *Il7R*), along with notably low expression of coinhibitory surface markers (PD-1, CTLA-4, LAG-3, TIM3 and TIGIT) (Extended Data Fig. 2h). Together, these data point to a conserved memory state, diminished proliferation activity and subtle cytotoxic and coinhibitory signatures in basal BCA-L cells, suggesting that the ability to preserve an inactive state is associated with CAR T cell longevity.

## Type 2 function in long-term responders

We extended the assessment to CD19-specific APC-stimulated CAR T cells to gain insights into their early immune activation kinetics. UMAP clustering of cells from the discovery cohort revealed 13 distinct clusters (Fig. 2a and Extended Data Fig. 3a). The BCA-L group exhibited a significantly higher prevalence in cluster 0, constituting 33.8% of all cells, and demonstrating an enriched expression of the memory score (Extended Data Fig. 3b). This suggests that they have a superior ability to preserve central memory states after activation. Consistently, the pseudo-bulk surface protein profile revealed elevated expression of the memory markers CCR7 and CD127, along with reduced expression of the prototypic apoptosis mediator FAS (also known as CD95 and APO-1)[17] in BCA-L and BCA-O cells, while other groups showed increased expression of coinhibitory markers (Extended Data Fig. 3c).

A dominant population in cluster 1 (16.5% of the entire population) showed an enriched type 1 score (*IFNG*, *TNF*, *CSF2* and *TBX21*) and primarily comprised CD8$^+$ cells (Fig. 2a,b). Despite a lower proportion among HD CAR T cells, patient BCA groups displayed indiscernible statistical differences in this cluster. Notably, BCA-L cells were significantly more abundant in cluster 7—a mixed population of CD4$^+$ and CD8$^+$ cells with an enrichment of type 2 score (*Il4*, *Il5*, *Il13* and *GATA3*) expression (Fig. 2a,b)—regardless of its relatively low percentage (3.14%) among all single cells. Genes defining type 1 or type 2 score have a variable expression distribution and level (Extended Data Fig. 3d), highlighting the necessity of using a group of genes to define a phenotypic signature. In the validation cohort, we noted similar clustering patterns, characterized by a higher proportion of memory cells, a comparable level of type 1 cells and a significantly increased proportion of type 2 cells in 5 year relapse-free patients in the BCA-O group (Fig. 2c,d and Extended Data Fig. 3e–g), suggesting uniform molecular signatures of long-term persistent CAR T cells. In both cohorts, subclustering analyses of CD4$^+$ or CD8$^+$ cells, separated by CITE-seq surface proteins, also identified this increase in type 2 signatures in long-term responders, despite a slightly lower percentage in CD8$^+$ cells across all patient groups (Extended Data Fig. 4a–d). Canonical makers defining the type 2 identity, including cytokines, transcription factors, chemokine receptors and *PTGDR2* (encoding CRTH2), collectively exhibited elevated expression in both CD4$^+$ and CD8$^+$ CAR T cells from patients in the HD, BCA-L and BCA-O groups (Extended Data Fig. 4e). Using this large-scale single-cell transcriptomic dataset, we constructed a molecular network to illustrate the activation of the type 2 pathway in CAR T cells from durable responders relative to those from other patients (Extended Data Fig. 4f).

The identification of heightened type 2 functionality in patients in the BCA-L and BCL-O groups prompted us to pursue additional validations. Intracellular flow cytometry analysis of APC-activated CD4$^+$ and CD8$^+$CAR$^+$ cells confirmed a significant increase in the proportion of cells expressing IL-3, IL-4, IL-5, IL-13 and IL-31 in the BCA-L group, with significance levels notably higher compared with in the poorly persistent groups, namely BCA1 and BCA2 (Fig. 2e and Extended Data Fig. 5a,b). This trend was largely preserved in the validation cohort (Extended Data Fig. 5c). We also used an independent multiplexed secretomic assay to measure secreted cytokines from activated infusion products in 32 patients (Extended Data Fig. 5d). The secretion index, quantified by the frequency of cells secreting a specific cytokine multiplied by the average signal intensity, was used to describe the secretomic power. The index of CAR T cells secreting type 2 cytokines IL-4, IL-5, IL-9, IL-13 and IL-21 was significantly higher in patients in the BCA-L and BCA-O groups (Fig. 2f), while IL-10 secretion alone showed indiscernible levels.

To identify the upstream regulatory factors linked to CAR T persistence, we conducted a single-cell assay for transposase accessible chromatin (ATAC) and transcriptome co-profiling of activated CAR T cells, comparing three patients from the BCA-L group with another

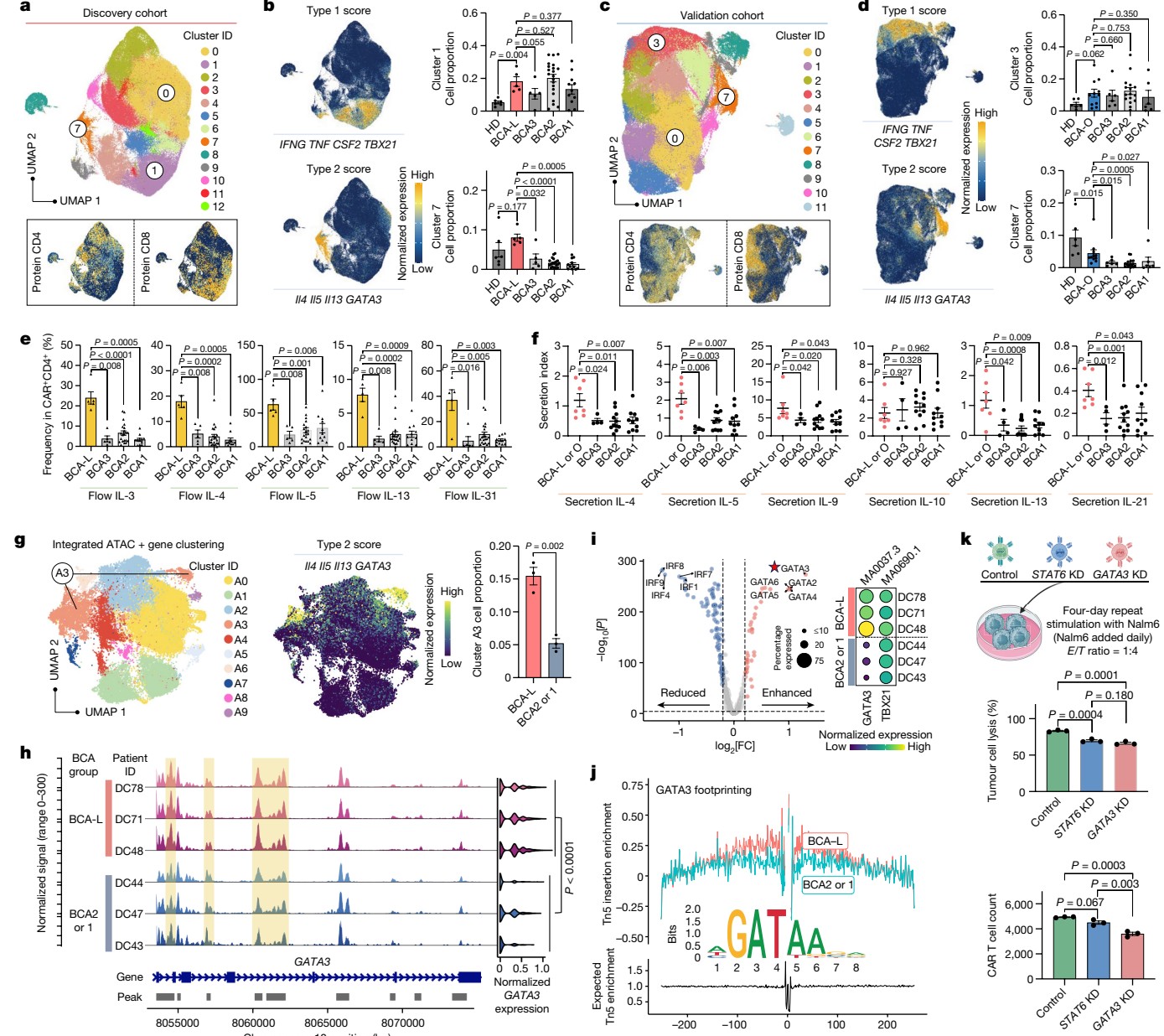

**Fig. 2 | Elevated type 2 functionality of CAR T products is associated with 8-year ultralong remission. a–d**, Unsupervised clustering analysis of CD19-3T3-stimulated CAR T cells from the discovery cohort (**a**) or validation cohort (**c**), along with the expression distribution of surface protein CD4 and CD8. The expression pattern of type 1 and type 2 score and cell proportion comparisons in specific clusters on the discovery cohort (**b**) or validation cohort (**d**) UMAP are presented. **e**, Frequency comparison of the type-2-cytokine⁺ population in CD4⁺CAR⁺ cells, assessed using intracellular flow cytometry. **f**, Comparison of type 2 cytokine secretion levels, assessed using a multiplexed secretomic assay on a cohort of 32 patients. **g**, Integrated ATAC–gene UMAP clustering of CD19-3T3-stimulated CAR T cells derived from six patients, with an enrichment of type 2 cells identified in cluster A3. A comparison of cell proportions in cluster A3 is presented between the patient groups. **h**, Pseudo-bulk chromatin accessibility tracks in the genomic region of *GATA3*, depicted separately for

each patient. Enhancer elements predicted by ENCODE are highlighted in a light yellow. **i**, Differential motif activities in BCA-L versus BCA2 or 1 CAR T cells (left). Right, the expression profile of type 2 motif MA0037.3 (GATA3) and type 1 motif MA0690.1 (TBX21) across each patient. FC, fold change. **j**, Motif footprinting trace showing the transcription-factor-binding dynamics of GATA3 in the two patient groups, alongside its position weight matrices. **k**, Evaluation of tumour cell lysis efficacy and CAR T cell count in an in vitro repeat stimulation assay using CAR T cells with knockdown of *STAT6* and *GATA3*. KD, knockdown. The diagram was created using BioRender. Data are mean ± s.e.m. from n = 48 (**b**), n = 46 (**d**), n = 42 (**e**), n = 32 (**f**) and n = 6 (**g**) patients or HDs, or n = 3 technical replicates for each condition (**k**). Significance levels were calculated using two-tailed Mann–Whitney U-tests (**b**, **d**, **e**, **f**, **h** and **i**), two-tailed unpaired Student's t-tests (**g**) and one-way analysis of variance (ANOVA) with Tukey's multiple-comparison test (**k**).

three patients from the BCA2 or BCA1 groups, who maintained BCA duration of less than 3 months (Supplementary Table 4). An integrated UMAP analysis combining ATAC and gene expression profiles revealed ten clusters, with type 2 cells enriched in cluster A3, demonstrating a consistent and significant increment in the BCA-L group (Fig. 2g). Pseudo-bulk chromatin accessibility signals in the genomic regions

of *GATA3*, the master regulator of type 2 T cells, and canonical type 2 cytokine genes were notably elevated in each of the patients in the BCA-L group (Fig. 2h and Extended Data Fig. 6a). Notably, comparable signals were observed for another key type 2 regulator, *STAT6*, across the six patients (Extended Data Fig. 6b). Consistent with previous transcriptomic clustering results from the complete cohorts,

the accessibility of *IFNG* and type 1 regulators, including *STAT1*, *STAT4* and *TBX21* (encoding T-bet), showed negligible differences (Extended Data Fig. 6c,d).

We next performed a differential motif analysis to identify potential transcription-factor-binding sites within open chromatin regions (Fig. 2i). Within the realm of significantly enhanced motif-binding activities observed in BCA-L CAR T cells compared with in the BCA2 or 1 cells, GATA3 emerged as the foremost differentially enriched site, along with several other GATA-family members. Examining the per-cell motif activity profile for each patient, TBX21 displayed consistent levels across all patients, whereas GATA3 showed notably enriched activities in single cells from the three patients in the BCA-L group (Fig. 2i). Notably, STAT6 did not emerge as a differential motif in this analysis. Conversely, transcription mediators belonging to the IRF family demonstrated marked reductions in BCA-L CAR T cells, among which IRF4 was actively involved in promoting T cell exhaustion and mediating impaired cellular metabolism[18]. We footprinted the binding dynamics of GATA3, STAT1 and TBX21 by using their positional information, revealing a discernible increase of accessibility for GATA3 within the BCA-L group (Fig. 2j). By contrast, the other two exhibited comparable levels to the BCA2 or 1 group (Extended Data Fig. 6e). To gauge the relative importance of STAT6 and GATA3 in orchestrating type 2 functionality, an in vitro repeat stimulation assay was conducted using CAR T cells with knockdown of *STAT6* or *GATA3* (Fig. 2k). While both knockdowns significantly attenuated the tumour-killing efficacy, the *GATA3* knockdown demonstrated a notably more pronounced compromise in the CAR T cell population number at the assay's end point, suggesting its central role in sustaining functional type 2 immunity in long-term persistent CAR T cells. Collectively, these data suggest that the prolonged therapeutic efficacy observed in patients in the BCA-L and BCA-O groups is associated with the heightened type 2 functionality in the infusion products regulated by GATA3 upregulation.

## Type 2 CAR T cells regulate dysfunction

Ligand–receptor (L–R) pairs can be used to infer intercellular immune communication through the coordinated expression of their cognate genes[19]. To investigate how type 2 functionality regulates CAR T cells to maintain their longevity, we examined cell–cell interactions based on L–R expression patterns in our scRNA-seq dataset. Within the network of L–R communication pathways emanating from type 2 cells in cluster 7, a notable hierarchy emerges, with the highest echelon of interactions explicitly converging on cluster 2 cells, accounting for 13.9% of the whole population. Signals involving type 2 cytokines were predominately received by *Il2RG*, *CD53* and *CSF2RB*, receptor genes that are broadly implicated in T cell survival, proliferation and downstream immune functions[20–22] (Fig. 3a), implying a potential regulatory role of type 2 CAR T cells for the entire population. Furthermore, we found that, in most identified clusters, L–R interactions toward cluster 2 cells prevalently engage type 2 cytokines (Fig. 3b). Moreover, the highest expression level of receptors that sense type 2 ligands, including *Il2RG*, *CD53*, *CSF2RB*, *Il4R* and *Il3RA*, was observed in cluster 2 as compared to the other clusters (Fig. 3c).

To delineate the distinctive features of cluster 2 cells, we conducted an analysis of differentially expressed genes (DEGs), revealing a significant increase in the expression of cytolytic effector genes *GNLY* and *GZMA* (Fig. 3d), accompanied by a concentrated distribution of cytotoxic score, indicating a highly cytolytic phenotype. Moreover, significant upregulations were noted for *Il32*, a gene that is specific to T cells undergoing apoptosis[23], and *CALM1*, a gene that contributes to T cell exhaustion pathways[24] within this cluster. Conversely, these cells largely lost their cytokine productions represented by significant downregulation of *IFNG*, *CSF2*, *Il2*, *Il3*, *Il5*, *Il13*, *XCL1/2* and *CCL3/4*. Consistently, pathway analysis revealed a significant inhibition of important pathways that are crucial for T cell activation and function, including mTOR, PI3K–AKT and JAK–STAT signalling[25] (Fig. 3e). Meanwhile, apoptosis signalling, MYC-mediated apoptosis, the T cell exhaustion signalling pathway and calcium-induced T lymphocyte apoptosis were jointly triggered, suggesting terminal effector signatures in these cells probably due to an excessive production of cytotoxicity. In this cluster, we also observed high expression of coinhibitory genes and proteins, particularly TIM3 and its encoding gene *HAVCR2* (Fig. 3f), and a robust positive correlation between cytotoxic score and coinhibitory score (*PDCD1*, *CTLA4*, *LAG3*, *HAVCR2* and *TIGIT*) across 23,608 single cells belonging to cluster 2 (Extended Data Fig. 7a). Furthermore, cyclins and cell cycle regulation were significantly suppressed in this cluster, with cells displaying extremely low expression of proliferation score (*MKI67*, *TYMS*, *TOP2A* and *ASPM*) (Fig. 3e,g), indicating a diminished proliferative ability. These data point to a late-differentiated dysfunctional state of cells in cluster 2[26]; notably, the cell proportion of the BCA-L group in this cluster is significantly lower than that of the short-term BCA groups (Fig. 3h). Even within this dysfunctional cluster, BCA-L cells exhibited a significantly reduced expression of coinhibitory score and coinhibitory antibody-derived tag (protein) score (PD-1, CTLA-4, LAG-3, TIM3 and TIGIT) (Extended Data Fig. 7b), implying that enhanced type 2 functionality may alleviate CAR T dysfunctions. All these analyses were also conducted in the validation cohort, confirming the regulatory effect of type 2 cells by observing it in patients in the BCA-O group (Extended Data Fig. 7c–j).

We next conducted in vitro functional studies to assess the impact of supplementing type 2 cytokines during CAR-specific activation (Extended Data Fig. 8a). First, CAR T cells from HDs were used to establish the optimized concentration, aiming for an alleviation in dysfunction while minimizing interference with type 1 functionality. The addition of 10 ng ml$^{-1}$ IL-4 resulted in a significant increase in cell proportion within the proliferative cluster, a decrease in the dysfunctional cytotoxic cluster and had a negligible impact on the type 1 CAR T enriched cluster (Extended Data Fig. 8b–e). By contrast, higher concentrations (20 ng ml$^{-1}$ and 50 ng ml$^{-1}$) led to a significant reduction in the percentage of type 1 CAR T cells. No statistically significant impact on the type 2 clusters was observed across all IL-4 concentrations. Thus, a concentration of 10 ng ml$^{-1}$ was determined to be the optimal dosage for this study. Notably, the introduction of 10 ng ml$^{-1}$ IL-5 or IL-13 did not alleviate dysfunction, despite a marked increase in the proliferative cluster (Extended Data Fig. 8f–k), which may require further investigation for dosage optimization.

We then supplied 10 ng ml$^{-1}$ IL-4 during the stimulation of CAR T cells manufactured from six patients in the BCA2 group (Fig. 3i), all sourced from our discovery cohort and maintaining a BCA duration of approximately 3 months. Unsupervised clustering analysis revealed a significant enhancement in proliferation (cluster 1) and mitigation of dysfunctional cytotoxicity (cluster 4) in patient CAR T cells after adding IL-4, with notably enriched expression of corresponding receptor genes observed in these two clusters (Fig. 3j,k and Extended Data Fig. 9a–d). The addition of IL-4 demonstrated minimal impact on their type 1 and type 2 functionality. The advantages of bolstering type 2 immunity were substantiated as shown in a signalling pathway profile, including upregulation of metabolic activities, functional immune programs and cell proliferation, coupled with downregulation of apoptosis in patient CAR T cells treated with 10 ng ml$^{-1}$ IL-4 compared with the original condition (Extended Data Fig. 9e). After excluding dysfunctional cluster 4 cells, a marked decrease in the upregulation level of functional pathways was observed, alongside a complete loss of the downregulation of apoptosis. Furthermore, the functional profile of short-term BCA2 CAR T cells, after IL-4 supplementation, displayed a pattern comparable to that of BCA-L CAR T cells, particularly in terms of type 2 pathway, oxidative phosphorylation (OXPHOS) metabolism, PI3K–AKT signalling, apoptosis signalling, type 1 pathway, mTOR signalling and cell cycle regulation (Fig. 3l).

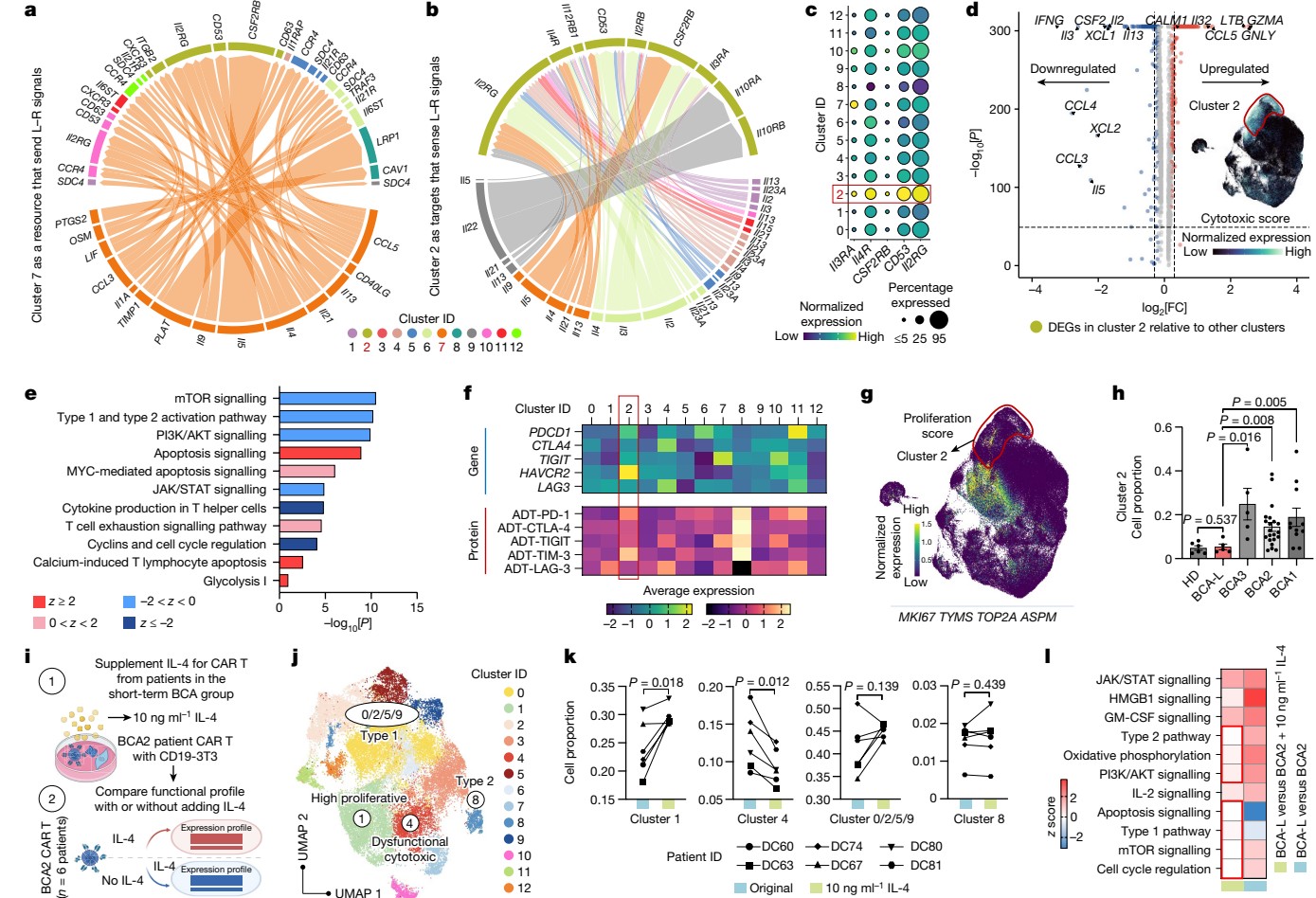

**Fig. 3 | Type 2 CAR T cells regulate dysfunctional subpopulation.**
**a**, Identification of L–R interactions originating from type 2 enriched cluster 7 cells, primarily directed towards cluster 2. **b**, Identification of L–R interactions targeting cluster 2 cells. **c**, The expression profile of type 2 receptor genes. **d**, DEGs specific to cluster 2 cells, along with the expression pattern of cytotoxic score. Genes defining this module include *GZMA*, *GZMB*, *GZMH*, *GNLY*, *PRF1* and *NKG7*. **e**, Signalling pathways regulated by the DEGs identified in cluster 2. **f**, Pseudo-bulk average expression of coinhibitory-related genes or proteins within each cluster. **g**, The expression pattern of proliferation score. **h**, Comparison of cell proportion in cluster 2. **i**, Experimental schematic for assessing the impact of IL-4 supplementation on the functional profile of CAR

T cells derived from patients in the short-term BCA2 group. The diagram was created using BioRender. **j**, UMAP clustering analysis of CAR T cells with and without 10 ng ml$^{-1}$ IL-4 added. **k**, Comparison of cell proportions in specific clusters. **l**, Comparison of regulatory pathways between CAR T cells from patients in the BCA-L group and six patients in the BCA2 group under the original and 10 ng ml$^{-1}$ IL-4 conditions. For **e** and **l**, $z > 0$, activated/upregulated; $z < 0$, inhibited/downregulated; $z \geq 2$ or $z \leq -2$, significant. Data are mean ± s.e.m. from $n = 48$ (**h**) and $n = 6$ (**k**) patients or HDs. Significance levels were calculated using two-tailed Mann–Whitney $U$-tests (**d** and **h**), right-tailed Fisher's exact tests (**e**) or two-tailed Wilcoxon matched-pairs signed-rank tests (**k**).

We further assessed the chromatin accessibility alterations after adding 10 ng ml$^{-1}$ IL-4 in CAR T cell activation for three patients in the BCA2 or 1 groups. Differential accessibility peak activity analysis unveiled a functional reprogramming of these less persistent cells after IL-4 treatment, highlighting significant upregulation of the type 1 marker *IFNG*, type 2 marker *Il13*, chemokines *XCL1* and *XCL2*, cytotoxic markers *GNLY* and *NKG7*, as well as activation/proliferation markers *VIM*[27] and *CD70*[28] (Extended Data Fig. 10a,b). Notably, the accessibility of *CSF2*, a pivotal immune modulator with profound effects on T cell functional activities[29], demonstrated the highest level of upregulation, while the activity of *Il32*, marking activation-induced cell death, exhibited substantial downregulation in the IL-4-supplemented condition (Extended Data Fig. 10c). These trends were consistently observed and supported by chromatin accessibility signal tracks in each patient. The improvement in overall functional fitness, reinforced by the enhanced motif binding of both type 1 (STAT1) and type 2 (GATA3) master regulators (Extended Data Fig. 10d), could be attributed to the effective regulation of dysfunctional population by type 2 cytokines. Together, these results establish an essential role of type 2 function

in regulating dysfunctional CAR T cell subset, thereby maintaining a balanced functional homeostasis and optimal fitness of the whole CAR T cell population.

## Type 2 cytokines in post-treatment sera
To study the patient response after CAR T infusion, we conducted a comprehensive proteomic profiling to measure serum proteins after treatment in 33 patients (including 4 patients in the BCA-L group) from our discovery cohort and 8 patients (including 3 patients in the BCA-O group) from the validation cohort (Fig. 4a). In both cohorts, longitudinal levels of type 2 cytokines were markedly elevated in long-term BCA-L and BCA-O patients compared with their counterparts, particularly for IL-13 (Fig. 4b and Extended Data Fig. 11a). By contrast, no significant variation was observed in the longitudinal levels of type 1 cytokines and selected chemokines between long-term responders and other patients (Extended Data Fig. 11b,c). In statistical analyses across serial timepoints, a significantly higher level of type 2 cytokines in patients in the BCA-L group was evident during baseline measurements

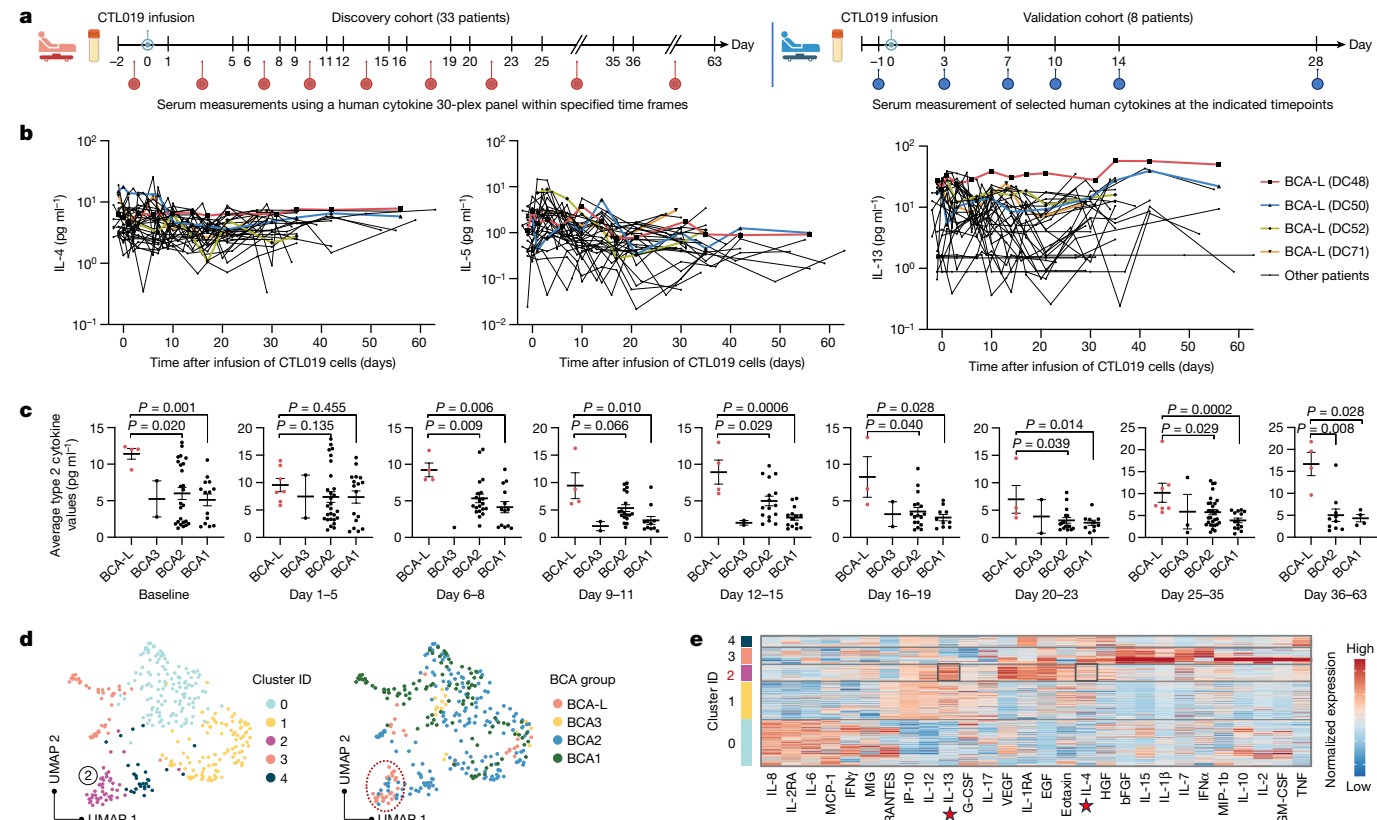

**Fig. 4 | Long-term responders exhibit higher type 2 cytokine levels in post-treatment sera. a**, Schematic of the serial proteomic profiling to measure serum proteins in 33 patients from the discovery cohort and 8 patients from the validation cohort. Timepoints are relative to the day of first infusion of CTL019 cells (day 0). The diagram was created using BioRender. **b**, Longitudinal levels of type 2 cytokines in patients from the discovery cohort. Four patients (with patient ID) in the BCA-L group are individually labelled with coloured lines. **c**, Comparison of the average type 2 cytokine levels at multiple timepoints between persistent groups in the discovery cohort. The value is the averaged expression of IL-4, IL-5 and IL-13. Data are mean ± s.e.m. from *n* = 45 (baseline),

*n* = 54 (day 1–5), *n* = 38 (day 6–8), *n* = 35 (day 9–11), *n* = 36 (day 12–15), *n* = 32 (day 16–19), *n* = 32 (day 20–23), *n* = 54 (day 25–35) and *n* = 18 (day 36–63) measurements. Significance levels were calculated using two-tailed Mann–Whitney *U*-tests. **d**, Unsupervised clustering analysis of 345 measurements of serum samples from 33 patients in the discovery cohort, grouped by cluster ID or BCA response. Each dot represents one measurement of an individual patient at each timepoint. Cluster 2 exhibits enrichment in BCA-L patients. **e**, The differentially expressed proteins defining each cluster. Stars highlight type 2 cytokines with notable expression levels.

(0 to 2 days before CTL019 infusion) (Fig. 4c). This occurrence fostered an environment rich in type 2 cytokines to instigate cellular kinetics, thereby leading to a balanced response for BCA-L cells. During the initial phase of infusion (day 1–5), no notable difference was observed, probably attributable to the transient decrease in circulating CTL019 cells resulting from their dispersion throughout the peripheral blood, bone marrow and other tissues[30,31]. From the period of rapid expansion (day 6–8) up to 2 months after infusion, a consistent and significant elevation in circulating levels of type 2 cytokines was observed in patients in the BCA-L group (Fig. 4c). No statistical differences were noted for type 1 cytokines across all of the investigated time frames (Extended Data Fig. 11d). Similar analyses were conducted in the validation cohort, affirming the notably higher levels of type 2 cytokines on days 7, 10, 14 and 28, whereas levels of type 1 cytokines remained comparable between the BCA groups (Extended Data Fig. 11e,f). We also examined whether the serum proteomic profile could mirror intrinsic immune response variations among different patients after CAR T treatment. UMAP clustering was performed on the measurement of 345 serum samples from the discovery cohort based on the detected values of 30 cytokines in our panel. This analysis resolved 5 subsets, with BCA-L profiles enriched in cluster 2 showing notably higher expression of IL-13 and IL-4 relative to other clusters (Fig. 4d,e). Together, these data validate the presence of elevated type 2 circulating cytokines in the post-treatment sera of long-term BCA-L and BCA-O patients—a pattern that is discernible through unsupervised analysis.

## Type 2[high] CAR T cells sustain tumour killing

Given the higher type 2 immunity observed in ultralong persistent CAR T cells, we next examined whether type 2[high] CAR T cells could elicit enhanced antitumour efficacy in vivo. Clustering analysis of scRNA-seq data from the 6 HDs identified ND463 as type 2[high] and ND585 as type 2[low], determined by the proportion of cells within the type 2 enriched cluster 6 (Extended Data Fig. 12a–d). There was no significant variability observed in type 1 score gene expression between these two groups (Extended Data Fig. 12e). Despite the anticipated higher ratio of CD4[+] subtype in type 2[high] CAR T cells, flow cytometry analysis revealed comparable CAR transduction efficiency, expression of memory markers and coinhibitory markers (Extended Data Fig. 12f–h).

NSG mice were used to establish a human leukaemia model by intravenous (i.v.) injection of $1 \times 10^6$ Nalm6 cells. After 7 days, the mice were randomly assigned to three groups, receiving either $2 \times 10^6$ CAR T cells or PBS (control) by infusion through the tail vein (Fig. 5a). Throughout the experiment, the treated mice maintained stable weights (Extended Data Fig. 13a). Both type 2[high] and type 2[low] CAR T cells demonstrated substantial efficacy in reducing tumour burden, as reflected by the complete clearance of tumour cells within 2 weeks (Extended Data Fig. 13b). However, flow cytometry analysis revealed that type 2[high] CAR T cell product had significantly superior expansion in vivo, with absolute CAR[+] cell counts in the peripheral blood reaching at least tenfold higher levels compared with in the type 2[low] group on both day 8

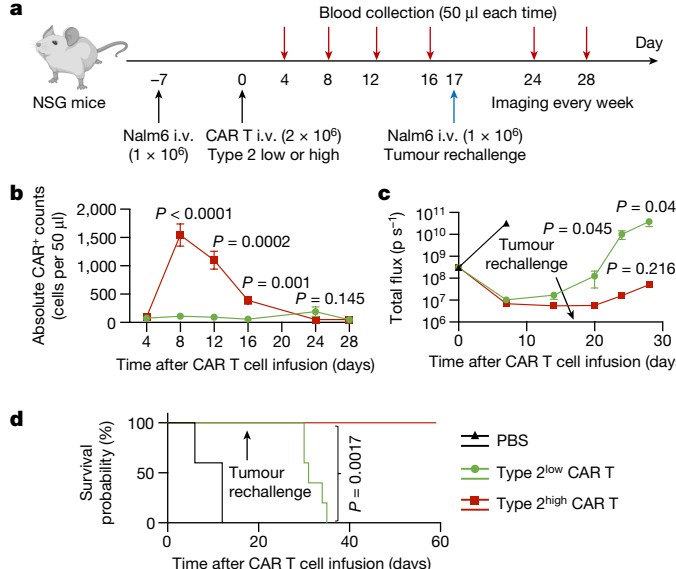

**Fig. 5 | Type 2[high] CAR T cells demonstrate enhanced antitumour activity after leukaemia rechallenge. a**, Schematic of the in vivo leukaemia model treated with type 2[low] (ND585) or type 2[high] (ND463) CAR T cells. NSG mice were i.v. injected with $1 \times 10^6$ Nalm6 cells. Then, 7 days later, mice were randomly assigned to three groups and were infused i.v. with $2 \times 10^6$ CAR T cells or PBS (control). The surviving mice were rechallenged with $1 \times 10^6$ Nalm6 cells 17 days after the CAR T infusion. The diagram was created using BioRender. **b**, CAR T cell expansion in the peripheral blood of Nalm6-bearing mice was measured at different timepoints after infusion. **c**, The tumour burden (total flux) was quantified as photons per second in mice after CAR T treatment. **d**, Kaplan–Meyer analysis of mouse survival. Data are mean ± s.e.m. from $n = 5$ mice for each group (**b** and **c**). Significance levels were calculated using two-tailed unpaired Student's $t$-tests (**b** and **c**) or log-rank Mantel–Cox tests (**d**).

and day 12 (Fig. 5b). To mimic tumour cell relapse, a second dose of $1 \times 10^6$ Nalm6 cells was administered to mice on day 17. Notably, type 2[high] CAR T cells exhibited a potent ability to elicit recall responses after leukaemia rechallenge (Fig. 5c), significantly prolonging the survival of tumour-bearing mice compared with their counterparts (Fig. 5d).

To elucidate the mechanisms driving the enhanced antitumour efficacy of type 2[high] CAR T cells in vivo, we assessed their functional properties at the peak of CAR activity on day 8. While the levels of central memory markers were similar (Extended Data Fig. 13c), type 2[high] CAR T cells displayed reduced expression of coinhibitory markers, including PD-1, TIM3, LAG-3 and KLRG1. Conversely, a significantly higher level of the apoptosis mediator FAS was noted in type 2[low] CAR T cells (Extended Data Fig. 13d). We also found a significant increase in the expression of IFNγ in the type-2[high]-treated group, suggesting that the elevated type 2 composition in the manufactured CAR T product does not impede the type 1 response (Extended Data Fig. 13e). There was no statistical difference in the expression of GZMB and TNF. After the initial elimination of leukaemic cells, type 2[high] CAR T cells acquired a favourable memory phenotype, characterized by significantly increased expression of CD27 and CD45RO on day 12. The significance of CD27 expression further intensified on day 16 (Extended Data Fig. 13f). Moreover, LAG-3 and KLRG1 maintained significantly reduced expression at both timepoints (Extended Data Fig. 13g).

To further confirm the functional role of the type 2 population on the observed superior response, we sorted out type 2 cells from the type 2[high] CAR T sample based on the surface marker expression of CCR3 and CCR4[32], conducting in vitro co-culture with Nalm6 cells over a 4 day period at an effector/target ($E/T$) ratio of 1:4 (Extended Data Fig. 13h). The exclusion of this population markedly compromised the anti-tumour ability, resulting in a reduction in the absolute CAR T cell

number at the end of the killing assay (Extended Data Fig. 13i). Phenotypic analysis revealed heightened coinhibitory signatures, diminished type 1 functionalities and decreased memory states in the type-2-sorted group (Extended Data Fig. 13j). This vulnerability was partially alleviated by supplementing with 10 ng ml$^{-1}$ IL-4 after type 2 cell removal. Notably, the addition of IL-4 further augmented CAR T cell count of the original type 2[high] CAR T cells. These data, consistent with our previous observations in the scRNA-seq analysis of long-term BCA-L CAR T cells, indicate that elevated type 2 functionality in vivo effectively reduces exhaustion and preserves central memory states of CAR T cells, which could be associated with their superior and long-lasting therapeutic activity even in the rechallenge setting.

## Type 2[low] CAR T cells can be revitalized

We next investigated whether enhancing type 2 functionality could elevate the performance of the type 2[low] CAR T sample (ND585), using two strategies: (1) priming CAR T products with type 2 cytokine before infusion and (2) incorporating type 2 cytokine into the manufacturing process, starting from apheresis T cells (Fig. 6a). Consistent with all of the previous studies, 10 ng ml$^{-1}$ IL-4 was used to prime ND585 CAR T products, resulting in primed CAR T cells. Two different IL-4 doses were assessed during the new manufacturing of ND585 T cells, that is, 10 ng ml$^{-1}$ or 50 ng ml$^{-1}$, generating enhanced type 2 CAR T cells referred to as ET2-L CAR T cells or ET2-H CAR T cells, respectively. Compared with the conventional condition supplemented with IL-7 and IL-15, the inclusion of IL-4 exhibited no discernible impact on CAR T cell expansion, viability and size, with the exception of an anticipated increase in CD4 percentage (Extended Data Fig. 14a,b).

All newly generated CAR T cells were administered to Nalm6-cell-bearing NSG mice (Extended Data Fig. 14c), and their efficacy was compared to the original type 2[low] CAR T cells manufactured from the identical donor. While all treatment groups successfully cleared the tumour burden within 1 week, only primed CAR T and ET2-L/H CAR T cells demonstrated the ability to completely reject the same amount of tumour cell rechallenge (Fig. 6b and Extended Data Fig. 14d), highlighting the therapeutic potential in CAR T cell treatment by enhancing type 2 functionality. Flow cytometry analysis at various timepoints after CAR T infusion revealed significant expansion of CAR$^+$ cells in the peripheral blood for the new products (Fig. 6c), coupled with significantly reduced expression of coinhibitory markers and increased IFNγ production at the peak on day 8 (Extended Data Fig. 14e,f). This could be linked to their superior ability to contribute to a significantly prolonged survival (Fig. 6d). Notably, the strategies did not alter the type 2 signatures (Extended Data Fig. 14g). To rigorously assess the use of the two strategies, we initiated a second tumour rechallenge on day 42, simulating a relatively late-stage relapse. After this third tumour cell injection, both the ET2-L and ET2-H CAR T cells showed robust tumour control abilities, outperforming the group treated with primed CAR T cells (Fig. 6b,d).

Finally, we implemented the priming strategy to modify the CAR T cells from a patient in the BCA2 group (DC) who demonstrated a BCA duration of 3 months before developing a CD19$^+$ relapse. During the initial 4 days of a repeated stimulation assay, both original and 10 ng ml$^{-1}$ IL-4-primed DC80 CAR T cells, co-cultured with Nalm6 cells at an $E/T$ ratio of 1:2, demonstrated potent tumour killing ability, achieving nearly 100% efficiency (Extended Data Fig. 14h). Similar to the in vivo results, the primed CAR T group exhibited a significant increase in CAR T cell count over this period and displayed significantly enhanced memory signatures, reduced coinhibitory markers, along with augmented expression of IL-2, IFNγ, GZMB, Ki-67 and IL-13 on day 4 (Extended Data Fig. 14i). We gradually decreased the $E/T$ ratio to 1:16 on day 8, which resulted in a near-total loss of tumour killing ability in the original DC80 CAR T cells. However, the IL-4-primed DC80 CAR T cells retained approximately 50% activity, probably attributed to

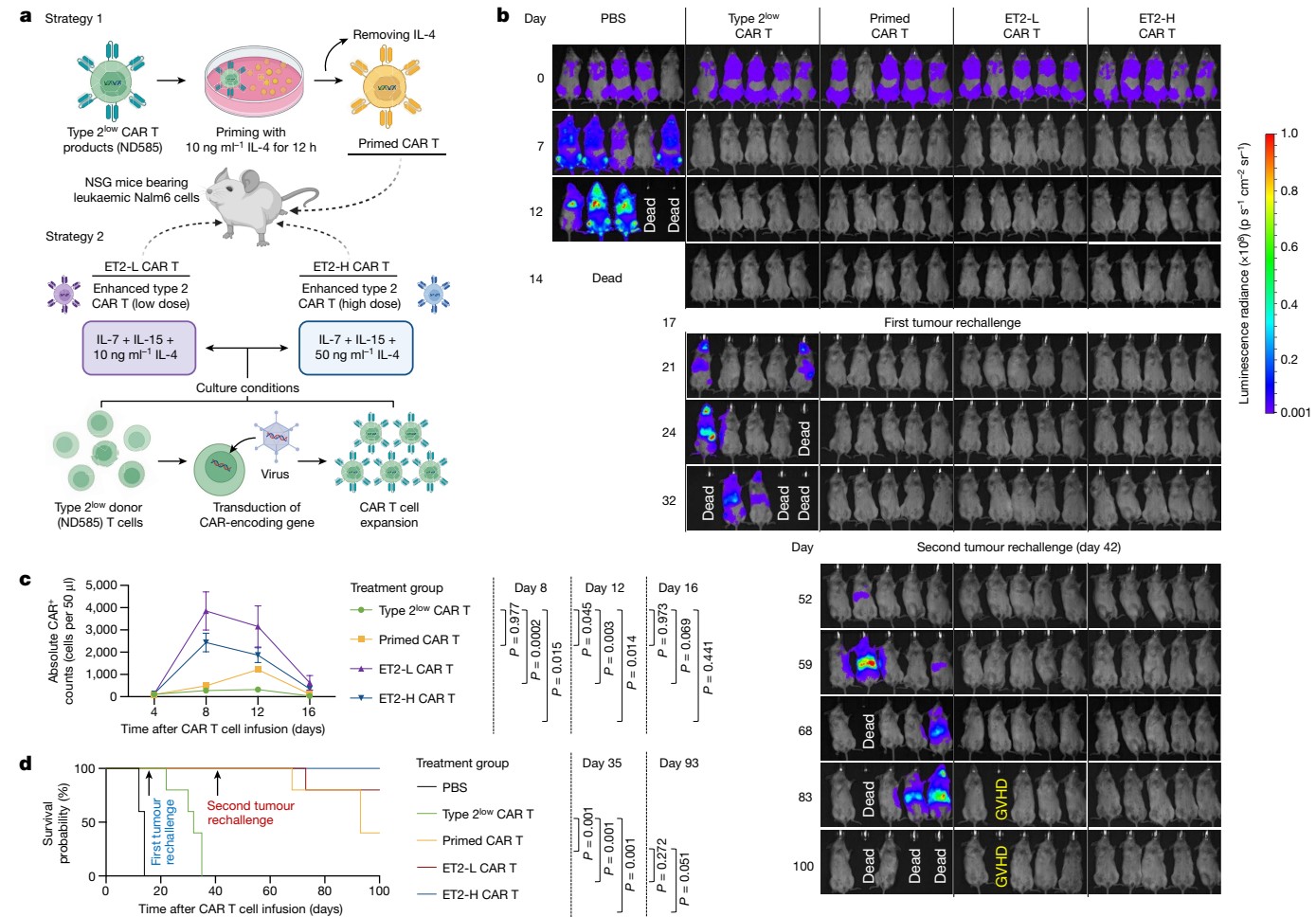

**Fig. 6 | Revitalizing type 2^low CAR T cells through enhanced type 2 functionality boost. a**, Schematic of the two strategies used to enhance the type 2 functionality of type 2^low CAR T cells derived from the donor ND585. The diagram was created using BioRender. **b**, The tumour burden was measured using bioluminescence at the indicated days after CAR T cell infusion.

GVHD, graft-versus-host disease. **c**, CAR T cell expansion in the peripheral blood of Nalm6-bearing mice measured at different timepoints after infusion. Data are mean ± s.e.m. from $n = 5$ mice for each group. **d**, Kaplan–Meyer analysis of mouse survival. Significance levels were calculated using one-way ANOVA with Tukey's multiple-comparison test (**c**) or log-rank Mantel–Cox test (**d**).

their preserved memory state, significantly heightened production of functional cytokines and increased proliferation on day 9 (Extended Data Fig. 14j). Notably, the original BCA2 cells exhibited a substantial reduction in the expression of coinhibitory molecules, suggesting that these cells might have reached a state in which they are no longer readily activated to execute tumour cell killing. Together, these findings indicate that augmenting type 2 functionality holds promise for revitalizing type 2^low CAR T cells, particularly through an improved manufacturing process by incorporating type 2 cytokines.

## Discussion

Despite a limited sample size, our recent study identified a deficiency of $T_H2$ cells in CD19^+ relapsed patients compared to durable responders[11]. The crucial contribution of type 2 functionality in maintaining CAR T longevity becomes even more marked in this expanded clinical cohort. Early studies with human antigen-specific T cell clones established that IL-4 promotes the growth of both CD4^+ and CD8^+ T cell clones[33]. Moreover, IL-4 aids in T cell survival by upregulating BCL2 and BCL-xL expression[34,35] while downregulating FAS receptor expression and caspase 3 activity, a crucial executioner enzyme in programmed cell death[35]. To elucidate how type 2 functionality effects CAR T cell longevity, we disentangled the intricate mechanisms through L–R analysis and identified that, among vigorous interactions, the type-2-enriched

cell cluster mainly regulates a subset of dysfunctional cells exhibiting overactivation of cytotoxicity, high expression of *HAVCR2* and TIM3, impaired immune function and attenuated proliferation. Notably, both chromatin accessibility profiles and in vitro functional assays revealed that augmenting type 2 functionality led to a significant enhancement of CAR T cells derived from patients who had achieved only short-term responses, improving various aspects such as type 1 functionality, cytotoxicity, proliferation and survival. This amelioration even induced a transcriptional reversion to states similar to BCA-L CAR T cells, potentially attributable to the regulation of dysfunctional cells. These findings suggest that the presence of type 2 CAR T cells maintains a homeostatic state of the entire population by suppressing hyperactive cytotoxicity in the early stage and mitigating irreversible exhaustion, thereby enhancing their persistence and fitness.

It is essential to harness these insights for improving future CAR T manufacturing practices. We have proposed and evaluated two strategies to address this, considering two scenarios: (1) adding IL-4 into the culture medium throughout the manufacturing process for products that have not yet been generated; and (2) priming CAR T products with IL-4 if remanufacturing may not be feasible for practical clinical applications. Through rigorous in vivo studies using a leukaemia mouse model, with follow-up exceeding 100 days, and two tumour rechallenges, both methods were proven to be highly effective in boosting the antitumour response of type 2^low CAR T cells. However, the optimal

level of type 2 functionality for clinical patient treatment is yet to be further investigated. Our scRNA-seq data indicate an average of 8% type 2 cells in BCA-L CAR T products, compared to less than 2% in short-term counterparts. While these findings could serve as potential thresholds, more thorough evaluations are required to establish clinical standards for implementing type 2 function in CAR T cell therapy.

In summary, our large-scale single-cell multi-omics dataset obtained from a large cohort of patients offers valuable insights into the molecular determinants of CAR T cell longevity. Our findings highlight a role of type 2 functionality in mediating a balanced type 1 and type 2 homeostatic state within the entire CAR T cell population, leading to ultralong-term persistence and fitness. Building on this, we propose several therapeutic strategies, supported by preclinical in vivo data in animal models, to enhance type 2 function in CAR T cell infusion products to mitigate dysfunction, ultimately extending the durability of response to CAR T cell therapy.

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

## Methods

### Samples from paediatric patients with ALL and HDs

The current study is a secondary investigation using patient samples collected from an existing clinical trial for which the University of Pennsylvania Institutional Board provided insight. Pre-infusion samples of CAR T cells were obtained from patients with relapsed/refractory B-ALL, participating in either a phase I/IIA pilot clinical trial designed to assess the safety and feasibility of CTL019 T cell therapy (ClinicalTrials.gov: NCT01626495) or a pilot study focusing on optimizing the timing of tocilizumab administration for managing CART19 therapy associated cytokine release syndrome (ClinicalTrials.gov: NCT02906371). Both trials were conducted jointly at the Children's Hospital of Philadelphia and the University of Pennsylvania. Before participation, the patients or their guardians provided written informed consent in accordance with the principles outlined in the Declaration of Helsinki. Laboratory procedures adhered strictly to the guidelines established by the International Conference on Harmonization for Good Clinical Practice, employing standardized operating procedures and protocols for the receipt, processing, freezing and analysis of samples. Stringent ethical regulations were rigorously adhered to throughout the study. Primary T lymphocytes from HDs were provided by the University of Pennsylvania Human Immunology Core. To ensure compliance with HIPAA regulations, all of the samples were deidentified before analysis.

### Generation of CTL019 cells

Autologous peripheral blood mononuclear cells were collected by standard leukapheresis. T cells were subsequently enriched through mononuclear cell elutriation, followed by thorough washing and activation using anti-CD3/CD28-coated paramagnetic beads. A lentiviral vector carrying a previously described CD19-specific CAR with a 4-1BB/CD3ζ transgene was constructed[36], after which it was used to transduce the cells during the activation phase and was washed out 3 days after the initiation of culture[37]. Cell expansion was facilitated using a rocking platform (WAVE Bioreactor System) for a duration of 8 to 12 days, and the beads were then magnetically removed. Finally, CTL019 cells were collected and cryopreserved for future use.

### Generation of human-CD19-expressing NIH/3T3 cell line

We developed artificial APCs using mouse fibroblast NIH/3T3 cells to activate CAR T cells targeting the corresponding antigen. The NIH/3T3 cell lines were originally procured from the American Type Culture Collection (ATCC) and maintained in DMEM medium (Gibco, 11995-065) supplemented with 10% fetal bovine serum (FBS; Gibco, 16000044), within a humidified incubator set at 37 °C with 5% $CO_2$. After reaching approximately 80% confluency, the cells were dissociated from the culture flask using 0.25% trypsin-EDTA (Gibco, 25200056) and transduced with a lentiviral vector encoding human CD19. Then, 3 days after transduction, $1 \times 10^6$ cells were labelled with an antibody specific to the transduced epitope and sorted using the FACSAria (BD) sorter to achieve a purity exceeding 99% after the introduction of the transgene. Routine screening for mycoplasma contamination and authentication was conducted both before and after genetic modification. Subsequently, stable expressing clones were selected for expansion in T25 or T75 flasks and stored cryogenically for future use.

### In vitro co-culture assay

CTL019 cells were thawed and cultured in OpTmizer T-Cell Expansion Basal Medium (Thermo Fisher Scientific, A1048501) supplemented with GlutaMAX Supplement (Thermo Fisher Scientific, 35050061) and 5% human serum AB (Gemini Bioproducts, GEM100-512) in a humidified incubator for an initial overnight rest on day 1. On day 2, dead cells were eliminated using the Dead Cell Removal Kit (Miltenyi Biotec,

130-090-101) according to the manufacturer's instructions, and a specific number of cells was enumerated using a haemocytometer before initiating co-culture assays. For stimulation with CD19-3T3 cells (APC cells), $1 \times 10^6$ CTL019 cells were combined with an equivalent number of APC cells in 2 ml of medium; for the assessment of unstimulated conditions, $1 \times 10^6$ cells were prepared. All suspensions were cultured in RPMI medium (Gibco, 11875-119) supplemented with 10% FBS in a tissue-culture-treated 24-well plate (Thermo Fisher Scientific) for 12 h in the incubator. For functional experiments evaluating the response of the CAR T cell population to type 2 cytokines, recombinant human IL-4, IL-5 or IL-13 (R&D Systems, 204-IL-010, 205-IL-010, 213-ILB-010, respectively) was added to the medium at the specified concentrations. After co-culture, the cells were collected by vigorous pipetting, and the cell suspension was passed through a 20 μm filter to remove clumps before staining with PE-labelled monoclonal anti-FMC63 single-chain variable fragment (scFv) antibody (CAR19) (Y45, ACRO Biosystems, FM3-HPY53) for 1 h at 4 °C. Subsequently, the cells were labelled with anti-PE MicroBeads UltraPure (Miltenyi Biotec, 130-105-639) and loaded onto a MACS column positioned within the magnetic field of a MACS Separator. CAR$^+$ cells, magnetically retained within the column, were isolated as the positively selected fraction, while untransduced CAR$^-$ cells flowed through.

### Sample hashing and staining with DNA-barcoded antibodies for CITE-seq

We used hashtag reagents for sample barcoding, enabling the amalgamation of eight samples into a single lane for subsequent demultiplexing during analysis. Specifically, for human samples, the hashtags consisted of two antibodies recognizing ubiquitous surface markers, CD298 and β2 microglobulin, each conjugated to the same oligonucleotide containing the barcode sequence. The magnetically selected CAR$^+$ cells obtained from the preceding step, originating from four individuals, along with the corresponding basal unstimulated CAR T cells from each individual, underwent blocking using 5 μl of human TruStain FcX Fc Blocking reagent (BioLegend, 422302). Subsequently, they were incubated with 1 μl (0.5 μg) of the respective TotalSeq-B anti-human Hashtag antibodies 1–9 (BioLegend) for 30 min at 4 °C. After staining, the samples were washed twice with 500 μl of cell staining buffer (BioLegend, 420201) and pooled into a single tube. The combined cells were then incubated in an antibody cocktail comprising 2 μl (1 μg) of each TotalSeq-B anti-human antibody (BioLegend) according to the manufacturer's protocol. The panel comprised a selection of antibodies targeting various cell surface markers including CD4 (RPA-T4, 300565), CD8 (SK1, 344757), CD45RA (HI100, 304161), CD45RO (UCHL1, 304257), CD62L (DREG-56, 304849), FAS (DX2, 305653), CD127 (A019D5, 351354), CD28 (G043H7, 302961), CD27 (O323, 302851), CCR7 (G043H7, 353249), HLA-DR (L243, 307661), CD69 (FN50, 310949), PD-1 (EH12.2H7, 329961), TIM3 (F38-2E2, 345053), LAG-3 (11C3C65, 369337), CTLA-4 (BNI3, 369629) and TIGIT (A15153G, 372727).

### scRNA-seq library preparation and sequencing

The scRNA-seq libraries were prepared using the Chromium Single-Cell 3′ Library and Gel Bead Kit v3.1 (10x Genomics, PN-1000268). Initially, 20,000 TotalSeq antibody-stained CAR T cells were suspended in PBS (Gibco, 14190-144) with 0.04% bovine serum albumin (BSA; Sigma-Aldrich, A7030) buffer and loaded onto the Chromium Next GEM chip G, where cells and uniquely barcoded beads were partitioned into nanolitre-scale gel beads-in-emulsion (GEMs). Within each GEM, cell lysis occurred, followed by reverse transcription of the released mRNA and isolation and amplification of the barcoded complementary DNA by PCR for 12 cycles. Subsequent separation of hashtag/surface protein oligo-derived cDNAs (<200 bp) and mRNA-derived cDNAs (>300 bp) was accomplished using 0.6× SPRI bead (Beckman Coulter) purification on cDNA reactions. After fragmentation, end repair and poly(A) tailing, sample indexes were incorporated, and amplification

was performed. The final libraries underwent quality-control checks before being sequenced on the Illumina NovaSeq 6000 sequencing system, with paired-end reads of 150 bp in length. Three samples were pooled and sequenced per 800G flow cell at a gene and hashtag/surface protein library pooling ratio of 8:1.

## scATAC and gene co-profiling library preparation and sequencing

Single-cell co-profiling of epigenomic landscape and gene expression in the same single nuclei was performed using the Chromium Next GEM Single Cell Multiome ATAC + Gene Expression kit (10x Genomics, PN-1000283). Initially, CAR$^+$ cells stimulated with CD19-3T3 from 6 patients, with or without a 10 ng ml$^{-1}$ IL-4 supplement, underwent washing, counting and nucleus isolation, with an optimized lysis time of 3 min. Subsequently, the isolated nucleus suspensions were incubated in a transposition mix containing a transposase enzyme, facilitating preferential fragmentation of DNA in open chromatin regions. Concurrently, adapter sequences were introduced to the ends of the DNA fragments. Approximately 9,250 nuclei were then loaded onto the Chromium Next GEM Chip J to target a final recovery of around 6,000 nuclei. During GEM generation, gel beads introduced a poly(dT) sequence that enables production of barcoded, full-length cDNA from mRNA for gene expression profiling and a spacer sequence facilitating barcode attachment to transposed DNA fragments for ATAC profiling. After GEM incubation, purification and pre-amplification PCR, separate ATAC and gene libraries were constructed using the standard protocol. After quality assessment, both libraries underwent paired-end 150 bp read sequencing on the Illumina NovaSeq 6000 sequencing system, achieving an average depth of 24,305 read pairs per nucleus for the ATAC library and 13,756 read pairs per nucleus for the gene library.

## Intracellular cytokine detection assay for CD19-3T3-stimulated patient CAR T cells

The co-cultured cells underwent a series of processing steps for immunostaining. Initially, they were washed twice in PBS and then stained for 20 min at room temperature with Live Dead Blue detection reagent (Thermo Fisher Scientific, L34962), diluted to 1:800 in PBS, to assess cell viability. Next, cells were washed twice in FACS staining buffer and subsequently stained for surface molecules for 20 min at room temperature. To fix the stained cells, the Cytofix/CytoPerm Fixation/ Permeabilization Kit (BD Biosciences, 554714) was used for 20 min at room temperature, while being protected from light. Subsequently, cells were washed twice with 1× perm/wash buffer and then stained for CAR19 and intracellular cytokines using antibodies in perm/wash buffer. This staining process was performed for 20 min at room temperature in the dark. Cells then underwent two additional washes with perm/wash buffer before being resuspended in FACS staining buffer for subsequent analysis. The following antigens were stained using the specified antibody clones: anti-FMC63 scFv (Y45, ACRO Biosystems, FM3-HPY53), CD3 (SK7, BD Biosciences, 564001), CD4 (OKT4, BioLegend, 317442), CD8a (RPA-T8, BioLegend, 301042), CD19 (HIB19, BD Biosciences, 561121), CD14 (M5E2, BD Biosciences, 561391), IL-3 (BVD3-1F9, BioLegend, 500606), IL-4 (MP4-25D2, BioLegend, 500834), IL-5 (TRFK5, BioLegend, 504306), IL-13 (JES10-5A2, BioLegend, 501916) and IL-31 (1D10B31, BioLegend, 659608). Cell-surface antibodies were used at a 1:100 dilution during staining, and intracellular antibodies at a 1:50 dilution. The samples were analysed on the Cytek Aurora flow cytometer, and data analysis was conducted using FlowJo v.10.8.0.

## Multiplexed secretomic assay

After stimulation with CD19-3T3 cells, around 30,000 magnetically enriched CAR$^+$ cells were processed for membrane staining (IsoPlexis, STAIN-1001-1). Subsequently, these cells were loaded onto the IsoCode chip (IsoPlexis, ISOCODE-1001-4), comprising 12,000 chambers prepatterned with an array of 32 cytokine capture antibodies. The chip was further incubated in the IsoLight machine for 16 h at 37 °C with 5% CO$_2$ supplementation. A cocktail of detection antibodies was then applied to detect the secreted cytokines, followed by fluorescence labelling. The resulting fluorescence signals were analysed using IsoSpeak v.2.8.1.0 (IsoPlexis) to determine the numbers of specific cytokine-secreting cells and the intensity level of each cytokine. For downstream analyses, the raw data pertaining to type-2-related cytokines, including IL-4, IL-5, IL-9, IL-10, IL-13 and IL-21, were extracted.

## Generation of *STAT6* and *GATA3* knockdown CAR T cells

To generate *STAT6* and *GATA3* knockdown CAR T cells, lentiviral particles containing short hairpin RNA against *STAT6* (shSTAT6) and shGATA3 were first produced in HEK293T cells. The cell lines were originally obtained from ATCC and tested negative for mycoplasma contamination. These cells were transfected with plasmids encoding pCMV-VSV-G (Addgene, 8454), pCMV-dR8.2 dvpr (Addgene, 8455), and either pLKO.1-puro_shSTAT6 (pLKO.1-puro, Addgene, 8453) or pLKO.1-puro_shGATA3, using the calcium phosphate transfection method. CAR T cells were subsequently spin-transduced with shSTAT6 or shGATA3 lentivirus particles on two consecutive days to ensure efficient transduction. The pool of shSTAT6 or shGATA3 sequences was as follows:

shSTAT6-1: 5′-CCGGAGCGGCTCTATGTCGACTTTCCTCGAGGAA AGTCGACATAGAGCCGCTTTTTTG-3′; shSTAT6-2: 5′-CCGGAGC ACCCTTGAGAGCATATATCTCGAGATATATGCTCTCAAGGGTGCTTTT TTG-3′; shGATA3-1: 5′-CCGGAGCCTAAACGCGATGGATATACTCGAGT ATATCCATCGCGTTTAGGCTTTTTTG-3′; shGATA3-2: 5′-CCGGCCC AAGAACAGCTCGTTTAACCTCGAGGTTAAACGAGCTGTTCTTGGGTTT TTG-3′. The resulting CAR T cells were expanded for an additional 2 days before use. The knockdown efficiency was assessed at both the gene and protein expression levels to confirm the efficacy of *STAT6* and *GATA3* silencing.

## Serial proteomic profiling of patient serum samples

The Cytokine Human Magnetic 30-Plex Panel (Invitrogen, LHC6003M) was used to detect serum proteins in a cohort comprising 33 patients in our discovery cohort. Serum samples, cryopreserved at −80 °C, spanning from 2 days before to 63 days after CTL019 infusion, were thawed and analysed according to the manufacturers' protocols. Measurements were conducted using the FlexMAP 3D instrument (Luminex), and data acquisition and analysis were performed using xPONENT software (Luminex). Moreover, the Olink Explore 384 panel (Olink Proteomics) was used to measure serum proteins in eight patients from our validation cohort, with all protein data reported in normalized expression values on a log$_2$ scale. In the discovery cohort, serum collections may have been performed on different days for various patients within a given time frame. In the validation cohort, serum collections were consistently conducted for all patients on specified days.

## Mice and tumour cell lines

NOD/SCID/IL-2Rγ$^{null}$ (NSG) mice (aged 6 weeks) were procured from Charles River Laboratory. All mice were housed in the Center of PhenoGenomics (CPG) animal facility at EPFL, kept in individually ventilated cages at 19–23 °C with 45–65% humidity and maintained under a 12 h–12 h dark–light cycle. All experimental procedures involving mice were ethically approved by Swiss authorities (Canton of Vaud, animal protocol ID 3533) and adhered to the guidelines set forth by the CPG of EPFL. Nalm6 cell lines, sourced from ATCC, were screened and confirmed to be free of mycoplasma contamination. These cells were stably transduced with GFP-luciferase lentivirus for downstream experimentation. Culturing of Nalm6 cells was conducted in RPMI medium supplemented with 10% FBS and 200 U ml$^{-1}$ penicillin– streptomycin (Gibco, 15140122).

## In vivo xenograft mouse studies

A total of $1 \times 10^6$ Nalm6-luciferase cells were i.v. injected into NSG mice to establish the leukaemia xenograft mouse model. Mice were randomized after tumour injection before initiating treatment. Then, 1 week later, $2 \times 10^6$ CAR T cells were adoptively infused through tail vein injection. Tumour growth was monitored weekly using the Xenogen IVIS fluorescence/bioluminescence imaging system (PerkinElmer). In brief, mice were intraperitoneally injected with bioluminescent substrate D-luciferin potassium salt (150 mg per kg; Abcam, ab143655). Then, 10 min after injection, the mice were anaesthetized and subjected to the luminescent imaging system to quantify tumour burden. The surviving mice were rechallenged with $1 \times 10^6$ Nalm6-luciferase cells 17 days after the CAR T infusion. Mice were euthanized when body weight loss was beyond 15% of the baseline weight, or any signs of discomfort were detected by the investigators or as recommended by the veterinarian who monitored the mice every other day.

## In vitro repeat stimulation assay

CTL019 cells were thawed and allowed to rest for 3–4 h before undergoing sequential staining with PE-labelled monoclonal anti-FMC63 scFv antibody and anti-PE MicroBeads, according to previously established protocols. CAR$^+$ cells, magnetically enriched using the MACS column, were then co-cultured with Nalm6 cells in a six-well plate at specified $E/T$ ratios. Throughout the assay period, the number of CAR T cells and Nalm6 cells was assessed daily using flow cytometry. Additional Nalm6 cells were added as necessary to maintain the designated $E/T$ ratio. Each evaluated condition was prepared with 3 or 4 technical replicates.

## Enhanced type 2 CAR T cell manufacture

The manufacturing process for enhanced type 2 CAR T cells largely adhered to previously established protocols, with a notable modification involving the addition of either 10 ng ml$^{-1}$ or 50 ng ml$^{-1}$ of IL-4 (R&D Systems, 204-IL-010) throughout the entire culture and expansion process. This supplementation was introduced in addition to the standard inclusion of 5 ng ml$^{-1}$ each of IL-7 (Miltenyi Biotec, 130-095-367) and IL-15 (Miltenyi Biotec, 130-095760) used in traditional manufacturing procedures.

## Flow cytometry

Mouse blood (50 µl) was collected from the tail at specified timepoints for peripheral CAR T cell analysis. The collected samples were resuspended in PBS with EDTA (2 mM), and the red blood cells were removed using ACK lysis buffer (Gibco, A1049201). For surface marker staining, cells were incubated with an antibody panel at 4 °C for 30 min, followed by live/dead staining. Cells were then washed and resuspended in PBS with 0.2% BSA for flow cytometry analysis. Intracellular cytokine staining was performed by first stimulating cells with a cell stimulation cocktail (Invitrogen, 00-4970-03) for 5 h at 37 °C to induce cytokine production. Subsequently, cells were stained for surface markers and live/dead dye as previously described, then fixed and permeabilized using the Cytofix/Cytoperm Kit (BD Biosciences). Intracellular staining with the indicated antibody panel was conducted according to the manufacturer's protocol. Data were collected using the Attune NxT Flow Cytometer with Attune NxT Software v.3 (Invitrogen) and analysed using FlowJo v.10.6.1 (Tree Star). Gate margins were determined by isotype controls and fluorescence-minus-one controls.

The following antibodies, each with their specified clones, were procured from BioLegend and used for flow cytometry analysis in both in vitro and in vivo functional assays: FAS (DX2, 305624), IL-13 (JES10-5A2, 501916), CD27 (LG.3A10, 124249), CD45RO (UCHL1, 304238), TNF (MAb11, 502940), CD3 (OKT3, 317306), GZMB (GB11, 515403), CD223 (LAG-3) (11C3C65, 369312), CD366 (TIM3) (F38-2E2, 345016), CD197 (CCR7) (G043H7, 353235), IL-4 (MP4-25D2, 500832), CD19 (HIB19, 302216), CD4 (OKT4, 317416), IL-5 (TRFK5, 504306), CD8 (SK1, 344724),

CD279 (PD-1) (EH12.2H7, 329952), KLRG1 (MAFA) (2F1/KLRG1, 138426), IFNγ (4S.B3, 502530) and the Zombie Aqua Fixable Viability Kit (423102). Monoclonal anti-FMC63 antibody (Y45, FM3-HPY53) was purchased from ACRO Biosystems. Cell-surface antibodies were used at a 1:100 dilution during staining, intracellular antibodies at a 1:50 dilution, and live/dead staining at a 1:1,000 dilution.

## Single-cell transcriptome data processing and analysis

A total of 44 paired scRNA-seq and CITE-seq libraries were sequenced; detailed data quality metrics are provided in Supplementary Table 2. The sequencing data underwent alignment to the GRCh38 human reference genome, followed by barcode and unique molecular identifier counting, ultimately generating a digital gene expression matrix using Cell Ranger v.6.1.2 (10x Genomics). The subsequent data analysis was conducted according to the Seurat v.4 pipeline[38]. The hashtag oligos expression was used to demultiplex cells back to their original sample of origin, while also identifying and excluding cross-sample doublets. Cells flagged as doublets (two barcodes detected) or lacking barcodes were omitted from the analysis. Only cells expressing a gene count ranging from 200 to 7,000 and exhibiting less than 10% mitochondrial gene content were retained for downstream analysis. For the whole-dataset analysis (Fig. 1b), a random grouping approach was implemented in which every 4 libraries were combined and designated as a single batch, resulting in 11 different batches. Subsequently, a fast integration method named reciprocal principal component analysis (PCA)[39] was used with the default parameters to mitigate potential batch effects and enable large-scale data integration. This involved splitting the dataset into Seurat objects based on sequencing batch, independent normalization and variable feature identification for each dataset. Integration and PCA were conducted on repeatedly variable features across datasets, with anchors identified using the FindIntegrationAnchors function and subsequent dataset integration with IntegrateData. Standard workflows for visualization and clustering were then implemented. For subclustering analyses of basal unstimulated or CD19-3T3-stimulated CAR T cells from both the discovery and validation cohorts, the sctransform normalization method in Seurat was used[40]. This method is specifically designed to capture sharper biological heterogeneities in scRNA-seq datasets, with no significant batch effects observed for these analyses. DEGs were identified using the FindMarkers function for pairwise comparisons between cell groups or clusters, applying a log-transformed fold change threshold of 0.25 to select significant genes. Moreover, module scores based on predefined gene sets were computed using the AddModuleScore function.

## scATAC and gene co-profiling data processing and analysis

Cell Ranger ARC v.2.0.2 (10x Genomics) was used to perform sample demultiplexing, barcode processing, identification of open chromatin regions, and simultaneous counting of transcripts and peak accessibility in single cells from the sequenced data. The output per barcode matrices underwent joint RNA and ATAC analysis using Signac (v.1.12.0)[41] and Seurat (v.4)[38]. Per-cell quality control metrics were computed, including the nucleosome banding pattern (stored as nucleosome_signal) and the transcriptional start site (TSS) enrichment score for the ATAC component. These metrics were used to identify and remove outliers, with the quality report of each sample meticulously documented in Supplementary Table 4. Quality filtering criteria adhered to the default settings. Specifically, cells were retained if they exhibited an ATAC peak count ranging from 1,000 to 100,000, a gene count ranging from 1,000 to 25,000, a nucleosome_signal below 2 and a TSS enrichment score exceeding 1. To enhance the accuracy of peak identification, we used MACS2 v.2.2.9.1 with the CallPeaks function, a widely used tool for peak calling in chromatin accessibility analysis[42]. Subsequently, we constructed a joint neighbour graph representing both gene expression and DNA accessibility

measurements using weighted nearest-neighbour methods in Seurat v.4. To investigate potential regulatory elements for genes of interest, we used the LinkPeaks function. This method identifies sets of peaks that may regulate gene expression by computing the correlation between gene expression and accessibility at nearby peaks, while correcting for biases due to GC content, overall accessibility and peak size. In preparation for motif analyses, we used the AddMotifs function to integrate DNA sequence motif information into the dataset. Furthermore, we computed a per-cell motif activity score using chromVAR[43]. For footprinting analysis of motifs with positional information, we used the Footprint function to gather and store all necessary data within the assay.

### L–R interaction analysis

The R toolkit Connectome (v.1.0.0)[44] was used to investigate cell–cell connectivity patterns using ligand and receptor expression values from our scRNA-seq datasets with the default parameters. The normalized Seurat object served as input, and cluster identities were used to define nodes in the interaction networks, resulting in an edge list connecting pairs of nodes through specific L–R mechanisms. We selected top-ranked interaction pairs for visualization, prioritizing those that are more likely to be biologically and statistically significant based on the scaled weights of each pair. The thickness of edges is directly proportional to correlation weights, with wider edges indicating a higher level of interaction. The sources.include and targets.include parameters were used to specify the source cluster emitting ligand signals and the target cluster expressing receptor genes that sense the ligands.

### Ingenuity pathway analysis

Ingenuity Pathway Analysis (Qiagen)[45] was used to identify the underlying signalling pathways regulated by the DEGs characterizing each identified cluster or response group. To achieve this, the DEG list, along with corresponding fold change values, $P$ values and adjusted $P$ values for each gene, were loaded into the dataset. Using the Ingenuity knowledge base (genes only) as a reference set, core expression analysis was performed. T-cell-related signalling pathways were specifically selected from the identified canonical pathways to represent the primary functional profile of each group. The activation or inhibition level of specific pathways was determined using the $z$-score metric. Conceptually, the $z$-score serves as a statistical measure, assessing how closely the actual expression pattern of molecules in our DEG dataset aligns with the expected pattern based on literature for a particular annotation ($z > 0$, activated/upregulated; $z < 0$, inhibited/downregulated; $z \geq 2$ or $z \leq -2$ can be considered to be significant). The significance of each identified signalling pathway was determined using the right-tailed Fisher's exact test, with the $P$ value reflecting the probability of association between molecules from our scRNA-seq dataset and the canonical pathway reference dataset.

### Statistical analysis

Statistical analyses were conducted using Prism v.10 (GraphPad) or R v.4.3.1. Unless otherwise specified, data are presented as mean ± s.e.m. For comparisons involving three or more groups, we used one-way ANOVA with Tukey's multiple-comparison test. For comparisons between two groups, two-tailed Mann–Whitney $U$-tests were used for nonparametric data, two-tailed Student's $t$-tests for parametric data and two-tailed Wilcoxon matched-pairs signed-rank tests for paired nonparametric data. For single-cell-level comparisons, typically depicted in violin plots (as shown in Extended Data Fig. 7b,j), the normalized expression value of the top 10% single cells from each patient in each BCA group was included in the comparison.

### Reporting summary

Further information on research design is available in the Nature Portfolio Reporting Summary linked to this article.

## Data availability

Raw and processed single-cell sequencing data for this study are available at the NCBI Gene Expression Omnibus under accession number GSE262072.

## Code availability

Single-cell analysis codes used in this study are available from the corresponding authors on reasonable request.

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

**Acknowledgements** We thank the patients and their families for their participation in this study; the members of the following research cores at the University of Pennsylvania and Children's Hospital of Philadelphia: the Translational and Correlative Studies Laboratory, for their provision of standardized flow cytometry, qPCR and cytokine multiplexing analyses, as well as for their expert biobanking of patient specimens involved in CAR T cell trials, the Clinical Cell and Vaccine Production Facility for cell processing and biobanking, the Human Immunology Core for providing HD cells and analytical support, and the Flow Cytometry Core for flow cytometry equipment maintenance and access to electronic sorters; the members of the research and medical teams involved in the collection of patient samples, CAR T manufacturing, clinical administration and monitoring, and patient consent and recruitment; the staff at the Yale Center for Genome Analysis for support in building sequencing libraries and the Yale High Performance Computing clusters for facilitating computational data analysis. The research was supported by Stand Up To Cancer (SU2C) (Convergence 2.0 Grant to R.F.), Packard Fellowship for Science and Engineering (2012-38215 to R.F.), National Institutes of Health (1U01CA232361 to S.A.G.; R01CA241762-01 and 1U01CA269409-01 to J.J.M.) and the Swiss National Science Foundation (315230_204202, IZLCZO_206035 and CRSII5_205930 to L.T.). This material is based on work supported under a collaboration by SU2C, a program of the Entertainment Industry Foundation and the Society for Immunotherapy of Cancer.

**Author contributions** Conceptualization: R.F., Z.B., S.A.G., J.J.M., C.H.J. and L.T. Data curation: Z.B., B.F., S.E.M. and C.D. Formal analysis: Z.B. and B.F. Investigation: Z.B., B.F., B.C.d.O., C.G., B.T., L.Y., Z.Z., L.P., G.Sferruzza, L.Z., X.Z. and A.B. Methodology: Z.B., R.F., J.J.M., B.F. and L.T. Project administration: Z.B. and R.F. Resources: Z.B., B.F., S.E.M., Z.Z., L.P., J.K., G.Su and M.Y. Scientific discussion: P.G.C. and S.C. Funding acquisition: R.F., S.A.G., J.J.M., C.H.J. and L.T. Writing—original draft: Z.B. Writing—review and editing: J.J.M., R.F., B.F., L.T., S.E.M. and P.G.C.

**Competing interests** R.F. is scientific founder and adviser for IsoPlexis, Singleron Biotechnologies and AtlasXomics. The interests of R.F. were reviewed and managed by Yale University Provost's Office in accordance with the University's conflict of interest policies. S.A.G. reports grants, personal fees and other support from Novartis and Vertex; grants from Jazz, Kite and Servier; personal fees and/or scientific advisory boards from Roche, GSK, CBMG, Janssen/J&J, Jazz, Adaptimmune, TCR2, Cellectis, Juno, Allogene and Cabaletta; in addition, S.A.G. has a patent for Toxicity management for antitumour activity of CARs, WO 2014011984 A1 issued. J.J.M. and C.H.J. hold patents related to CAR T cell manufacturing and biomarker discovery. C.H.J. is a scientific founder of Tmunity Therapeutics and DeCART Therapeutics, and is a member of the scientific advisory boards of AC Immune, BluesphereBio, Cabaletta, Carisma, Cartog-raphy, Cellares, Celldex, Decheng, Poseida, Verismo, WIRB-Copernicus and Ziopharm. L.T. is a co-founder, share-holder and advisor for Leman Biotech. The interests of L.T. were reviewed and managed by EPFL. S.C. is a co-founder of EvolveImmune, Cellinfinity, NumericGlobal and Chen Consulting, all unrelated to this study. The other authors declare no other competing interests.

**Additional information**
**Correspondence and requests for materials** should be addressed to Li Tang, Carl H. June, J. Joseph Melenhorst, Stephan A. Grupp or Rong Fan.

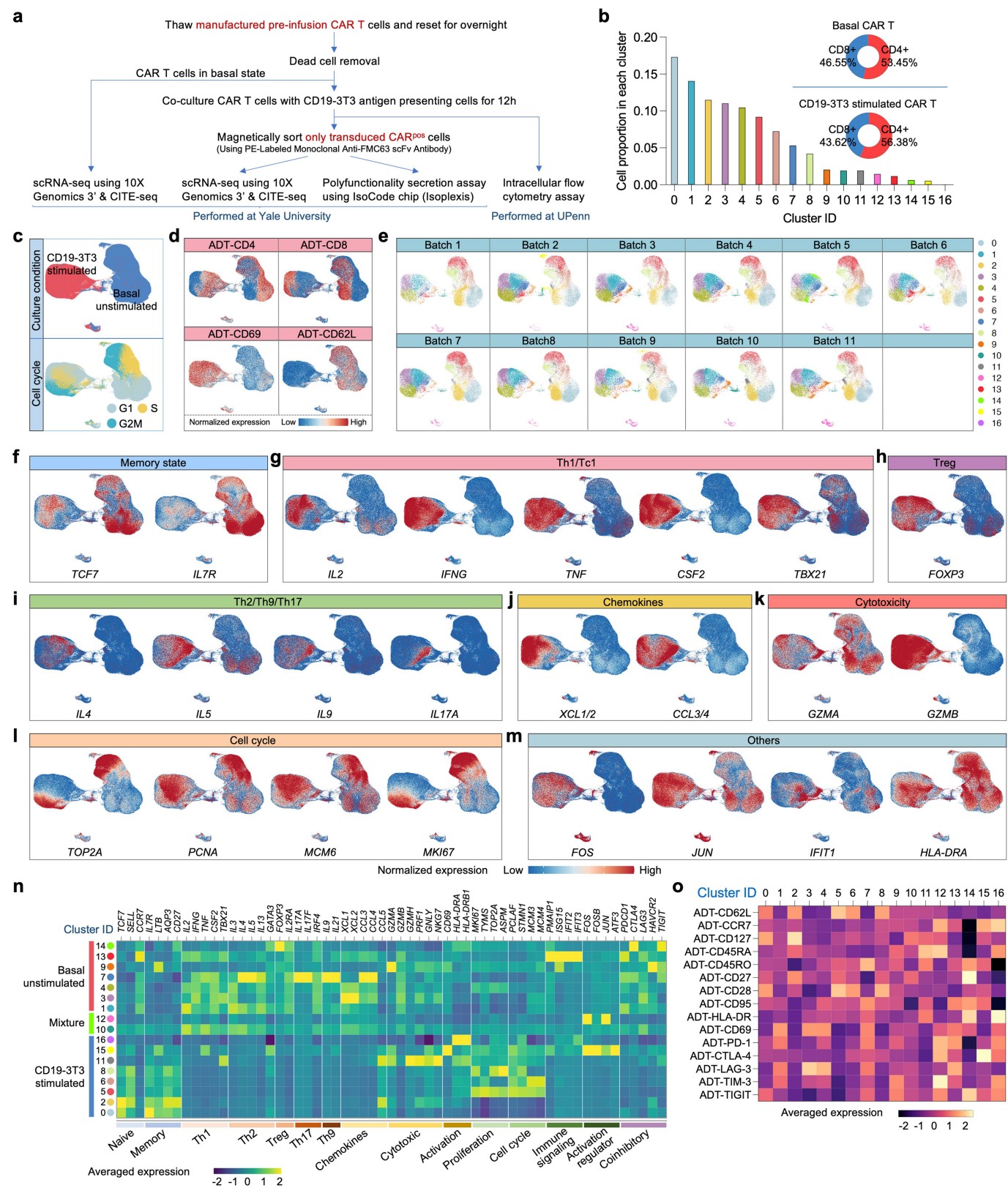

**Extended Data Fig. 1 | Single-cell multi-omics profiling of pre-infusion CAR T cells. a**, Experimental pipeline of this study. **b**, Cell proportion in each identified cluster, and the ratio of CD4+ and CD8+ cells in each stimulation condition determined by CITE-seq surface protein data. **c**, UMAP distribution of all the single cells grouped by in vitro stimulation condition (upper panel) or cell cycle (lower panel). **d**, Expression of surface protein ADT-CD69 (activated T cell marker), ADT-CD62L (naïve T cell marker), ADT-CD4 and ADT-CD8 (subtype T cell marker) on the UMAP. **e**, UMAP distribution of 695,819 single

CAR T cells split by sequencing batch, suggesting minimum batch effect. **f**–**m**, Expression distribution of memory states (**f**), Th1/Tc1 (**g**), Treg (**h**), Th2/Th9/Th17 (**i**), selected chemokines (**j**), cytotoxicity (**k**), cell cycle (**l**), and other (**m**) gene markers on the UMAP shown in Fig. 1b. **n**, CAR T cell signatures within each identified cluster based on cell states, functions, regulations, or subsets. The heatmap illustrates the averaged expression of curated T cell marker genes in each cluster. **o**, Heatmap showing the average expression level of ADT proteins of all the single cells in each identified cluster.

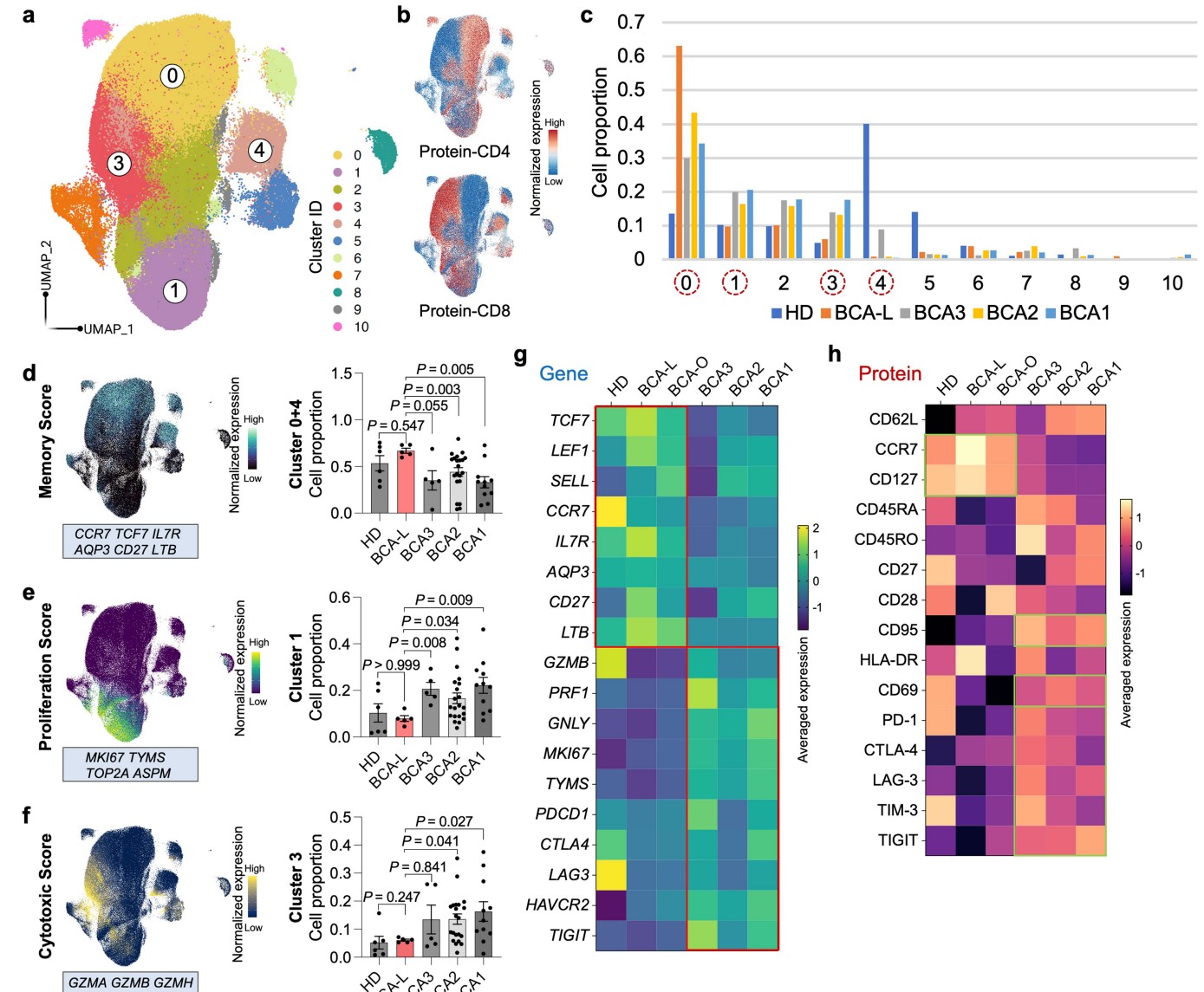

**Extended Data Fig. 2 | Clinical persistence correlates with characteristics in basal CAR T cells. a**, UMAP clustering of basal CAR T cells from the Discovery Cohort patients and donors. **b**, Expression distribution of surface protein ADT-CD4 and ADT-CD8 on the UMAP. **c**, Comparison of cell proportion in each identified cluster between persistence groups. **d**, Expression distribution of Memory Score on the UMAP, and the cell proportion comparison in Cluster 0 + 4. **e**, Expression distribution of Proliferation Score on the UMAP, and the cell proportion comparison in Cluster 1. **f**, Expression distribution of Cytotoxic Score on the UMAP, and the cell proportion comparison in Cluster 3. **g**, Heatmap showing the average expression level of feature genes of all the basal CAR T cells in each BCA group. **h**, Heatmap showing the average expression level of ADT proteins of all the basal CAR T cells in each BCA group. In **d**–**f**, genes defining each module are listed below. Scatter plot shows mean ± s.e.m. from $n = 48$ (**d**–**f**) patients or healthy donors. Significance levels were calculated with two-tailed Mann-Whitney test (**d**–**f**).

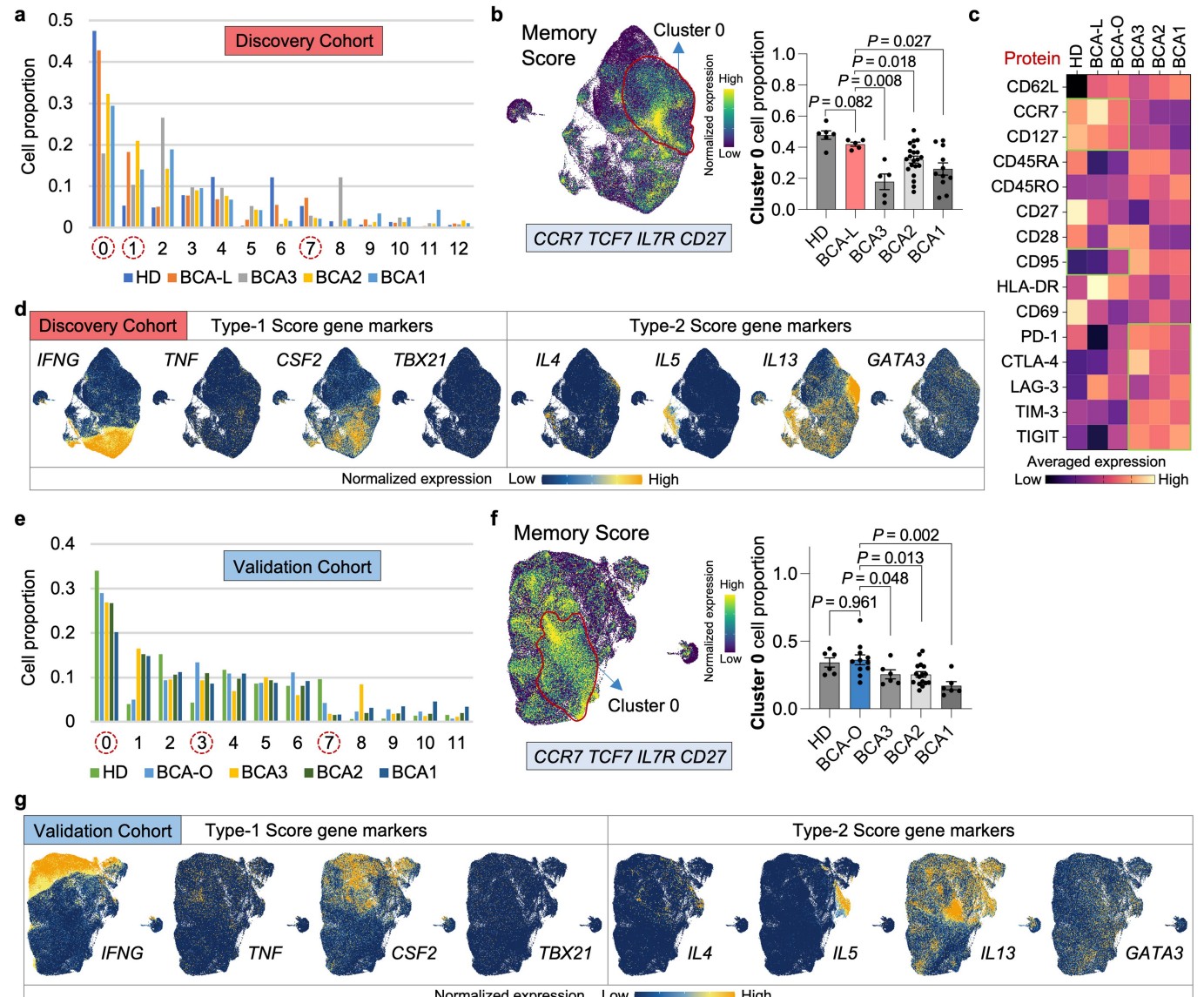

**Extended Data Fig. 3 | Transcriptomic clustering and proteomic analysis of activated CAR T cells. a**, Cell proportion comparison in each identified cluster of the Discovery Cohort UMAP (Fig. 2a) across persistence groups. **b**, Expression distribution of Memory Score on the Discovery Cohort UMAP in Fig. 2a, and the cell proportion comparison in Cluster 0. **c**, Heatmap showing the average expression levels of ADT proteins in all activated CAR T cells within each BCA group. **d**, Expression distribution of each gene defining the Type-1 and Type-2 Score on the Discovery Cohort UMAP in Fig. 2a. **e**, Cell proportion comparison in each identified cluster of the Validation Cohort UMAP (Fig. 2c) across persistence groups. **f**, Expression distribution of Memory Score on the Validation Cohort UMAP in Fig. 2c, and the cell proportion comparison in Cluster 0. **g**, Expression distribution of each gene defining the Type-1 and Type-2 Score on the Validation Cohort UMAP in Fig. 2c. In **b** and **f**, genes defining the module are listed below. Scatter plot shows mean ± s.e.m. from n = 48 (**b**) and n = 46 (**f**) patients or healthy donors. Significance levels were calculated with two-tailed Mann-Whitney test (**b**, **f**).

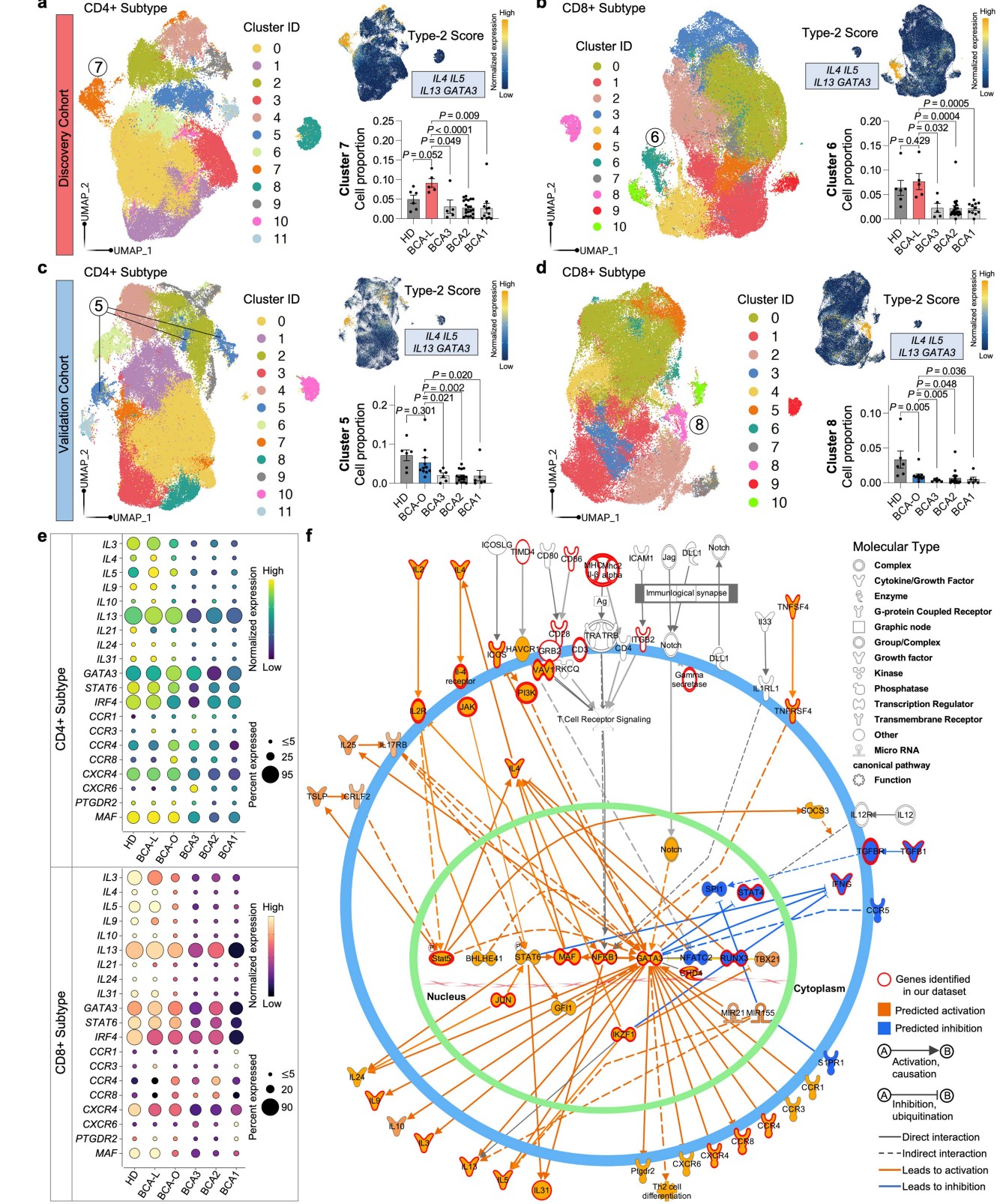

**Extended Data Fig. 4 |** See next page for caption.

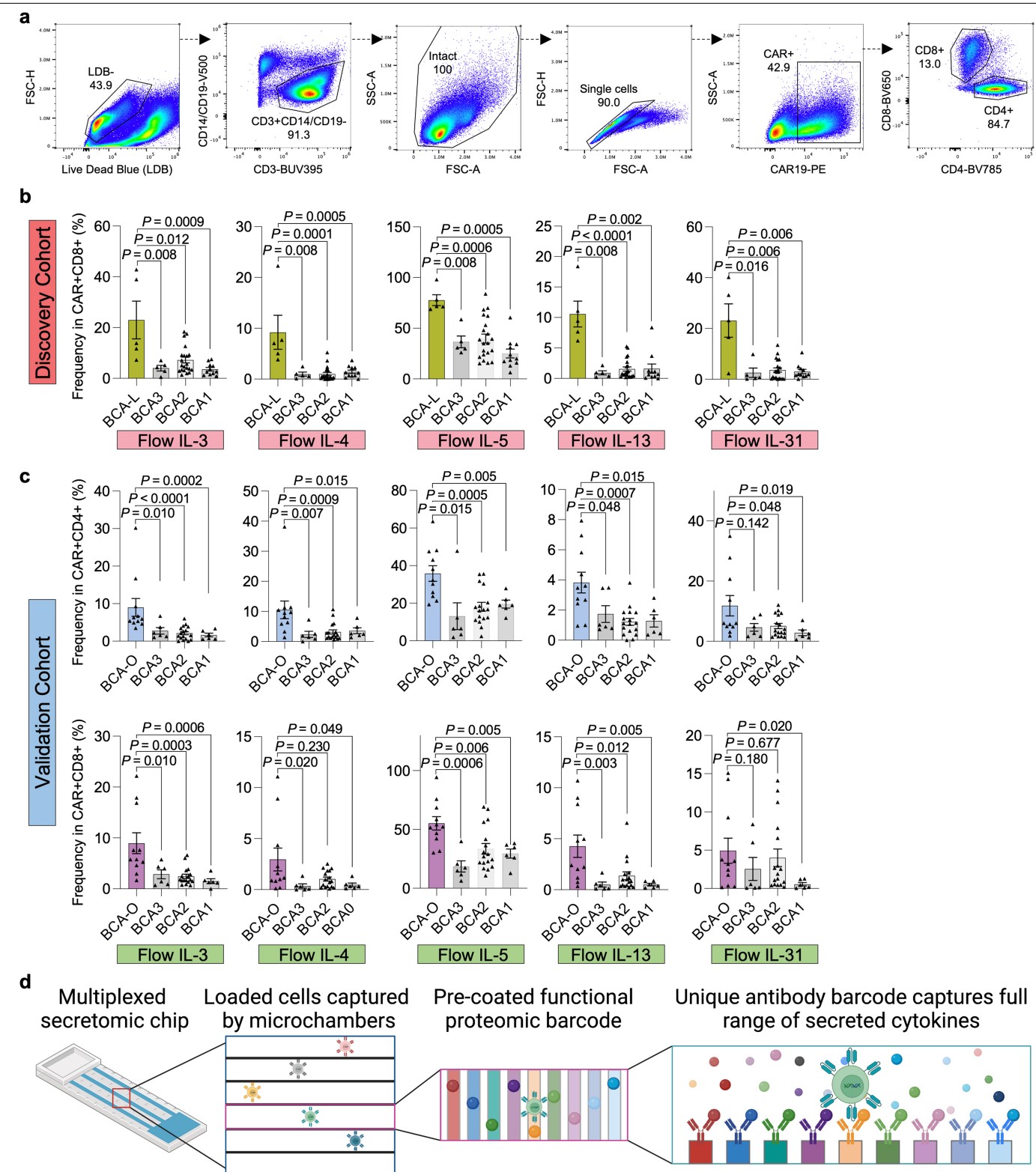

**Extended Data Fig. 5 | Independent evaluation of type 2 cytokines using flow cytometry and multiplexed secretomic assay. a**, Gating strategy employed for flow cytometry data analysis. Live Dead Blue (LDB) was utilized for live cell selection, followed by CD3 + CD14/CD19 − T cell gating and intact, single-cell filtering. CAR (FMC63) expression was employed for the selection of successfully transduced CAR+ cells, with subsequent analysis of CD4+ and CD8+ subpopulations conducted separately. **b**, Frequency comparison of the type-2-cytokine+ population in CD8 + CAR+ cells between persistence groups in the Discovery Cohort. **c**, Frequency comparison of the type-2-cytokine+ population in CD4+ or CD8 + CAR+ cells between persistence groups in the Validation Cohort. **d**, Schematic principle of multiplexed secretomic assay to measure functional cytokines. The diagram was created using BioRender. Scatter plot shows mean ± s.e.m. from $n = 42$ (**b**) and $n = 40$ (**c**) patients. Significance levels were calculated with two-tailed Mann-Whitney test (**b**, **c**).

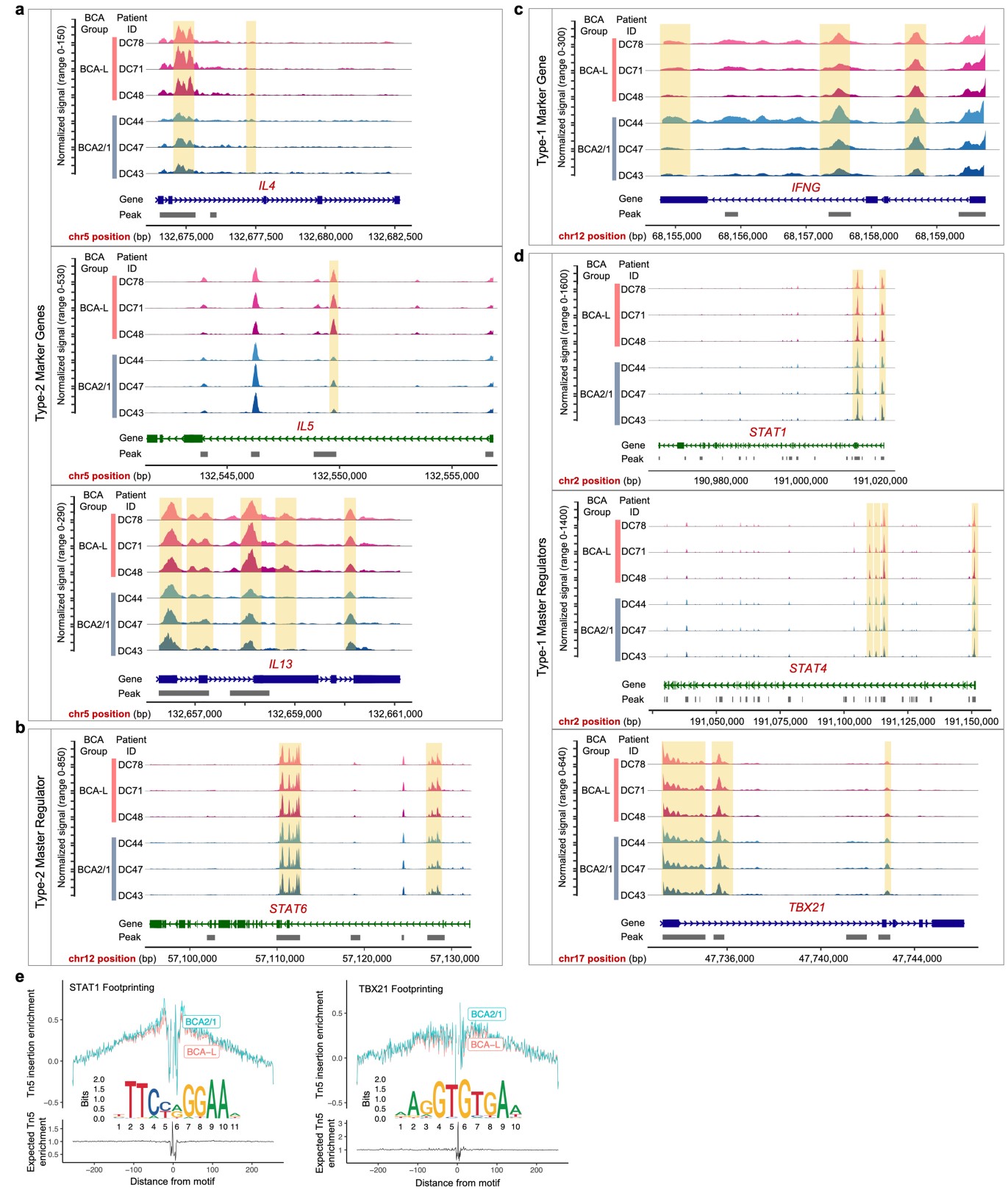

**Extended Data Fig. 6 | Single-cell ATAC analysis of type 1 or type 2 markers of activated CAR T cells. a–d**, Pseudo-bulk chromatin accessibility tracks in the genomic region of type 2 marker genes (**a**), type 2 master regulator (**b**), type 1 marker gene (**c**), or type 1 master regulators (**d**), depicted separately for each patient. The enhancer elements predicted by ENCODE within the region of each gene are highlighted in a light-yellow shade. **e**, Motif footprinting trace showing transcription factor binding dynamics of STAT1 or TBX21 in the two patient groups, alongside their respective position weight matrices.

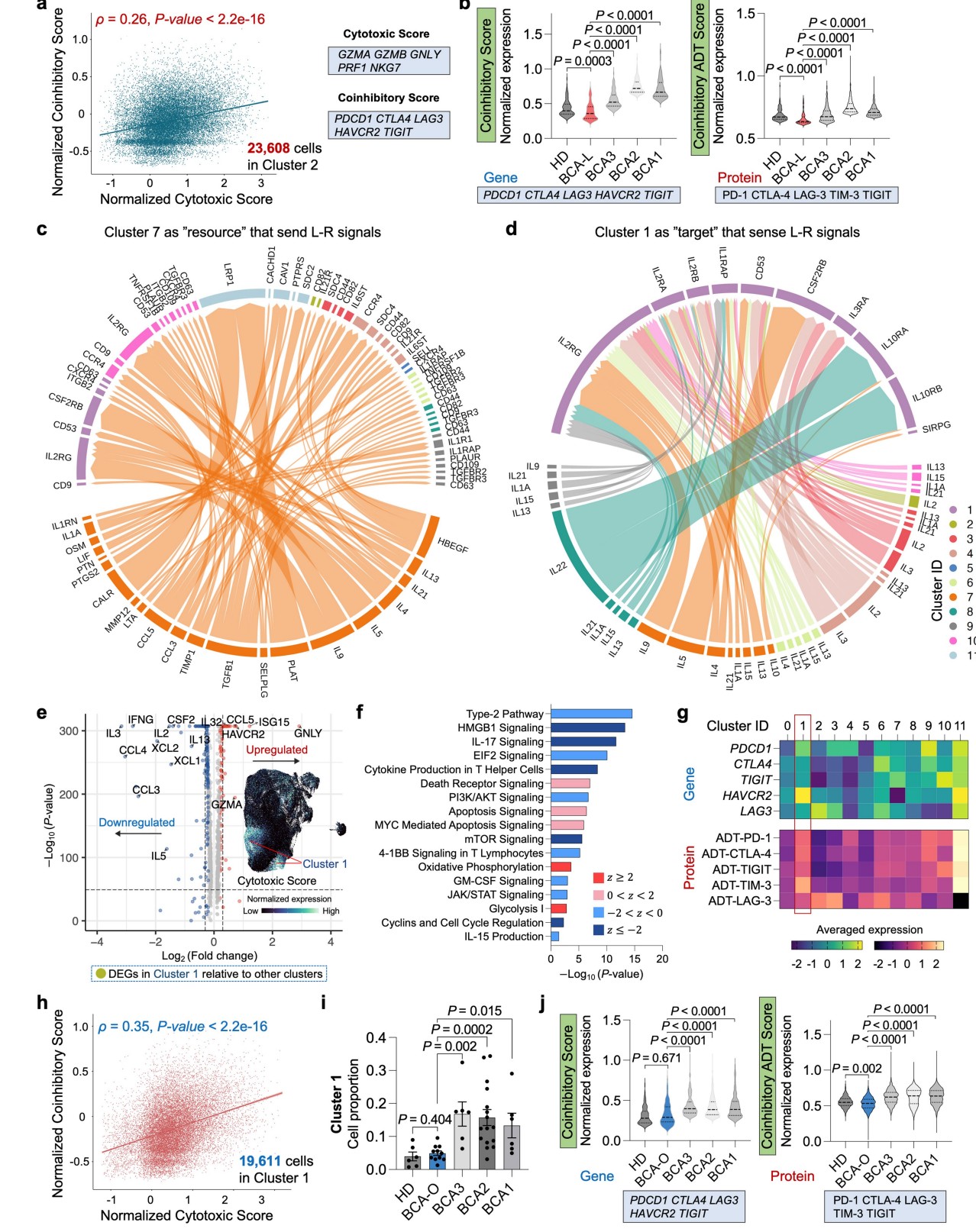

**Extended Data Fig. 7** | See next page for caption.

**Extended Data Fig. 7 | Ligand-receptor regulatory analysis of type 2 CAR T cells. a**, Correlation between Cytotoxic Score and Coinhibitory Score of all the Cluster 2 cells in Fig. 2a. The regression line is indicated, with the 95% confidence area shown in shaded colour. Spearman correlation coefficient and the associated $p$-value are shown. Genes defining each module are listed. **b**, Single-cell expression comparisons of Coinhibitory Score and Coinhibitory ADT Score in Cluster 2 cells between persistence groups in the Discovery Cohort. **c**, Identification of ligand-receptor (L-R) interactions originating from type 2 enriched Cluster 7 cells in Fig. 2c, predominantly interacting with Cluster 1 cells through type 2 L-R pairs. The thickness of edges is proportional to correlation weights, and edge colour corresponds to the Cluster ID. **d**, Identification of L-R interactions targeting Cluster 1 cells in Fig. 2c, revealing that cells from the majority of other clusters predominantly regulate these cells through type 2 L-R pairs. **e**, Differentially expressed genes (DEGs) specific to Cluster 1 in comparison to all other clusters in Fig. 2c, along with the expression distribution of the Cytotoxic Score. Genes defining this module include *GZMA*, *GZMB*, *GZMH*, *GNLY*, *PRF1*, and *NKG7*. **f**, Corresponding signalling pathways regulated by the DEGs identified in Cluster 1. Pathway terms are ranked by $-\log 10$ ($p$-value).

$z$ score is computed and used to reflect the predicted activation level ($z > 0$, activated/upregulated; $z < 0$, inhibited/downregulated; $z \geq 2$ or $z \leq -2$ can be considered significant). **g**, Heatmap showing the average expression levels of coinhibitory-related genes or ADT proteins across all single cells within each identified cluster in Fig. 2c. The expression of TIM-3 and its encoding gene HAVCR2 is observed to be high in Cluster 1. **h**, Correlation between Cytotoxic Score and Coinhibitory Score of all the Cluster 1 cells in Fig. 2c. The regression line is indicated, with the 95% confidence area shown in shaded colour. Spearman correlation coefficient and the associated $p$-value are shown. Genes defining each module are listed in (**a**). **i**, Comparison of cell proportion in Cluster 1 between persistence groups in the Validation Cohort. **j**, Single-cell expression comparisons of Coinhibitory Score and Coinhibitory ADT Score in Cluster 1 cells between persistence groups in the Validation Cohort. Violin plot shows expression distribution of the top 10% single cells from each individual among $n = 48$ (**b**) and $n = 46$ (**j**) patients or healthy donors. Scatter plot shows mean ± s.e.m. from $n = 46$ (**i**) patients or healthy donors. Significance levels were calculated with two-tailed Spearman's rank correlation test (**a**, **h**), two-tailed Mann-Whitney test (**b**, **e**, **i**, and **j**), or right-tailed Fisher's Exact Test (**f**).

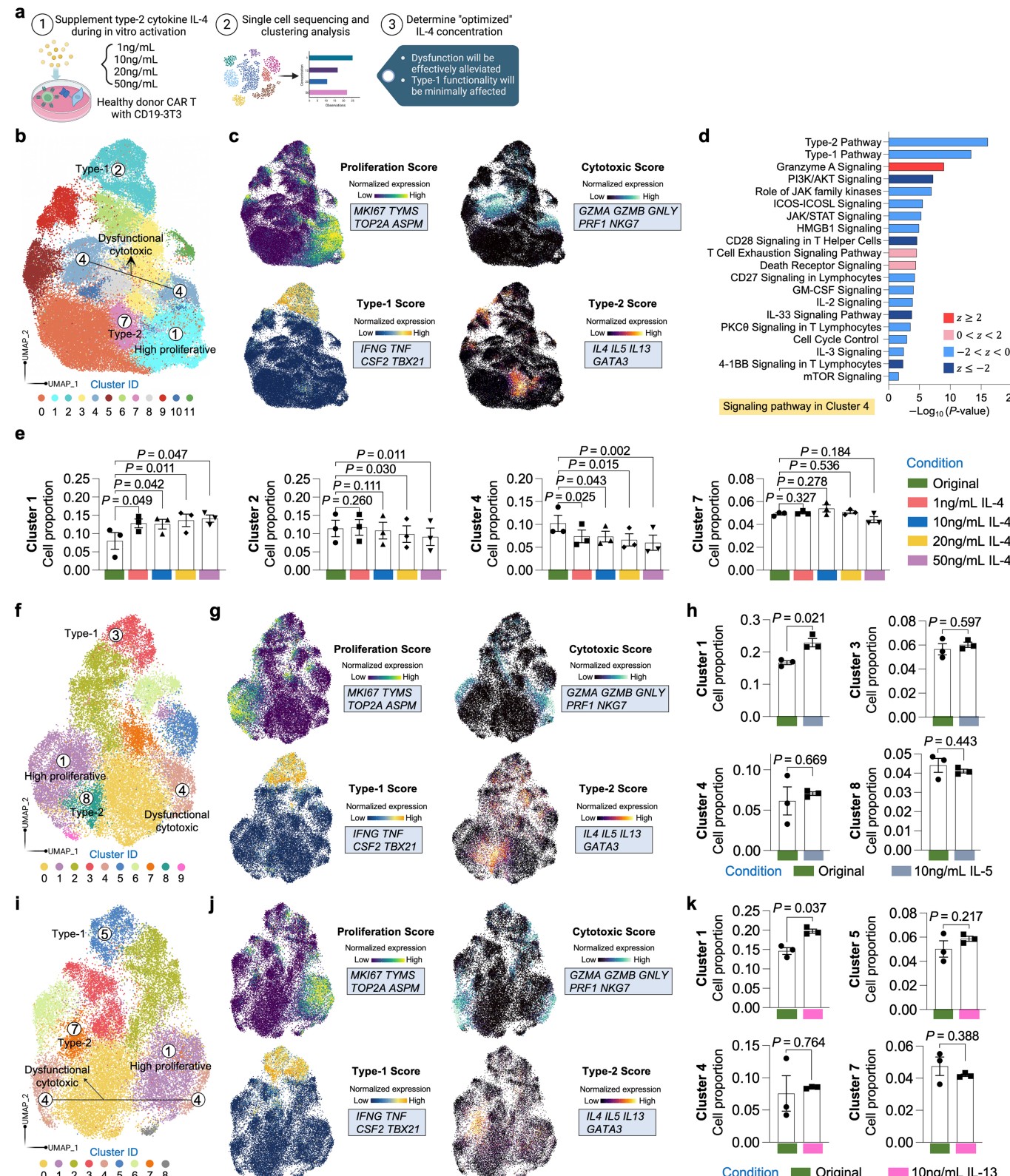

**Extended Data Fig. 8** | See next page for caption.

**Extended Data Fig. 8 | Clustering analysis of donor CAR T cell population response to type 2 cytokines. a**, Schematic outlining the experimental design for optimizing IL-4 concentration using donor CAR T cells. Three donors (ND317, ND365, ND502) were used for this study. The diagram was created using BioRender. **b**, UMAP clustering of CAR T cells from healthy donors, comparing conditions with and without varying concentrations of IL-4 added during in vitro CAR-specific activation. Characteristic clusters enriched for high proliferative (Cluster 1), type 1 (Cluster 2), type 2 (Cluster 7), and dysfunctional cytotoxic (Cluster 4) CAR T are indicated. **c**, Expression distribution of Proliferation Score, Type-1 Score, Type-2 Score, and Cytotoxic Score on the UMAP in (**b**). **d**, Corresponding signalling pathways regulated by the DEGs identified in Cluster 4 in (**b**). Functional pathways are downregulated in this cluster, whereas T cell exhaustion and death receptor signalling are activated. Pathway terms are ranked by −log 10 ($p$-value). $z$ score is computed and used to reflect the predicted activation level ($z > 0$, activated/upregulated; $z < 0$, inhibited/downregulated; $z \geq 2$ or $z \leq -2$ can be considered significant). **e**, Comparison of cell proportions in Clusters 1, 2, 4, and 7 identified in (**b**) between conditions. "Original" denotes no IL-4 added during CAR-specific activation. **f**, UMAP clustering of CAR T cells from healthy donors, with and without 10 ng/mL of IL-5 added during in vitro CAR-specific activation. Characteristic clusters enriched for high proliferative (Cluster 1), type 1 (Cluster 3), type 2 (Cluster 8), and dysfunctional cytotoxic (Cluster 4) CAR T are indicated. **g**, Expression distribution of Proliferation Score, Type-1 Score, Type-2 Score, and Cytotoxic Score on the UMAP in (**f**). **h**, Comparison of cell proportions in Clusters 1, 3, 4, and 8 identified in (**f**) between conditions. "Original" denotes no IL-5 added during CAR-specific activation. **i**, UMAP clustering of CAR T cells from healthy donors, with and without 10 ng/mL of IL-13 added during in vitro CAR-specific activation. Characteristic clusters enriched for high proliferative (Cluster 1), type 1 (Cluster 5), type 2 (Cluster 7), and dysfunctional cytotoxic (Cluster 4) CAR T are indicated. **j**, Expression distribution of Proliferation Score, Type-1 Score, Type-2 Score, and Cytotoxic Score on the UMAP in (**i**). **k**, Comparison of cell proportions in Clusters 1, 4, 5, and 7 identified in (**i**) between conditions. "Original" denotes no IL-13 added during CAR-specific activation. In **c**, **g**, and **j**, Genes defining each module are listed below. Scatter plot shows mean ± s.e.m. from $n = 3$ (**e**, **h**, **k**) healthy donors. Significance levels were calculated with right-tailed Fisher's Exact Test (**d**), or two-tailed Wilcoxon matched-pairs signed rank test (**e**, **h**, and **k**).

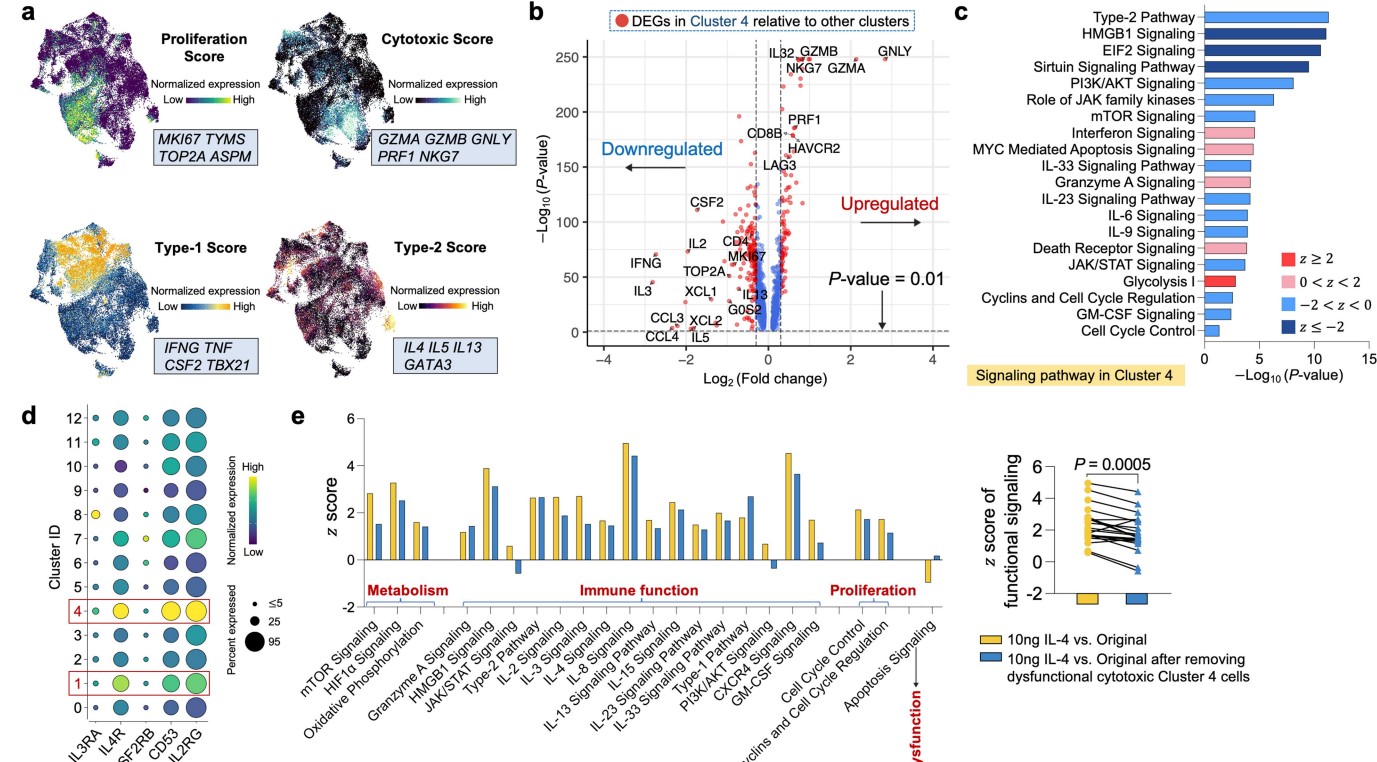

**Extended Data Fig. 9 | Clustering analysis of patient CAR T cells activated with IL-4 supplementation. a**, Expression distribution of Proliferation Score, Type-1 Score, Type-2 Score, and Cytotoxic Score on the UMAP in Fig. 3j. Genes defining each module are listed below. **b**, Differentially expressed genes (DEGs) specific to Cluster 4 in comparison to all other clusters in Fig. 3j. **c**, Corresponding signalling pathways regulated by the DEGs identified in Cluster 4 in Fig. 3j. Functional pathways are downregulated in this cluster, whereas T cell exhaustion and death receptor signalling are activated. **d**, Dot plot showing expression profile of type 2 receptor genes across all clusters identified in Fig. 3j, with notable high expression observed in Clusters 1 and 4. The size of circle represents proportion of single cells expressing the gene, and the colour shade indicates normalized expression level. **e**, Comparison of signalling pathways in BCA2 patient CAR T cells supplemented with 10 ng/mL IL-4 relative to the original condition, with or without the exclusion of dysfunctional cytotoxic Cluster 4 cells in Fig. 3j. A statistical comparison of the activation z score for functional signalling pathways ($n = 21$), including those regulating metabolism, immune function, and proliferation, was conducted. In **c** and **e**, pathway terms are ranked by $-\log 10$ ($p$-value). z score is computed and used to reflect the predicted activation level ($z > 0$, activated/upregulated; $z < 0$, inhibited/downregulated; $z \geq 2$ or $z \leq -2$ can be considered significant). Significance levels were calculated with two-tailed Mann-Whitney test (**b**), right-tailed Fisher's Exact Test (**c**), or two-tailed Wilcoxon matched-pairs signed rank test (**e**).

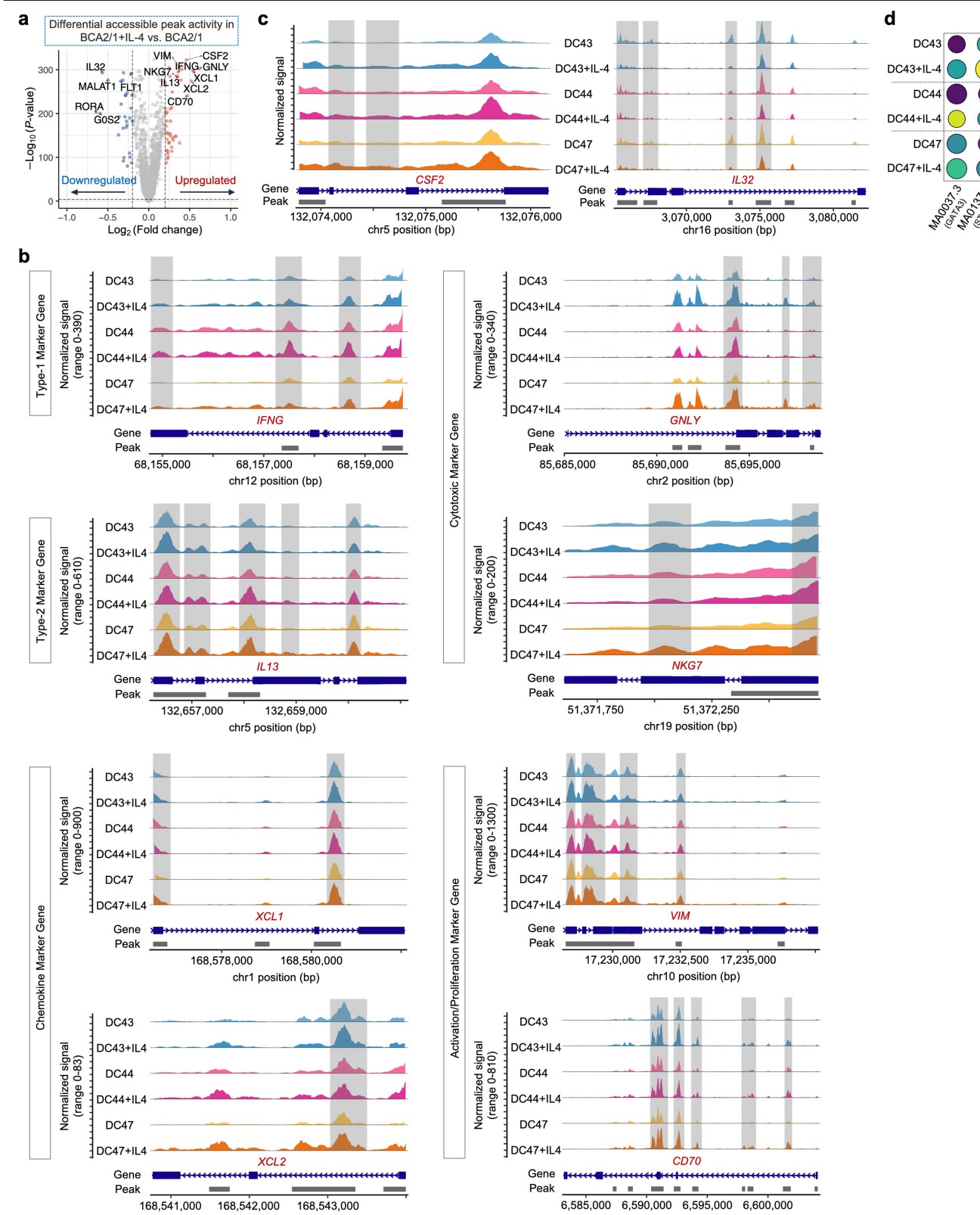

**Extended Data Fig. 10 | Single-cell ATAC analysis of patient CAR T cell response to IL-4. a**, Volcano plot showing differential accessible peak activities in BCA2 or 1 CAR T cells with and without the addition of 10 ng/mL IL-4. **b**, Pseudo-bulk chromatin accessibility tracks in the genomic region of top-ranked genes upregulated in 10 ng/mL IL-4 treated BCA2 or 1 CAR T cells compared to the original condition, including type 1 marker gene, type 2 marker gene, cytotoxicity marker genes, chemokine marker genes or activation/proliferation marker genes, depicted separately for each patient with and without the addition of IL-4. **c**, Pseudo-bulk chromatin accessibility tracks in the genomic region of *CSF2* and *Il32*, depicted separately for each patient with and without the addition of IL-4. **d**, Expression profile of type 2 motif MA0037.3 (GATA3) and type 1 motif MA0690.1 (STAT1) across each patient with and without the addition of IL-4. In **b** and **c**, the enhancer elements predicted by ENCODE within the region of each gene are highlighted in a light-grey shade. Significance levels were calculated with two-tailed Mann-Whitney test (**a**).

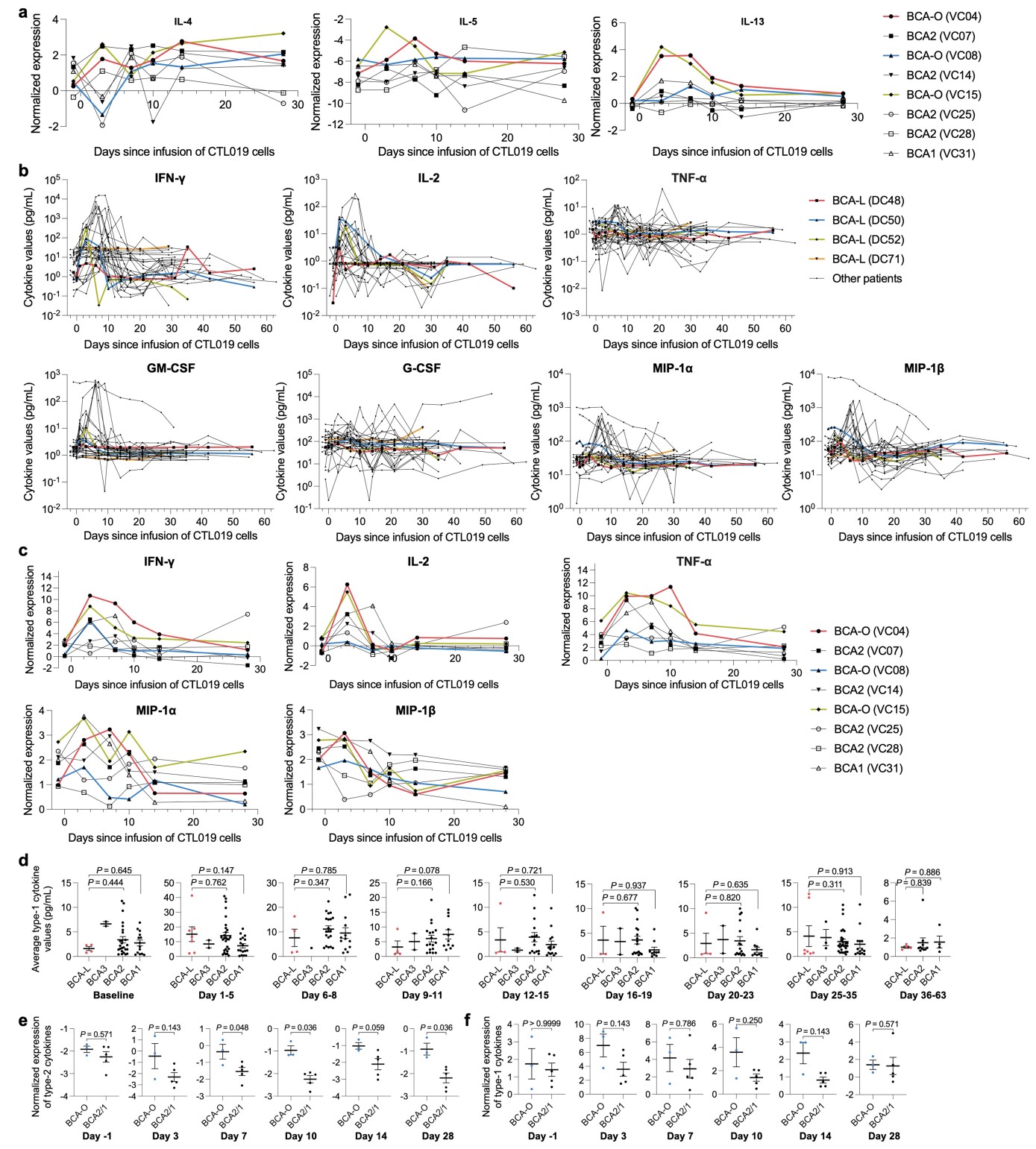

**Extended Data Fig. 11 | Longitudinal evaluation of cytokine profile in post-treatment sera. a**, Longitudinal levels of type 2 cytokines in patients from the Validation Cohort. Three patients (with Patient ID) in the BCA-O group are individually labelled with colour lines. **b**, **c**, Longitudinal levels of type 1 cytokines and representative chemokines in patients from the Discovery Cohort (**b**) and the Validation Cohort (**c**). Seven patients (with Patient ID) in the BCA-L or BCA-O group are individually labelled with colour lines. **d**, Comparison of the average type 1 cytokine levels between persistent groups in the Discovery Cohort. The value is the averaged expression of IFN-γ, IL-2, and TNF-α. **e**, Comparison of the average type 2 cytokine levels at multiple time

points between persistent groups in the Validation Cohort. The value represents the average of the normalized expression levels of IL-4, IL-5, and IL-13. **f**, Comparison of the average type 1 cytokine levels between persistent groups in the Validation Cohort. The value represents the average of the normalized expression levels of IFN-γ, IL-2, and TNF-α. Scatter plot shows mean ± s.e.m. from *n* = 45 (Baseline), *n* = 54 (Day 1–5), *n* = 38 (Day 6–8), *n* = 35 (Day 9–11), *n* = 36 (Day 12–15), *n* = 32 (Day 16–19), *n* = 32 (Day 20–23), *n* = 54 (Day 25–35), *n* = 18 (Day 36–63) (**d**,) and *n* = 8 (**e**, **f**) measurements. Significance levels were calculated with two-tailed Mann-Whitney test (**d**–**f**).

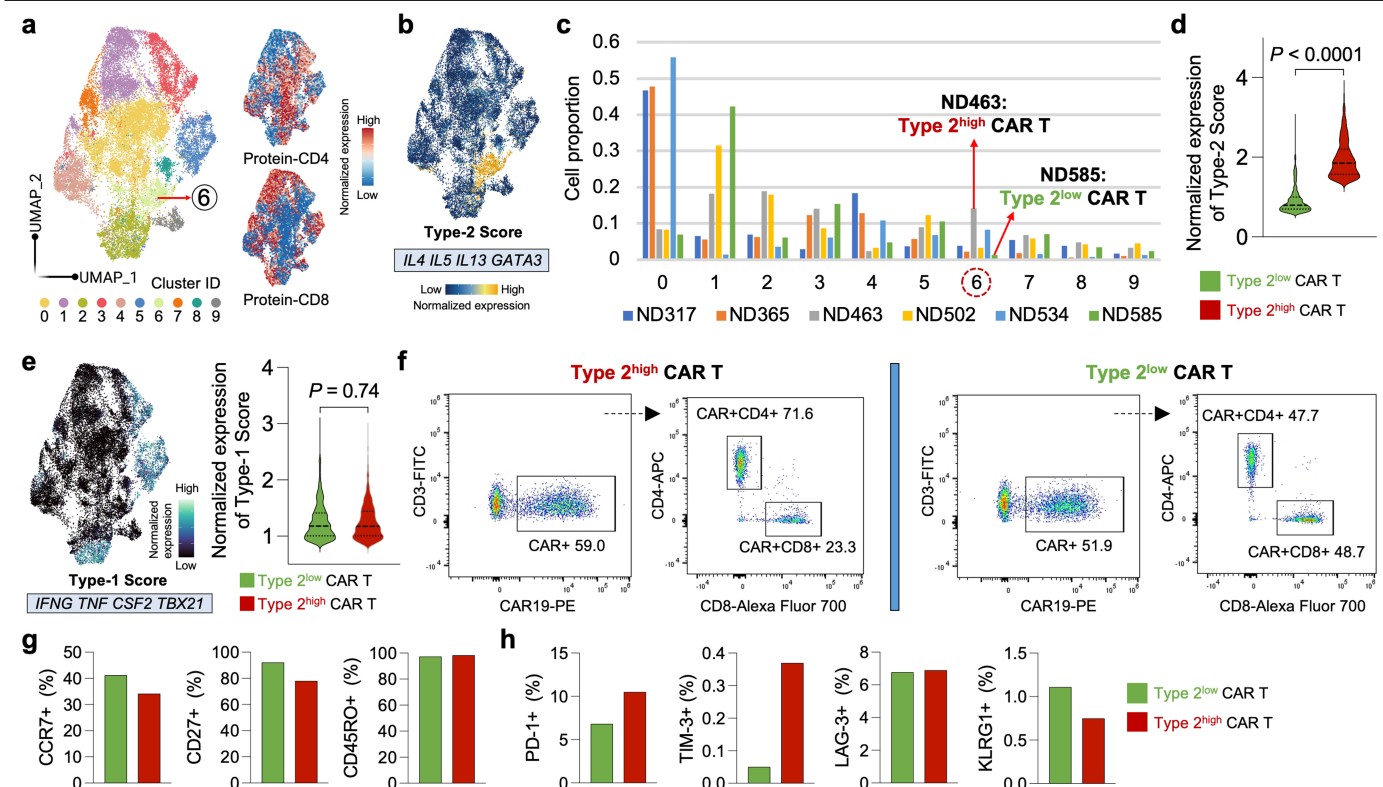

**Extended Data Fig. 12 | Selecting type 2^low or type 2^high CAR T for in vivo leukaemia model study. a**, UMAP clustering of CAR-specific stimulated CAR T cells from six healthy donors, along with the expression distribution of surface proteins ADT-CD4 and ADT-CD8. **b**, Expression distribution of Type-2 Score on the UMAP in (**a**), with the gene module found to be enriched in Cluster 6. **c**, Comparison of cell proportions in each identified cluster among different donors. ND463 is selected as type 2^high CAR T, and ND585 is selected as type 2^low CAR T based on the proportion in Cluster 6. **d**, Single-cell expression comparison of Type-2 Score between type 2^low and type 2^high CAR T cells. **e**, Expression distribution of Type-1 Score on the UMAP in (**a**), and single-cell expression comparison between type 2^low and type 2^high CAR T cells. **f**, Flow cytometry analysis of CAR, CD4, and CD8 expression in type 2^low and type 2^high CAR T cells before infusion into mice. **g**, **h**, Comparison of memory marker expression (**g**), and coinhibitory marker expression (**h**) in CAR+ pre-infusion cells between type 2^low and type 2^high CAR T cells. In **b** and **e**, genes defining each module are listed below. Significance levels were calculated with two-tailed Mann-Whitney test (**d**, **e**).

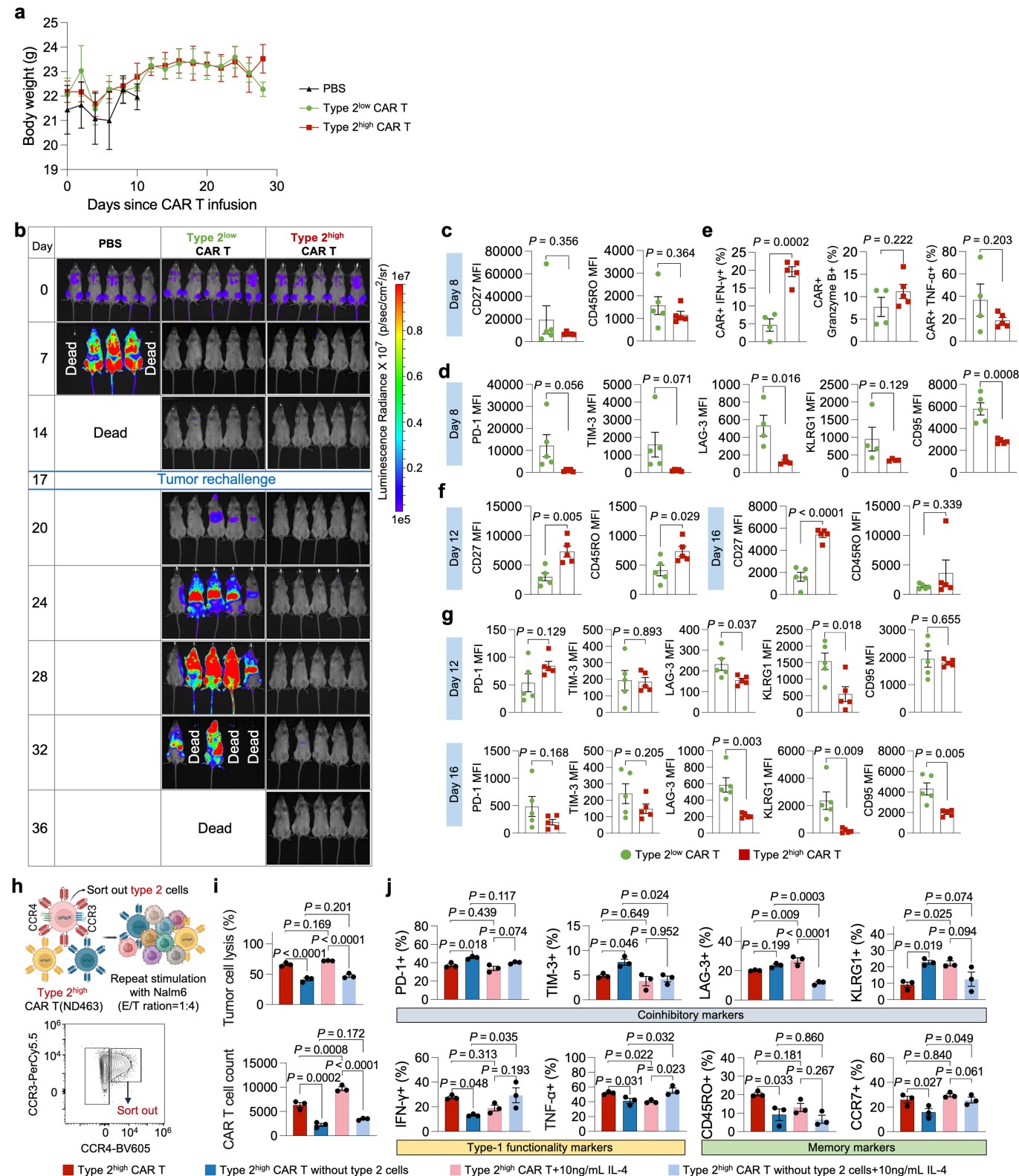

**Extended Data Fig. 13 | In vivo leukaemia model study using type 2^low or type 2^high CAR T cells. a**, Body weight of mice since CAR T cell infusion. **b**, Tumour burden measured by bioluminescence at indicated days since CAR T cell infusion. **c–e**, Flow cytometry analysis of memory markers (**c**), coinhibitory markers (**d**), and type 1 functionality markers (**e**) of peripheral CAR+ cells at day 8 post-CAR T cell infusion. **f**, **g**, Flow cytometry analysis of memory markers (**f**) and coinhibitory markers (**g**) of peripheral CAR+ cells at day 12 and day 16 post-CAR T cell infusion. **h**, Schematic of an in vitro repeat stimulation assay using type 2^high CAR T products, with type 2 cells sorted out based on the surface markers CCR3 and CCR4. The diagram was created using BioRender. **i**, Evaluation of tumour cell lysis efficacy and CAR T cell count at the endpoint under different conditions. **j**, Flow cytometry analysis of coinhibitory, type 1 functionality, and memory markers of CAR T cells at the endpoint. Scatter plot shows mean ± s.e.m. from n = 5 mice for each group (**a**, **c–g**), or n = 3 technical replicates for each condition (**i**, **j**). Significance levels were calculated with two-tailed unpaired Student's t-test (**c–g**), or one-way ANOVA with Tukey's multiple comparisons test (**i**, **j**).

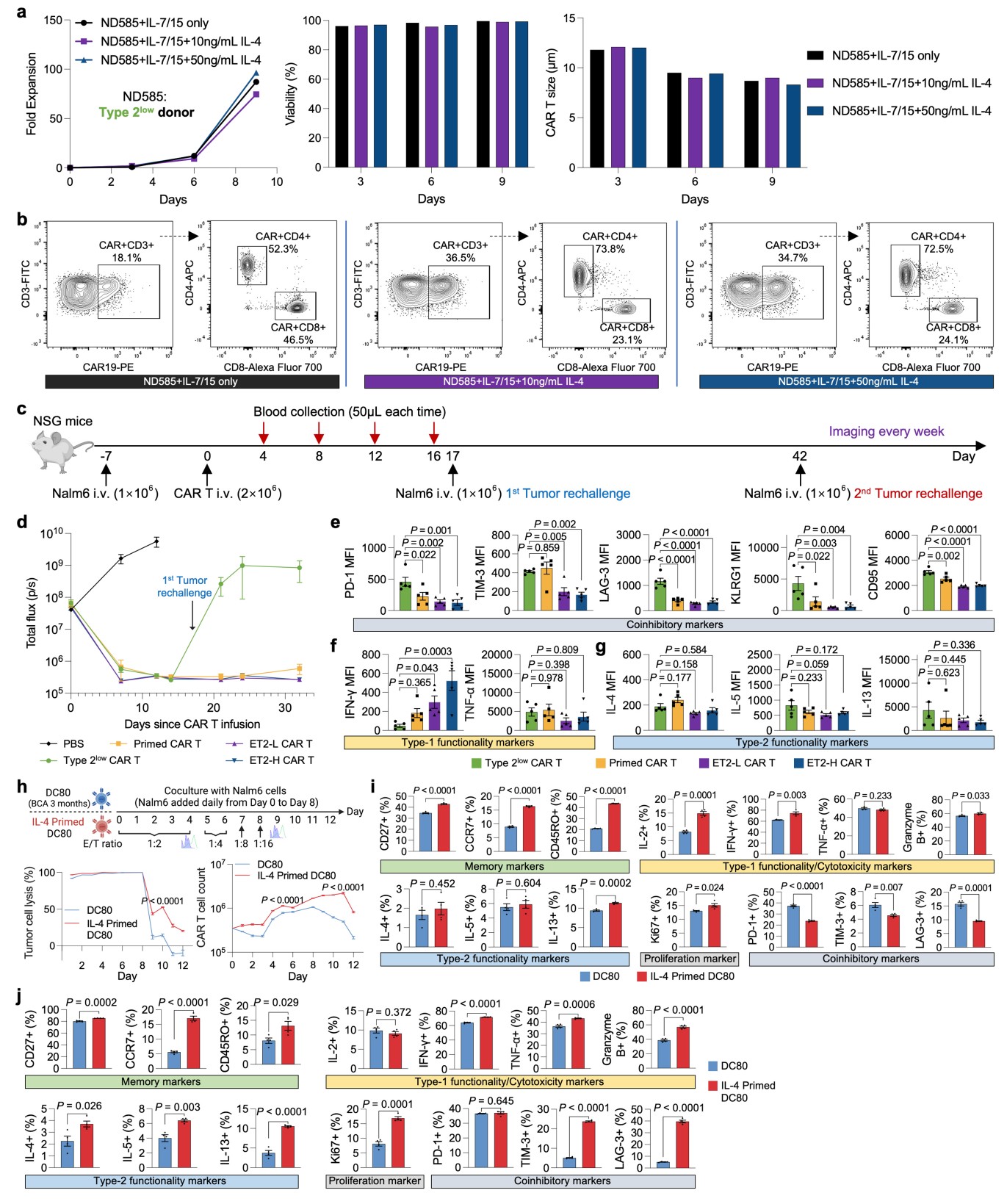

**Extended Data Fig. 14** | See next page for caption.

**Extended Data Fig. 14 | Functional evaluation of type 2 enhanced CAR T products. a**, Evaluation of the CAR T expansion, viability, and size of the newly designed "ET2-L/H CAR T" during the manufacturing process. **b**, Flow cytometry analysis of the CAR transduction efficiency and CD4/CD8 ratio of different conditions. **c**, Schematic of in vivo leukaemia model treated with type 2$^{low}$, Primed, ET2-L, or ET2-H CAR T cells. NSG mice were intravenously (i.v.) injected with $1 \times 10^6$ Nalm6 cells. Seven days later, mice were randomly assigned to five groups and were infused i.v. with $2 \times 10^6$ CAR T cells or PBS (control). The survival mice were rechallenged with $1 \times 10^6$ Nalm6 cells 17 days post CAR T infusion, followed by another $1 \times 10^6$ Nalm6 cells rechallenge 42 days post infusion. The diagram was created using BioRender. **d**, Tumour burden (total flux) quantified by photons/s in mice since CAR T treatment. **e**–**g**. Flow cytometry analysis of coinhibitory (**e**), type 1 functionality (**f**), and type 2 functionality markers (**g**) of peripheral CAR+ cells at day 8 post-CAR T cell infusion. **h**, Evaluation of tumour cell lysis efficacy and CAR T cell count in an in vitro repeat stimulation assay using patient-derived CAR T cells (DC80) with a BCA duration of 3 months, with or without 10 ng/mL IL-4 priming for 12 h. Nalm6 cells were introduced daily over a 9-day span at different effector-to-target (E/T) ratios as depicted in the schematic. Flow cytometry analysis was conducted on day 4 and day 9. Significance levels on specific days are denoted due to space constraints. The diagram was created using BioRender. **i**, **j**, Flow cytometry analysis of memory, type 1 functionality/cytotoxicity, type 2 functionality, proliferation, and coinhibitory markers of CAR T cells at day 4 (**i**) or day 9 (**j**) in the in vitro repeat stimulation assay. Scatter plot shows mean ± s.e.m. from $n = 5$ mice for each group (**d**–**g**), or $n = 4$ technical replicates for each condition (**h**–**j**). Significance levels were calculated with one-way ANOVA with Tukey's multiple comparisons test (**e**–**g**), or two-tailed unpaired Student's *t*-test (**h**–**j**).

| | |
|---|---|

# Reporting Summary

## Statistics

For all statistical analyses, confirm that the following items are present in the figure legend, table legend, main text, or Methods section.

| n/a | Confirmed | |
|---|---|---|
| ☐ | ☒ | The exact sample size (*n*) for each experimental group/condition, given as a discrete number and unit of measurement |
| ☐ | ☒ | A statement on whether measurements were taken from distinct samples or whether the same sample was measured repeatedly |
| ☐ | ☒ | The statistical test(s) used AND whether they are one- or two-sided *Only common tests should be described solely by name; describe more complex techniques in the Methods section.* |
| ☒ | ☐ | A description of all covariates tested |
| ☐ | ☒ | A description of any assumptions or corrections, such as tests of normality and adjustment for multiple comparisons |
| ☐ | ☒ | A full description of the statistical parameters including central tendency (e.g. means) or other basic estimates (e.g. regression coefficient) AND variation (e.g. standard deviation) or associated estimates of uncertainty (e.g. confidence intervals) |
| ☐ | ☒ | For null hypothesis testing, the test statistic (e.g. *F*, *t*, *r*) with confidence intervals, effect sizes, degrees of freedom and *P* value noted *Give P values as exact values whenever suitable.* |
| ☒ | ☐ | For Bayesian analysis, information on the choice of priors and Markov chain Monte Carlo settings |
| ☒ | ☐ | For hierarchical and complex designs, identification of the appropriate level for tests and full reporting of outcomes |
| ☐ | ☒ | Estimates of effect sizes (e.g. Cohen's *d*, Pearson's *r*), indicating how they were calculated |

*Our web collection on statistics for biologists contains articles on many of the points above.*

## Software and code

Policy information about availability of computer code

| | |
|---|---|
| Data collection | 1. The scRNA-seq libraries were prepared using the Chromium Single-Cell 3' Library and Gel Bead Kit v3.1 (10x Genomics, Cat# PN-1000268). 2. Single-cell ATAC+Gene co-profiling was performed using the Chromium Next GEM Single Cell Multiome ATAC + Gene Expression kit (10x Genomics, Cat# PN-1000283). 3. For flow cytometry analysis of CD19-3T3 stimulated CAR T cells from 82 ALL patients, data were collected using a Cytek Aurora flow cytometer. 4. The multiplexed secretomic assay was performed using a IsoLight (IsoPlexis) machine. 5. For flow cytometry analysis of CAR T cells isolated from mouse peripheral blood, data were collected using an Attune NxT Flow Cytometer with Attune NxT Software v.3 (Invitrogen). |
| Data analysis | 1. Single-cell transcriptome data processing and analysis: The sequencing data underwent alignment to the GRCh38 human reference genome, followed by barcode and unique molecular identifier counting, ultimately generating a digital gene expression matrix using Cell Ranger v6.1.2 (10x Genomics). The subsequent data analysis was conducted according to the Seurat v4 pipeline. The hashtag oligos expression was used to demultiplex cells back to their original sample-of-origin, while also identifying and excluding cross-sample doublets. Cells flagged as doublets (two barcodes detected) or lacking barcodes were omitted from the analysis. Only cells expressing a gene count ranging from 200 to 7,000 and exhibiting less than 10% mitochondrial gene content were retained for downstream analysis. 2. Single-cell ATAC+Gene co-profiling data processing and analysis: The Cell Ranger ARC v2.0.2 (10x Genomics) was utilized to perform sample demultiplexing, barcode processing, identification of open chromatin regions, and simultaneous counting of transcripts and peak accessibility in single cells from the sequenced data. The output per barcode matrices underwent joint RNA and ATAC analysis using Signac v1.12.0 and Seurat v4. Quality filtering criteria adhered to default settings. Specifically, cells were retained if they exhibited an ATAC peak count ranging from 1,000 to 100,000, a gene count ranging from 1,000 to 25,000, a nucleosome_signal below 2, and a TSS enrichment score exceeding 1. To enhance the accuracy of peak identification, we employed MACS2 v2.2.9.1 with the "CallPeaks" function. |

3. Flow cytometry analysis: Data acquired from the Cytek Aurora flow cytometer was analzyed using FlowJo v10.8.0. Data acquired from the Attune NxT Flow Cytometer was analzyed using FlowJo v10.6.1 (Tree Star).
4. The fluorescent signals in multiplexed secretomic assay were analyzed by the IsoSpeak v2.8.1.0 (IsoPlexis) software.
5. Ligand-receptor interaction analysis was performed using the R toolkit Connectome v1.0.0.
6. Ingenuity Pathway Analysis (IPA, QIAGEN) was used to reveal the underlying signaling pathways.
7. Statistical analyses were performed with Prism v10 (GraphPad) or R v4.3.1.

For manuscripts utilizing custom algorithms or software that are central to the research but not yet described in published literature, software must be made available to editors and reviewers. We strongly encourage code deposition in a community repository (e.g. GitHub). See the Nature Portfolio guidelines for submitting code & software for further information.

# Data

Policy information about availability of data

All manuscripts must include a data availability statement. This statement should provide the following information, where applicable:
- Accession codes, unique identifiers, or web links for publicly available datasets
- A description of any restrictions on data availability
- For clinical datasets or third party data, please ensure that the statement adheres to our policy

Raw and processed single-cell sequencing data for this study can be accessed in the NCBI Gene Expression Omnibus (GEO) database under the accession number GSE262072.

# Research involving human participants, their data, or biological material

Policy information about studies with human participants or human data. See also policy information about sex, gender (identity/presentation), and sexual orientation and race, ethnicity and racism.

| | |
|---|---|
| Reporting on sex and gender | Sex and/or gender was not considered in this study. |
| Reporting on race, ethnicity, or other socially relevant groupings | Race, ethnicity, or other socially relevant groupings were not considered in this study. |
| Population characteristics | The detailed information was provided in the supplementary table 1. We did not perform covariate analysis in this secondary correlation investigation. |
| Recruitment | Pre-infusion CAR T samples were acquired from patients with relapsed/refractory B-ALL who enrolled in a Phase I/IIA pilot clinical trial designed to assess the safety and feasibility of CTL019 T cell therapy (ClinicalTrials.gov number, NCT01626495), or a pilot study of the tocilizumab optimization timing for CART19 associated cytokine release syndrome (ClinicalTrials.gov number, NCT02906371). |
| Ethics oversight | The current study is a secondary investigation using patient samples collected from an existing clinical trial for which the University of Pennsylvania Institutional Board provided insight. |

Note that full information on the approval of the study protocol must also be provided in the manuscript.

# Field-specific reporting

Please select the one below that is the best fit for your research. If you are not sure, read the appropriate sections before making your selection.

☒ Life sciences  ☐ Behavioural & social sciences  ☐ Ecological, evolutionary & environmental sciences

For a reference copy of the document with all sections, see nature.com/documents/nr-reporting-summary-flat.pdf

# Life sciences study design

All studies must disclose on these points even when the disclosure is negative.

| | |
|---|---|
| Sample size | This study reports single-cell multiomics profiling of pre-infusion CAR T cells from 82 pediatric ALL patients and 6 healthy donors. No statistical methods were used to pre-determine sample size. Patient grouping was based on their clinical responses, with at least five patients included in each persistence group. To ensure findings are reproducible, functional in vitro study was performed with at least 3 technical replicates and leukemia mouse model study was performed with 5 animals per each group. This sample size was determined based on experience and well-established, previously published studies. |
| Data exclusions | There was no specific data exclusion criteria. |
| Replication | The characteristics of CAR T cells in this study were analyzed using various assays and independently conducted experiments at different research centers. These included single-cell RNA and CITE-seq multiomics at Yale University, single-cell ATAC+Gene co-profiling at Yale University, flow cytometry at the University of Pennsylvania, multiplexed secretomic assays at Yale University, and serum proteomic assays at the Children's Hospital of Philadelphia. Additionally, leukemia mouse model studies were performed at École Polytechnique Fédérale de |

Lausanne (EPFL). For each sample pool, two independent single-cell RNA and CITE-seq libraries were prepared and sequenced. CAR T cells derived from at least 3 different patients or healthy donors were used in each experiment to ensure robust conclusions.

| | |
|---|---|
| Randomization | Mice were randomized prior to CAR T treatment to ensure equivalent tumor burden among groups. To uncover the hallmarks of CAR T longevity, we correlated the single-cell multi-omics profiles with the duration of B-cell aplasia (BCA), a widely used pharmacodynamic measurement indicative of CAR T persistence. Consequently, we classified all patients into five persistence groups based on their clinically observed BCA duration, with no randomization performed. |
| Blinding | Blinding was not performed in this study design that involved deep characterization of pre-infusion CAR T cells from 82 ALL patients. All analyses were based on comparisons between two patient groups classified according to their clinically observed responses, making blinding not possible. |

# Reporting for specific materials, systems and methods

We require information from authors about some types of materials, experimental systems and methods used in many studies. Here, indicate whether each material, system or method listed is relevant to your study. If you are not sure if a list item applies to your research, read the appropriate section before selecting a response.

### Materials & experimental systems

| n/a | Involved in the study |
|---|---|
| ☐ | ☒ Antibodies |
| ☐ | ☒ Eukaryotic cell lines |
| ☒ | ☐ Palaeontology and archaeology |
| ☐ | ☒ Animals and other organisms |
| ☐ | ☒ Clinical data |
| ☒ | ☐ Dual use research of concern |
| ☒ | ☐ Plants |

### Methods

| n/a | Involved in the study |
|---|---|
| ☒ | ☐ ChIP-seq |
| ☐ | ☒ Flow cytometry |
| ☒ | ☐ MRI-based neuroimaging |

## Antibodies

| | |
|---|---|
| Antibodies used | All TotalSeq™-B anti-human antibodies were purchased from Biolegend: CD4 (RPA-T4, 300565), CD8 (SK1, 344757), CD45RA (HI100, 304161), CD45RO (UCHL1, 304257), CD62L (DREG-56, 304849), CD95 (DX2, 305653), CD127 (A019D5, 351354), CD28 (CD28.2, 302961), CD27 (O323, 302851), CCR7 (G043H7, 353249), HLA-DR (L243, 307661), CD69 (FN50, 310949), PD-1 (EH12.2H7, 329961), TIM-3 (F38-2E2, 345053), LAG-3 (11C3C65, 369337), CTLA-4 (BNI3, 369629), TIGIT (A15153G, 372727). <br><br> The following antibodies with indicated clones were used for flow cytometry analysis of CD19-3T3 stimulated CAR T cells from 82 ALL patients: PE-labeled monoclonal anti-FMC63 scFv (CAR19) (Y45, ACRO Biosystems, FM3-HPY53), CD3 (SK7, BD Biosciences, 564001), CD4 (OKT4, Biolegend, 317442), CD8a (RPA-T8, Biolegend, 301042), CD19 (HIB19, BD Biosciences, 561121), CD14 (M5E2, BD Biosciences, 561391), IL-3 (BVD3-1F9, Biolegend, 500606), IL-4 (MP4-25D2, Biolegend, 500834), IL-5 (TRFK5, Biolegend, 504306), IL-13 (JES10-5A2, Biolegend, 501916), IL-31 (1D10B31, Biolegend, 659608). <br><br> The following antibodies with indicated clones were purchased from Biolegend and used for flow cytometry analysis of CAR T cells isolated from mouse peripheral blood: CD95 (Fas) (DX2, 305624), IL-13 (JES10-5A2, 501916), CD27 (LG.3A10, 124249), CD45RO (UCHL1, 304238), TNF-α (MAb11, 502940), CD3 (OKT3, 317306), Granzyme B (GB11, 515403), CD223 (LAG-3) (11C3C65, 369312), CD366 (TIM-3) (F38-2E2, 345016), CD197 (CCR7) (G043H7, 353235), IL-4 (MP4-25D2, 500832), CD19 (HIB19, 302216), CD4 (OKT4, 317416), IL-5 (TRFK5, 504306), CD8 (SK1, 344724), CD279 (PD-1) (EH12.2H7, 329952), KLRG1 (MAFA) (2F1/KLRG1, 138426), IFN-γ (4S.B3, 502530), and Zombie Aqua™ Fixable Viability Kit (423102). Monoclonal Anti-FMC63 Antibody (Y45, FM3-HPY53) was purchased from ACRO Biosystems. |
| Validation | For TotalSeq™-B antibodies, per manufacturer's website (https://www.biolegend.com/en-us/quality/quality-control): "Bulk lots are tested by PCR and sequencing to confirm the oligonucleotide barcodes. They are also tested by flow cytometry to ensure the antibodies recognize the proper cell populations. Bottled lots are tested by PCR and sequencing to confirm the oligonucleotide barcodes". Detailed validation information for each antibody is available at the following sites: <br><br> 1. TotalSeq™-B0072 anti-human CD4 Antibody: https://www.biolegend.com/en-us/products/totalseq-b0072-anti-human-cd4-antibody-16820 <br> 2. TotalSeq™-B0046 anti-human CD8 Antibody: https://www.biolegend.com/en-us/products/totalseq-b0046-anti-human-cd8-antibody-18042 <br> 3. TotalSeq™-B0063 anti-human CD45RA Antibody: https://www.biolegend.com/en-us/products/totalseq-b0063-anti-human-cd45ra-antibody-16850 <br> 4. TotalSeq™-B0087 anti-human CD45RO Antibody: https://www.biolegend.com/en-us/products/totalseq-b0087-anti-human-cd45ro-antibody-16853 <br> 5. TotalSeq™-B0147 anti-human CD62L Antibody: https://www.biolegend.com/en-us/products/totalseq-b0147-anti-human-cd62l-antibody-16892 <br> 6. TotalSeq™-B0156 anti-human CD95 (Fas) Antibody: https://www.biolegend.com/en-us/products/totalseq-b0156-anti-human-cd95-fas-antibody-18636 <br> 7. TotalSeq™-B0390 anti-human CD127 (IL-7Rα) Antibody: https://www.biolegend.com/en-us/products/totalseq-b0390-anti-human-cd127-il-7ralpha-antibody-16859 <br> 8. TotalSeq™-B0386 anti-human CD28 Antibody: https://www.biolegend.com/en-us/products/totalseq-b0386-anti-human-cd28- |

antibody-16842

9. TotalSeq™-B0154 anti-human CD27 Antibody: https://www.biolegend.com/en-us/products/totalseq-b0154-anti-human-cd27-antibody-16839

10. TotalSeq™-B0148 anti-human CD197 (CCR7) Antibody: https://www.biolegend.com/en-us/products/totalseq-b0148-anti-human-cd197-ccr7-antibody-16857

11. TotalSeq™-B0159 anti-human HLA-DR Antibody: https://www.biolegend.com/en-us/products/totalseq-b0159-anti-human-hla-dr-antibody-16879

12. TotalSeq™-B0146 anti-human CD69 Antibody: https://www.biolegend.com/en-us/products/totalseq-b0146-anti-human-cd69-antibody-16873

13. TotalSeq™-B0088 anti-human CD279 (PD-1) Antibody: https://www.biolegend.com/en-us/products/totalseq-b0088-anti-human-cd279-pd-1-antibody-16863

14. TotalSeq™-B0169 anti-human CD366 (Tim-3) Antibody: https://www.biolegend.com/en-us/products/totalseq-b0169-anti-human-cd366-tim-3-antibody-19028

15. TotalSeq™-B0152 anti-human CD223 (LAG-3) Antibody: https://www.biolegend.com/en-us/products/totalseq-b0152-anti-human-cd223-lag-3-antibody-19187

16. TotalSeq™-B0151 anti-human CD152 (CTLA-4) Antibody: https://www.biolegend.com/en-us/products/totalseq-b0151-anti-human-cd152-ctla-4-antibody-18639

17. TotalSeq™-B0089 anti-human TIGIT (VSTM3) Antibody: https://www.biolegend.com/en-us/products/totalseq-b0089-anti-human-tigit-vstm3-antibody-16855

For antibodies used in flow cytometry, each antibody has been validated by the manufacturer for use to detect human species targets. Detailed validation information for each antibody is available at the following sites:

1. PE-labeled monoclonal anti-FMC63 scFv (CAR19) (Y45, ACRO Biosystems, FM3-HPY53): https://www.acrobiosystems.com/P3508-PE-Labeled-Monoclonal-Anti-FMC63-Antibody-Mouse-IgG1-%28Y45%29-%28Site-specific-conjugation%29-%28Preservative-free%29.html

2. CD3 (SK7, BD Biosciences, 564001): https://www.bdbiosciences.com/en-us/products/reagents/flow-cytometry-reagents/research-reagents/single-color-antibodies-ruo/buv395-mouse-anti-human-cd3.564001

3. CD4 (OKT4, Biolegend, 317442): https://www.biolegend.com/en-us/products/brilliant-violet-785-anti-human-cd4-antibody-7978

4. CD8a (RPA-T8, Biolegend, 301042): https://www.biolegend.com/en-us/products/brilliant-violet-650-anti-human-cd8a-antibody-7652

5. CD19 (HIB19, BD Biosciences, 561121): https://www.bdbiosciences.com/en-us/products/reagents/flow-cytometry-reagents/research-reagents/single-color-antibodies-ruo/v500-mouse-anti-human-cd19.561121

6. CD14 (M5E2, BD Biosciences, 561391): https://www.bdbiosciences.com/en-us/products/reagents/flow-cytometry-reagents/research-reagents/single-color-antibodies-ruo/v500-mouse-anti-human-cd14.561391

7. IL-3 (BVD3-1F9, Biolegend, 500606): https://www.biolegend.com/en-us/products/pe-anti-human-il-3-antibody-921

8. IL-4 (MP4-25D2, Biolegend, 500834): https://www.biolegend.com/en-us/products/apc-cyanine7-anti-human-il-4-antibody-13184

9. IL-5 (TRFK5, Biolegend, 504306): https://www.biolegend.com/en-us/products/apc-anti-mouse-human-il-5-antibody-989

10. IL-13 (JES10-5A2, Biolegend, 501916): https://www.biolegend.com/en-us/products/brilliant-violet-421-anti-human-il-13-antibody-13228

11. IL-31 (1D10B31, Biolegend, 659608): https://www.biolegend.com/en-us/products/alexa-fluor-488-anti-human-il-31-antibody-13170

12. CD95 (Fas) (DX2, 305624): https://www.biolegend.com/en-us/products/brilliant-violet-421-anti-human-cd95-fas-antibody-7252

13. IL-13 (JES10-5A2, 501916): https://www.biolegend.com/en-us/products/brilliant-violet-421-anti-human-il-13-antibody-13228

14. CD27 (LG.3A10, 124249): https://www.biolegend.com/en-us/products/brilliant-violet-605-anti-mouserathuman-cd27-antibody-19163

15.CD45RO (UCHL1, 304238): https://www.biolegend.com/en-us/products/brilliant-violet-605-anti-human-cd45ro-antibody-8569

16. TNF-α (MAb11, 502940): https://www.biolegend.com/en-us/products/brilliant-violet-711-anti-human-tnf-alpha-antibody-9034

17. CD3 (OKT3, 317306): https://www.biolegend.com/en-us/products/fitc-anti-human-cd3-antibody-3644

18. Granzyme B (GB11, 515403): https://www.biolegend.com/en-us/products/fitc-anti-human-mouse-granzyme-b-antibody-6066

19. CD223 (LAG-3) (11C3C65, 369312): https://www.biolegend.com/en-us/products/percp-cyanine5-5-anti-human-cd223-lag-3-antibody-13552

20. CD366 (TIM-3) (F38-2E2, 345016): https://www.biolegend.com/en-us/products/percp-cyanine5-5-anti-human-cd366-tim-3-antibody-8438

21. CD197 (CCR7) (G043H7, 353235): https://www.biolegend.com/en-us/products/pe-dazzle-594-anti-human-cd197-ccr7-antibody-9811

22. IL-4 (MP4-25D2, 500832): https://www.biolegend.com/en-us/products/pe-dazzle-594-anti-human-il-4-antibody-10216

23. CD19 (HIB19, 302216): https://www.biolegend.com/en-us/products/pe-cyanine7-anti-human-cd19-antibody-1911

24. CD4 (OKT4, 317416): https://www.biolegend.com/en-us/products/apc-anti-human-cd4-antibody-3657

25. IL-5 (TRFK5, 504306): https://www.biolegend.com/en-us/products/apc-anti-mouse-human-il-5-antibody-989

26. CD8 (SK1, 344724): https://www.biolegend.com/en-us/products/alexa-fluor-700-anti-human-cd8-antibody-9062

27. CD279 (PD-1) (EH12.2H7, 329952): https://www.biolegend.com/en-us/products/alexa-fluor-700-anti-human-cd279-pd-1-antibody-12365

28. KLRG1 (MAFA) (2F1/KLRG1, 138426): https://www.biolegend.com/en-us/products/apc-cyanine7-anti-mouse-human-klrg1-mafa-antibody-12486

29. IFN-γ (4S.B3, 502530): https://www.biolegend.com/en-us/products/apc-cyanine7-anti-human-ifn-gamma-antibody-6965

# Eukaryotic cell lines

Policy information about cell lines and Sex and Gender in Research

| | |
|---|---|
| Cell line source(s) | NIH/3T3, Nalm6, and HEK293T cell lines originally obtained from the American Type Culture Collection (ATCC). |
| Authentication | Cell lines were authenticated using STR profiling at least once every 3 years from receipt. |

| Mycoplasma contamination | All cell lines were tested negative for mycoplasma contamination. |
| --- | --- |
| Commonly misidentified lines (See ICLAC register) | No commonly misidentified cell lines were used in this study. |

# Animals and other research organisms

Policy information about studies involving animals; ARRIVE guidelines recommended for reporting animal research, and Sex and Gender in Research

| Laboratory animals | Six-week-old NOD/SCID/IL-2Rγnull (NSG) mice were procured from Charles River Laboratory (Lyon, France). All mice were housed in the Center of PhenoGenomics (CPG) animal facility at EPFL, kept in individually ventilated cages at 19-23°C with 45-65% humidity, and maintained on a 12-hour dark/light cycle. |
| --- | --- |
| Wild animals | The study did not involve wild animals. |
| Reporting on sex | Sex was not considered in study design. |
| Field-collected samples | The study did not involve samples collected in the field. |
| Ethics oversight | Experimental procedures in mouse studies were approved by the Swiss authorities (Canton of Vaud, animal protocol ID 3533) and performed in accordance with the guidelines from the CPG of EPFL. |

Note that full information on the approval of the study protocol must also be provided in the manuscript.

# Clinical data

Policy information about clinical studies

All manuscripts should comply with the ICMJE guidelines for publication of clinical research and a completed CONSORT checklist must be included with all submissions.

| Clinical trial registration | ClinicalTrials.gov number, NCT01626495, NCT02906371 |
| --- | --- |
| Study protocol | https://clinicaltrials.gov/ProvidedDocs/95/NCT01626495/Prot_SAP_000.pdf<br>https://clinicaltrials.gov/ct2/show/NCT02906371 |
| Data collection | This study is not a clinical study but uses biospecimens collected under the aforementioned clinical trials. Data collection occurred at the times indicated in the manuscript (from September 2012 to July 2022) at the University of Pennsylvania and Children's Hospital of Philadelphia. |
| Outcomes | This study is not a clinical study but uses biospecimens collected under the aforementioned clinical trials. Primary and secondary outcomes can be found in the above clinical study and protocol. |

# Flow Cytometry

## Plots

Confirm that:

☒ The axis labels state the marker and fluorochrome used (e.g. CD4-FITC).

☒ The axis scales are clearly visible. Include numbers along axes only for bottom left plot of group (a 'group' is an analysis of identical markers).

☒ All plots are contour plots with outliers or pseudocolor plots.

☒ A numerical value for number of cells or percentage (with statistics) is provided.

## Methodology

| Sample preparation | For flow cytometry analysis of CD19-3T3 stimulated CAR T cells from 82 ALL patients, the cocultured cells underwent a series of processing steps for immunostaining. Initially, they were washed twice in PBS and then stained for 20 minutes at room temperature (RT) with Live Dead Blue detection reagent (Thermo Fisher Scientific, Cat# L34962), diluted to 1:800 in PBS, to assess cell viability. Following this, cells were washed twice in FACS staining buffer and subsequently stained for surface molecules for 20 minutes at RT. To fix the stained cells, the Cytofix/CytoPerm Fixation/Permeabilization Kit (BD Biosciences, Cat# 554714) was utilized for 20 minutes at RT, while being protected from light. Subsequently, cells were washed twice with 1x Perm/Wash buffer and then stained for CAR19 and intracellular cytokines using antibodies in Perm/Wash buffer. This staining process was carried out for 20 minutes at RT in the dark. Then, cells underwent two additional washes with Perm/Wash buffer before being re-suspended in FACS staining buffer for subsequent analysis. Cell-surface antibodies were used at a 1:100 dilution during staining, and intracellular antibodies at a 1:50 dilution. Samples were run on the Cytek Aurora and analysis was performed using FlowJo v10.8.0.<br><br>For flow cytometry analysis of CAR T cells isolated from mouse peripheral blood, mouse blood (50μL) was collected from the |
| --- | --- |

tail at specified time points for peripheral CAR T cell analysis. The collected samples were resuspended in PBS with EDTA (2mM), and the red blood cells were removed using ACK lysis buffer (Gibco, Cat# A1049201). For surface marker staining, cells were incubated with an antibody panel at 4°C for 30 minutes, followed by live/dead staining using Zombie Aqua Fixable Dye (BioLegend, Cat# 423101). Cells were then washed and resuspended in PBS with 0.2% BSA for flow cytometry analysis. Intracellular cytokine staining was performed by first stimulating cells with a Cell Stimulation Cocktail (Invitrogen, Cat# 00-4970-03) for 5 hours at 37°C to induce cytokine production. Subsequently, cells were stained for surface markers and live/dead dye as previously described, then fixed and permeabilized using a Cytofix/Cytoperm Kit (BD Biosciences). Intracellular staining with the indicated antibody panel was conducted following the manufacturer's protocol. Cell-surface antibodies were used at a 1:100 dilution during staining, intracellular antibodies at a 1:50 dilution, and live/dead staining at a 1:1,000 dilution. Data were collected using an Attune NxT Flow Cytometer with Attune NxT Software v.3 (Invitrogen) and analyzed using FlowJo 10.6.1 (Tree Star).

Instrument

Samples were run on the Cytek Aurora or Attune NxT Flow Cytometer.

Software

Data were analyzed using FlowJo v10.8.0 or v10.6.1.

Cell population abundance

Post-sort purity was evaluated for all the samples and confirmed prior to data analysis.

Gating strategy

For flow cytometry analysis of CD19-3T3 stimulated CAR T cells from 82 ALL patients, the gating strategy is described in Extended Data Fig. 5a, with gates drawn using FMO controls: Live cells (Live Dead Blue-); Non-monocyte Non-B-cell (CD3 +CD14-CD19-); Lymphocytes (FSC-A vs. SSC-A); Singlets (FSC-A vs. FSC-H); CAR19+ cells (CAR19 vs. SSC-A); CD4+ or CD8+ CAR (CD4 vs. CD8).

For flow cytometry analysis of CAR T cells isolated from mouse peripheral blood, we used standard gating strategies: Lymphocytes (FSC-A vs. SSC-A); Singlets (FSC-A vs. FSC-H); Live cells (fixable Aqua dye signals); CAR19+ cells (CD3+CAR19+). Gate margins were determined by isotype controls and fluorescence-minus-one controls.

☒ Tick this box to confirm that a figure exemplifying the gating strategy is provided in the Supplementary Information.

