## [Peer Review File · Nature]

Manuscript Title: Single-cell CAR T atlas reveals type2 function in 8-year leukemia remission

Reviewer Comments & Author Rebuttals

Reviewer Reports on the Initial Version:

Referees' comments:

Referee #1 (Remarks to the Author):

The manuscript submitted by Bai et al. is a very impressive study, presenting high throughput omics data on 82 patients who received CD19 CAR T (CTL019), separated on the basis of extended B cell aplasia (BCA) and prolonged responses to therapy. Transcriptomic analyses revealed a high correlation between a specific cluster related to type 2 function and long-term event-free responses. This was further examined by measuring cytokine secretion using a multiplex secretomic assay, highlighting the power of a small cluster to influence other dominant clusters in the infusion product. This study supports the authors' previous findings, using similar methodology, from a smaller cohort (10.1126/sciadv.abj2820). While the authors should be commended for performing significant new studies to strengthen this conclusion, mechanistic studies are critical.

In the current manuscript, the question of whether CAR T manufacturing should include IL-4 stimulation and why IL-4 is important are not fully addressed. There are no mechanistic experiments of gain/loss-of-function and more specifically, an assessment of the relative importance of Th2 differentiation as compared to STAT6 signaling, GATA3 upregulation, and/or metabolic changes (potentially due to STAT6 signaling) are missing (see Comments 6 and 8 below).

Specific comments:

1. Abstract, page 1, line #7: The authors state that, "unexpectedly, we identified that elevated type-2..." Nonetheless, they previously reported that a Type 2 immune response is correlated with durable responses. Thus, the word is confusing, especially for readers who are familiar with their previous publication.

Main, p.2: In presenting their results on type-2 function in the introductory section, it would be helpful to cite their previous publication (reference 16). Presently, it is used to explain the experimental strategy in this study (results section). This is also the case in describing the data for Figure 2.

2. Figure 1g: The authors classified the 5 groups of CAR T patients based on the duration of B cell aplasia. The discovery cohort group (BCA-L) exhibits 82-106 mo BCA duration, while the validation cohort group (BCA-O) exhibits 48-72 mo BCA. As such, are these two cohorts comparable?

3. Figure 2 (page 5): The authors discuss the upregulation of clusters 0,1,7 in BCA-L. However, cluster 2 is significantly lower in this group and in HD, as compared to the relapsed patients group. It would be

important to discuss the potential significance of this cluster (especially as it also decreases following IL-4 stimulation (Fig 4c).

4. Figure 2 j, k: in silico analyses may not be sufficient to draw firm conclusions on the ligand/receptor interactions and the function of specific transcription factors. In the absence of experimental data, this can likely be moved to extended data and not emphasized.

5. Figure 3: As indicated above, in silico analyses do not demonstrate actual interactions (but rather predict molecular interactions) and as such, the title of the figure “Type-2 subpopulation regulates Tim-3+ dysfunctional effector CAR T cells through ligand-receptor interaction” is misleading. This is a hypothesis that could potentially be shown after the functional assays in Figures 4 and 5.

6. Figure 4: Cluster 2 (Type-1) is shown as being downregulated (Figure 4c), but the z score of the Type-1 pathway in Figure 4h is upregulated. Is this contradictory? Additionally, is it surprising that there is no z score for IL-4 signaling? Can you please add the label for cluster 7 in Figure 4b?

As regards the impact of IL-4 stimulation, it would be expected that clusters not expressing IL4R would not be affected. Is this the case? Conversely, is IL4Ra/gc (or potentially IL13Ra) expressed at high levels on cluster 2? It would be of interest to determine whether IL-4 signaling (i.e. STAT6 phosphorylation/ GATA3 upregulation) is differentially induced in cells from cluster1/2 as compared to other clusters. While potentially outside the scope of the present study, does IL4 stimulation of sorted cluster 2 or cluster 4 cells directly alter their phenotype (i.e. alleviate exhaustion)? Of note, this would not necessarily be predicted by the accompanying paper where Fc-IL-4 improves function without altering the exhausted phenotype of the CAR T cell. Please comment. Does an IL-4R blocking antibody inhibit this response?

7. Figure 5d-j: The authors compare the in vivo cytotoxicity of CAR T cells from 2 patients, with Type-2 Low and High activity, respectively. However, many parameters can potentially contribute to differences in their ability to control leukemia growth. It would be more appropriate to compare CAR T activity from identical donors (even healthy donors) that were treated with or without IL-4. This is complementary to the experiments presented in the accompanying manuscript.

8. Figure 6: Is the imbalance of “metabolic and functional programs” between BCA-L and RL+/- associated with a type2 response? The title of this figure refers to “metabolic and functional programs” but the data are based on transcriptional profiles. It would be helpful to validate at least some of these data by functional assays (i.e. flux, glycolysis, mTOR signaling (S6 phosphorylation), metabolomics).

9. Figures 4 and 6: Is the impact of IL-4 on CAR T cell activity dependent on Th2 polarization and/or a mechanistic pathway (i.e. STAT6 signaling and/or GATA3-driven reprogramming) altering metabolic/functional programs? Addressing this point would significantly strengthen this study.

Minor points:

1. Page 4, 3rd paragraph, line #10: The authors refer to a “proteomic analysis” but this appears to be a “surface-omic” analysis.
2. Extended figure 1i: The authors state that the expression of HLA-DR in clusters 6 and 8 and GRAZM in cluster 11 may be the result of tonic signaling. However, these authors have suggested that a CD19-CAR with a 4-1BB domain results in minimal tonic signaling.
3. Extended figure 2c: The legends for BCA-A2 and 1 are missing.
4. Figure 5b,c: The units of cytokines in Figure 5 b and c differ (Average type-2 cytokine values (pg/ml) in b and Normalized expression of type-2 cytokines in c). Please harmonize. It might also be helpful to harmonize the axes.
5. Figure 5d-j/Extended figure 11: “High” cells expanded but show low PD-1 expression on Day 8 (Figure 5i). Furthermore, the MFI of PD1 staining varies markedly between days 8, 12, and 16 (for “Low,” MFIs of $1e4$, $5e2$, and $5e3$, respectively, are presented). Is this correct? Other activation/ proliferation markers (CD25, CD71) can be assessed.
6. P. 10; The authors indicate that “The metabolic program of T cells shifts from oxidative phosphorylation (OXPHOS) to aerobic glycolysis upon activation to meet their energy demands⁴². Surprisingly, we observed that CD19-3T3 activated BCA-L cells maintained a significantly higher level of OXPHOS than cells from the other three groups (Fig. 6b and Extended Data Fig. 12d)”. However, T cell activation is known to result in a significant induction of OXPHOS; activation does not lead to a shift from OXPHOS to glycolysis but rather to a higher relative induction of the latter resulting in an increased ECAR/OCR ratio.
7. Figure 6/ Extended data 12: Were these experiments performed with non-stimulated or stimulated CAR T cells?
8. It would be helpful to add line numbers to be able to refer to specific questions.

Referee #2 (Remarks to the Author):

Bai and colleagues report an exciting large single-cell multi-omics dataset analyzing the pre-infusion CAR T cells from 82 pediatric B-ALL patients. These patients have been followed up for 5 to 10 years, providing the unique opportunity to understand the biological features associated long-term CAR T cell persistence (5-8 years). Remarkably, the analysis uncovered an elevated type-2 cytokine population in long term responder. Receptor-Ligand analysis suggested that this Type-2 cytokine CAR T cell population was maintaining a more functional type-1 function. Serum analysis from patient pre- and post-infusion confirmed higher levels of type-2 cytokines in long term responders. Using sc-RNA seq to select a healthy donor with high or low type 2 T cell population, they showed that high type 2 CAR T cells performed better in a Nalm6 xenograft rechallenge model. Finally, they show that CAR T cells from non-responders or early-relapse patients displayed a transcriptional program associated with dysfunctional mTOR signaling and lower metabolic activity. Together, the deep analysis of CAR T cells from a unique patient cohort uncovered novel insights in CAR T cell function and dysfunction, and open new ways to improve response to CAR T cell therapy. This study is likely to have a broad impact to the field.

Comments:

1. The authors do not mention the CAR expression. It has been reported by multiple teams that CAR expression levels can impact tonic signaling, T cell differentiation and function. Was it detected and was the level associated with certain clusters? Especially the one the authors suggested was associated with tonic signaling. If possible, such analysis should be included.
2. The Ligand-Receptor (L-R) analysis is really interesting. Does the L-R analysis take in consideration the percentage of cells expressing each receptor/ligand? From figure 2a, it looks like cluster 1 and 7 each represent approximatively 5% of the total CAR T cell population.
3. In Extended Data Fig 8: The difference in serum IL-4 seems mostly non significant in the serum, especially in the validation cohort. The authors should highlight this result and provide a hypothesis why it contradicts the pre-infusion CAR T cells data. Also, as mentioned by the authors, IL-13 in the contrary, is highly elevated in both BCA cohorts. How does IL-13 addition affect CAR T cells? In vitro experiments would help understanding the functional impacts of elevated IL-13 on CAR T cell function.
4. Fig5: In the in vivo experiment comparing type 2 high and low, while selecting two different donors by scRNA seq is interesting, the difference in CD4/CD8 ratio can potentially have a major impact in CAR T cells function, beyond the sole Type-2 cytokine effect. For example, It has been reported by Stan Riddell's group that CD4 CAR T cells alone are more functional than CD8 CAR T cells alone, and the ratio of each is critical for optimal antitumor response in a Nalm6 model. Also, in Extended Data Figure 10c, it is clear that ND463 has a much lower representation of cluster 1 and 4 T cell. We understand it might be impossible to find two donors that can only be differentiated by their Type 2 signature, but the current experiment is not conclusive. The authors should at least perform an experiment with similar CD4/CD8 ratio. Or better, they should pick a Type-2 low donor and pre-treat with IL-4 since they claim it can rescue the phenotype.
5. The relationship between mTor signaling and CD19- relapse is interesting. The definition of CD19- relapse depends on CD19 detection and it appears that many are actually CD19 very low and not CD19 -. Could the differences observed have a link to CAR T cell sensitivity to low antigen densities?
6. In Ext Fig2 D: incomplete legend
7. Page 7 : "these data revealed that type 2 marked cluster 7 cells mainly regulate a subpopulation". These are correlation studies which at best suggest an interaction but does not prove by it any mechanistic study. Thus, I think at this point the author should say "suggest that cluster 7 regulates".
8. Schematic representation in Fig 5a: the discovery cohort is a bit confusing. Does the circle in between number suggest that the harvest was performed at some point in between these days? If so it should be clarified in the legend.

Referee #3 (Remarks to the Author):

Single-cell multi-omics reveals type-2 function and metabolic fitness in maintaining CAR T cell longevity associated with 8-year leukemia remission.

In the paper entitled "Single-cell multi-omics reveals type-2 function and metabolic fitness in maintaining CAR T cell longevity associated with 8-year leukemia remission" Zhiliang and colleagues determined the molecular factors that contribute to the ultra-long persistence of CAR T cells by conducting a thorough single-cell RNA sequencing analysis of pre-infusion CAR T cells. Additionally, they correlated the transcriptional signatures of these cells with the corresponding 5-year or 8-year clinical outcomes. Remarkably, their findings indicated a notable elevation of type-2 signatures, rather than type-1 signatures, in the pre-infusion CAR T products, which was significantly associated with long-term persistence in patients. Through an examination of ligand-receptor interactions, they also discovered that type-2 cytokines contribute to long-term CAR T persistence through regulating the dysfunctional signatures of TIM3⁺ terminal effectors. Furthermore, they demonstrated that culturing BCA2 cells (CAR T cells with an average of ~4 months persistence) with the type-2 cytokine IL-4 resulted in a transcriptional and metabolic reprogramming, transforming them into BCA-L cells (CAR T cells with an average of ~101 months persistence). Notably, in a leukemic mouse model, the researchers showed that high type-2 CAR T cell products exhibited superior expansion and demonstrated increased antitumor activity upon with leukemia rechallenge.

Overall, the authors present an original and provocative study, employing rigorous analysis of pre-infusion CAR T cells and tracking their clinical responses over 8 years. Importantly, they establish, for the first time, a correlation between the type-2 response and ultra-long CAR T persistence with relatively large clinical datasets, which holds significant implications for understanding the molecular mechanisms of remarkable persistence of CAR T cells over an extended period. These findings will undoubtedly captivate a wide audience. Nonetheless, it is crucial to conduct further analysis of the scRNA-Seq and functional data to substantiate the bold conclusions drawn in the paper. In general, the authors have a clear hypothesis that they are proposing, but their experiments and analysis do not fully support the hypothesis and the experimental approach could be refined to more directly test their ideas to develop more solid conclusions. It is known that CD4 T cell help is critical to avoid Cd8 T cell exhaustion and to promote memory and so if the authors are suggesting that Th2-like cells provide the best help, then they should develop experiments to more directly examine this. With more definite tests of the hypothesis and better delineation of the subsets and their functional roles, the authors may have quite a novel clinical finding that would be of high impact to the field.

Suggested improvements:

1. Cite-seq analysis has been conducted to aid in the identification of CD4 and CD8 subsets within the

scRNA-Seq analysis of pre-infusion CAR T cells. However, the subsequent analysis of clusters and cells reveals a mixture of CD4 and CD8 T cells. It is advisable to segregate the CD4 and CD8 cells and perform separate downstream analyses to strengthen their point. For instance, refer to Figure 2a and Figure 2c for illustrations. For instance, they need to better define the Th2-like cells...are they canonical CD4+ Th2 cells? Are there CD8 T cells in the Th2 cluster and if so, do they exhibit canonical Th2 properties. Talk about other surface markers typically and used to define Th2 cells (chemokine receptors, CRTH2, Transcription factors, etc).

2. The authors need to better delineate the function of the Th2-like cells from the other clusters. Although the researchers observed an elevation of type-2 signatures in persistent CAR T products, it comprised less than 5% of the total population. On the other hand, the memory subsets, which account for over 30% of the total population, showed elevated levels in the persistent CAR T cells too. These memory subsets have previously been implicated in the long-term persistence of CAR T cells. Therefore, the authors must ascertain the relative contribution of the elevated presence of memory cells vs. the elevation of type 2 response in determining the ultra-long survival of CAR T cells by purifying, separating and mixing different populations of cells together.

3. In figure 4, the authors demonstrated that the addition of IL4 could transcriptionally revert BCA2 cells to BCA-L states. However, it is crucial to present functional data indicating that the inclusion of IL4 can also improve the functionality of BCA2 cells. This is especially important because the experiment was conducted in the regular culture system, and the exhaustion marker may simply indicate the activation state of the cells.

4. In figure 5, the mouse leukemia model utilizing type-2 high cells failed to adequately address their point, mainly due to the limited representation of the type-2 high cells among the total transferred cells. To overcome this limitation, the author should consider (a) sorting out the type-2 cells and conducting the experiments anew. A potential surrogate for the sorting strategy could involve utilizing CXCR3-CCR6- CCR4+ as a distinguishing marker (<https://pubmed.ncbi.nlm.nih.gov/20042588/>). (b) blocking type-2 signaling through neutralizing antibodies. This is essential to draw more solid conclusions since it is experimentally tractable.

5. In figure 6, it is crucial to comprehend the significant role of elevated type-2 responses in contributing to the metabolic and functional reprogramming associated with non-response, CD19-positive and CD19-negative relapse. However, the comparison of metabolic differences between the responders vs. non-response dose not add noteworthy significance or impact, and it seems to diverge from the overall theme of the paper.

Additional comments:

1. Exhaustion signature was calculated only based on 4 genes; more exhaustion genes should be included since these are also "activation markers" and exhaustion is truly defined by loss of cytokine production like IL-2, TNFa and IFNg.

2. It would be beneficial to include IL10 and IL21 production in single cell secretome assay.
3. Figure 2b and Figure 2d showed expression score of type-1 and type-2 signatures. Additionally, it would be beneficial to include the expression scores of individual type-1 and type-2 genes.

Referee #4 (Remarks to the Author):

In this manuscript, Bai et al. conducted single-cell analysis on approximately 700k pre-infusion CAR T cells obtained from 82 ALL patients and 6 healthy donors. They investigated the relationship between the type-2 function of CAR T cells and the outcome of CAR T cell therapy based on long-term clinical follow-up data. In addition, the authors performed in vitro experiments and utilized a leukemic mouse model to validate their findings, with a particular focus on the importance of type-2 functionality in persistent CAR T cells. This study presents a valuable dataset for assessing CAR T cell persistency. The manuscript is well-written and easily understandable. The robustness of the results and the overall integrity of the manuscript maybe further enhanced by addressing the following issues.

Major comments:

1. In lines 122-123, the authors state that “basal state CAR T cells are partitioned into 8 sub-clusters...”, but the corresponding figures do not show or indicate these eight clusters. It should be noted that Extended Data Fig. 1 identifies a total of 17 clusters.
2. In lines 215-216 and lines 218-240, while the authors demonstrated elevated levels of IL-4 and IL-25 in Cluster 7 as a type-2 functionality cluster, there is a lack of analysis or validation regarding STAT6 and GATA3 activation. For Fig 3e-3g, expression profile of all Type-2 related genes need to be analyzed and presented. Similarly in Fig. 2k, the key molecules in this cascade model should be validated in patient samples before drawing conclusions about the upstream regulation of type-2 function in the present study.
3. The detailed approaches applied for ligand-receptor analysis and communication pathway analysis in Figs. 3a and 3b should be stated in Methods section.
4. The definition of “Type-2 L-R interactions” also needs to be clarified. Are any L-R interactions that includes type-2-related proteins can only be categorized into Type-2 L-R interactions?
5. In Fig. 3b, what does the term "top-ranking level of interactions" specifically refer to? The reviewer suggests that the methodology used to determine Cluster 7 to 2 communication should be clarified.
6. In lines 277-288, The observation of a lower Exhaustion Score and Exhaustion ADT Score in Cluster 2 of BCA-L cells does not provide sufficient evidence to conclude that type-2 marked Cluster 7 cells primarily regulate Tim-3+ dysfunctional CAR T cells.

7. In Fig. 3c, which Cluster is the top-ranking cluster that communicate with Cluster 2, is it also Cluster 7?

8. In Figs. 4h and 4i, is it possible to calculate the fold change that quantifies the improvement in functional fitness of CAR T cells upon the addition of IL-4? This would help in assessing the extent of the observed change.

9. In leukemic mouse model experiments, how are the type-2 -low and -high CAR T cells produced prior to infusion? How to define the type-2 functionality in the future potential manufactural application?

10. Please provide the GEO dataset accession number.

Minor points:

1. In Extended Data Fig.2c, labels for groups are not all indicated.

2. In line 330-331, the expression “Baseline measurements (-2 to 0 days before CTL019 infusion)” is confusing. Maybe more appropriate with “0 to 2 days before CTL019 infusion”.

Author Rebuttals to Initial Comments:

Response to Reviewers

We are very appreciative of all the constructive feedback from the reviewers. We believe the revised manuscript has properly addressed all the major concerns and it has been significantly improved. All the major changes are highlighted in yellow in the manuscript attached. Thank you for your thoughtful guidance and constructive input to this study!

Our major revisions, which include the incorporation of new data into the revised manuscript, are summarized below in the order of the new figures:

1. A comprehensive global correlation analysis unveiling associations among distinct transcriptomic and surface proteomic states within each category of CAR T cells (Fig. 1d).
2. Sub-clustering analysis of CD4+ or CD8+ activated CAR T cells, separated by CITE-seq surface proteins (Extended Data Fig. 4a–e).
3. Single-cell ATAC and gene co-profiling of CD19-specific activated CAR T cells, comparing 3 BCA-L patients with another 3 patients maintaining BCA duration of less than 3 months (Fig. 2g–j and Extended Data Fig. 6).
4. Functional in vitro repeat stimulation assay using CAR T cells with knockdown of STAT6 or GATA3 (Fig. 2k).
5. Single-cell transcriptomic analysis of donor CAR T cell population response to other type-2 cytokines including IL-5 and IL-13 (Extended Data Fig. 8f–k).
6. Single-cell ATAC analysis evaluating chromatin accessibility alterations following the IL-4 treatment of CAR T cells that only maintain BCA duration of less than 3 months (Fig. 3m–o and Extended Data Fig. 10).
7. A significantly expanded patient sera proteomic dataset post CAR T infusion was provided to further enhance the clinical relevance of this study (Fig. 4, Extended Data Fig. 11, and Supplementary Fig. 1–3).
8. Functional in vitro assay using Type-2 High CAR T products, with type-2 cells sorted out based on the surface markers CCR3 and CCR4 (Fig. 5h–j).
9. To investigate whether enhancing type-2 functionality could elevate the performance of the Type-2 Low CAR T, two strategies were proposed and evaluated using in vivo leukemia mouse model: priming CAR T products with type-2 cytokine before infusion and incorporating type-2 cytokine into the manufacturing process (Fig. 6a–d and Extended Data Fig. 13a–i).

10. Functional in vitro assay employing the IL-4 priming strategy on patient CAR T cells mediating BCA duration of 3 months (Fig. 6e–i and Extended Data Fig. 13j–o).

11. We concur with the feedback from Reviewer #3 regarding the divergence of the original Fig. 6, which depicted metabolic programs associated with clinical response, from the overall theme of this study. Consequently, we have relocated it to the supplementary figures and briefly referenced these findings in the "Discussion" section.

12. Accordingly, the title has been modified to “Single-cell multi-omics reveals type-2 function in CAR T cells associated with 8-year leukemia remission”.

Point-by-point replies to Referees' comments:

Referee #1 (Remarks to the Author):

The manuscript submitted by Bai et al. is a very impressive study, presenting high throughput omics data on 82 patients who received CD19 CAR T (CTL019), separated on the basis of extended B cell aplasia (BCA) and prolonged responses to therapy. Transcriptomic analyses revealed a high correlation between a specific cluster related to type 2 function and long-term event-free responses. This was further examined by measuring cytokine secretion using a multiplex secretomic assay, highlighting the power of a small cluster to influence other dominant clusters in the infusion product. This study supports the authors' previous findings, using similar methodology, from a smaller cohort (10.1126/sciadv.abj2820). While the authors should be commended for performing significant new studies to strengthen this conclusion, mechanistic studies are critical.

Response: Thank you very much for your enthusiastic affirmation of this study and for recognizing our previous work!

In the current manuscript, the question of whether CAR T manufacturing should include IL-4 stimulation and why IL-4 is important are not fully addressed. There are no mechanistic experiments of gain/loss-of-function and more specifically, an assessment of the relative importance of Th2 differentiation as compared to STAT6 signaling, GATA3 upregulation, and/or metabolic changes (potentially due to STAT6 signaling) are missing (see Comments 6 and 8 below).

Response: Thank you for providing such valuable suggestions! In response to the inquiry regarding the inclusion of IL-4 stimulation in CAR T manufacturing, we selected the most Type-2 Low CAR T donor sample (ND585) from a comparison with five other donors. We then introduced IL-4 into the manufacturing process, starting from apheresis T cells of this specific donor (Fig. R1a). Two different IL-4 doses were assessed during the new manufacturing of ND585 T cells, namely 10ng/mL or 50ng/mL, generating Enhanced Type-2 CAR T referred to as "ET2-L CAR T" or "ET2-H CAR T". Subsequently, we compared the in vivo tumor killing efficiency of this modified CAR T product with the original CAR T product, which was manufactured using the standard cytokine recipe of IL-7 and IL-15. Additionally, we explored an alternative approach of priming CAR T products with IL-4 (Primed CAR T) before infusion, providing insights for practical clinical applications where re-manufacturing may not be feasible. While all treatment groups successfully cleared the tumor burden within one week, only Primed CAR T and ET2-L/H CAR T demonstrated the ability to completely reject the same amount of tumor cell rechallenge (Fig. R1b), highlighting the therapeutic potency of enhanced type-2 functionality. Flow cytometry analysis at various time points post-CAR T infusion revealed significant expansion of CAR+ cells in the peripheral blood for the new products (Fig. R1c), coupled with significantly reduced expression of coinhibitory markers and increased IFN- γ production at the peak on day 8 (Extended Data Fig. 13g, h). This could be linked to their superior capacity contributing to a significantly prolonged survival (Fig. R1d). To rigorously assess the resilience of the two strategies, we initiated a second tumor rechallenge on day 42, simulating a relatively late-stage relapse. Following this third tumor cell injection, both the ET2-L and ET2-H CAR T showcased robust tumor control capabilities, outperforming the group treated with Primed CAR T (Fig. R1b, d). All these results, along with detailed descriptions, are presented in our revised Fig. 6a-d and Extended Data Fig. 13a-i.

Fig. R1 | Revitalizing Type-2 Low CAR T via enhanced type-2 functionality boost.

a, Schematic representation depicting two strategies employed to enhance the type-2 functionality of Type-2 Low CAR T derived from the donor ND585. **b**, Tumor burden measured by bioluminescence at indicated days since CAR T cell infusion (n=5 mice for each group). **c**, CAR T cell expansion in the peripheral blood of Nalm6-bearing mice measured at different timepoints after infusion (n=5 mice for each group). **d**, Kaplan–Meyer curves showing mouse survival.

We provide multiple datasets to answer why IL-4 is important and unravel the underlying regulatory mechanisms, including:

(1) To reveal the upstream regulatory factors linked to CAR T persistence, we conducted a single-cell ATAC and gene co-profiling of CD19-specific stimulated CAR T cells (Fig. R2a), comparing 3 BCA-L patients with another 3 patients from the BCA2 or BCA1 groups, who maintained BCA duration of less than 3 months. Pseudo-bulk chromatin accessibility signals in the genomic regions of *GATA3* were notably higher in each of the BCA-L patients (Fig. R2b). Interestingly, comparable signals were observed for another key type-2 regulator, *STAT6*, across the six patients (Fig. R2c). Within the realm of significantly enhanced motif binding activities observed in BCA-L CAR T cells compared to the BCA2/1 cells, *GATA3* emerged as the foremost differentially enriched site, along with several other GATA family members (Fig. R2d). Delving into the per-cell motif activity profile for each patient, *TBX21* displayed consistent levels across all patients, whereas *GATA3* showcased notably enriched activities in single cells from the three BCA-L patients. It's noteworthy that *STAT6* did not emerge as a differential motif in this analysis. These results are presented in our revised Fig. 2g-j and Extended Data Fig. 6.

Fig. R2 | Single-cell ATAC+Gene co-profiling of CD19-specific activated patient CAR T cells.

a, Integrated ATAC-Gene UMAP clustering of CAR-specific stimulated CAR T cells derived from 6 patients. Unsupervised clustering reveals 10 distinct clusters, with an enrichment of type-2 cells in Cluster A3. Comparison of cell proportions in Cluster A3 is presented between patient groups. **b**, **c**, Pseudo-bulk chromatin accessibility tracks in the genomic region of *GATA3* (**b**) and *STAT6* (**c**), depicted separately for each patient. The enhancer elements predicted by ENCODE within this region are highlighted in a light-yellow shade. **d**, Volcano plot showing

differential motif activities in BCA-L vs. BCA2/1 CAR T cells, with GATA3 identified as the most enhanced motif in the BCA-L group. Dot plot showing expression profile of type-2 motif MA0037.3 (GATA3) and type-1 motif MA0690.1 (TBX21) across each patient.

(2) To gauge the relative importance of STAT6 and GATA3 in orchestrating type-2 functionality, an in vitro repeat stimulation assay was conducted using CAR T cells with knockdown of STAT6 or GATA3 (Fig. R3a). While both knockdowns significantly attenuated tumor killing efficacy and CAR T cells count, the GATA3 knockdown demonstrated a notably more pronounced compromise in the CAR T population number at the assay's endpoint (Fig. R3b), suggesting its central role in sustaining functional type-2 immunity in long-term persistent CAR T cells. These results are presented in our revised Fig. 2k.

Fig. R3 | Loss-of-function experiment to evaluate the relative importance of GATA3 and STAT6.

a, b, Evaluation of tumor cell lysis efficacy and CAR T cell count in an in vitro repeat stimulation assay using CAR T cells with knockdown of STAT6 or GATA3. Nalm6 cells were added daily over a 4-day period at an effector-to-target (E/T) ratio of 1:4.

(3) To further investigate the regulatory role of type-2 cytokine on CAR T cells, we assessed the chromatin accessibility alterations following the inclusion of 10 ng/mL IL-4 in stimulating CAR T cells from three BCA2/1 patients, who maintained BCA duration of less than 3 months. Differential accessibility peak activity analysis unveiled a functional reprogramming of these less persistent cells after IL-4 treatment, highlighting significant upregulation of type-1 marker *IFNG*, type-2 marker *IL13*, chemokines *XCL1/2*, cytotoxic markers *GNLY* and *NKG7*, as well as activation/proliferation markers *VIM* and *CD70* (Fig. R4a). Notably, the accessibility of *CSF2*, a pivotal immune modulator with profound effects on T cell functional activities, demonstrated the highest level of upregulation, while the activity of *IL32*, marking activation-induced cell death, exhibited substantial downregulation in the IL-4 supplemented condition (Fig. R4b). These trends were consistently supported by signal tracks observed in each patient. The improvements in overall functional fitness, reinforced by the enhanced motif binding of both type-1 (STAT1) and type-2 (GATA3) master regulators (Fig. R4c), could be attributed to

the effective regulation of dysfunctional population by type-2 cytokines. These results are presented in our revised Fig. 3m-o and Extended Data Fig. 10.

Fig. R4 | Single-cell ATAC profiling of short-term patient CAR T cells in response to type-2 cytokine treatment.

a, Volcano plot showing differential accessible peak activities in BCA2/1 CAR T cells with and without the addition of 10ng/mL IL-4. **b**, Pseudo-bulk chromatin accessibility tracks in the genomic region of *CSF2* and *IL32*, depicted separately for each patient with and without the addition of IL-4. The enhancer elements predicted by ENCODE within this region are highlighted in a light-grey shade. **c**, Dot plot showing expression profile of type-2 motif MA0037.3 (GATA3) and type-1 motif MA0690.1 (STAT1) across each patient with and without the addition of IL-4.

Specific comments:

1. Abstract, page 1, line #7: The authors state that, “unexpectedly, we identified that elevated type-2...” Nonetheless, they previously reported that a Type 2 immune response is correlated with durable responses. Thus, the word is confusing, especially for readers who are familiar with their previous publication.

Main, p.2: In presenting their results on type-2 function in the introductory section, it would be helpful to cite their previous publication (reference 16). Presently, it is used to explain the experimental strategy in this study (results section). This is also the case in describing the data for Figure 2.

Response: We genuinely appreciate your feedback! Based on your suggestion, we have revised the abstract, introduction, and results sections accordingly.

2. Figure 1g: The authors classified the 5 groups of CAR T patients based on the duration of B cell aplasia. The discovery cohort group (BCA-L) exhibits 82-106 mo BCA duration, while the validation cohort group (BCA-O) exhibits 48-72 mo BCA. As such, are these two cohorts comparable?

Response: We acknowledge the distinction between these two cohorts. Initially, we conducted our analysis with all patients combined, identifying 16 patients (5 BCA-L and 11 BCA-O) exhibiting long-term response, defined as more than 5 years of relapse-free remission. The major findings drawn from this combined analysis are totally consistent with the current version of the manuscript. However, following extensive discussions with senior authors overseeing these clinical trials, we decided to maintain the original trial identity and classify the patients into two cohorts. These patients were part of CHOP's initial two clinical trials and received the same CAR T cell product. While the CAR T structure design remained consistent, certain factors such as recruitment criteria, manufacturing protocol, and patient treatment processes may have evolved over the 4-year interval between the launch of the Discovery Cohort (NCT01626495) trial in 2012 and the Validation Cohort (NCT02906371) trial launched in 2016. Although there may be some confounding variables between the cohorts, our focus remains on identifying universal biological mechanisms underlying long-term CAR T persistence. Both BCA-L and BCA-O patients have exhibited robust responses lasting more than five years, providing invaluable opportunities to explore and address this critical question. Moreover, all data presented in Figures 2 and 3, as well as the corresponding extended figures, include comprehensive analyses from both cohorts, with major findings consistently supported across both groups. While we understand that using terms like "Cohort 1" and "Cohort 2" could help avoid potential confusion, we have opted to retain the current group names to ensure readers' familiarity with the terminology. We hope you can appreciate that clinically these groups remain comparable, and importantly, the conclusions drawn are consistently robust irrespective of how the cohorts are grouped.

3. Figure 2 (page 5): The authors discuss the upregulation of clusters 0,1,7 in BCA-L. However, cluster 2 is significantly lower in this group and in HD, as compared to the relapsed patients group. It would be important to discuss the potential significance of this cluster (especially as it also decreases following IL-4 stimulation (Fig 4c).

Response: We respectfully want to clarify that this cluster has undergone in-depth analysis in the original **Figure 3** (now also depicted in the revised Figure 3), rather than Figure 2. Through ligand-receptor interaction analysis, we identified that, among vigorous interactions, type-2 cells mainly regulate **Cluster 2** cells exhibiting overactivation of cytotoxicity, high expression of *HAVCR2* and TIM-3, impaired immune function, and attenuated proliferation. The proportion of this dysfunctional cluster was significantly reduced in BCA-L and BCA-O patients compared to less durable groups. Following IL-4 inclusion into the CAR T activation process, this signature decreases.

4. Figure 2 j, k: in silico analyses may not be sufficient to draw firm conclusions on the ligand/receptor interactions and the function of specific transcription factors. In the absence of experimental data, this can likely be moved to extended data and not emphasized.

Response: Thank you for your constructive suggestion! We have relocated this part to Extended Data Figure 4f and adjusted the manuscript accordingly by removing the detailed descriptions.

5. Figure 3: As indicated above, in silico analyses do not demonstrate actual interactions (but rather predict molecular interactions) and as such, the title of the figure “Type-2 subpopulation regulates Tim-3+ dysfunctional effector CAR T cells through ligand-receptor interaction” is misleading. This is a hypothesis that could potentially be shown after the functional assays in Figures 4 and 5.

Fig. R5 | Type-2 CAR T cells regulate dysfunctional subpopulation.

a, Identification of ligand-receptor (L-R) interactions originating from type-2 enriched Cluster 7 cells. **b**, Identification of L-R interactions targeting Cluster 2 cells. **c**, Dot plot showing expression profile of type-2 receptor genes across all clusters, with notable high expression observed in Cluster 2. **d**, Differentially expressed genes

(DEGs) specific to Cluster 2 in comparison to all other clusters, along with the expression distribution of the Cytotoxic Score. **e**, Corresponding signaling pathways regulated by the DEGs identified in Cluster 2. **f**, Heatmap showing the average expression levels of coinhibitory-related genes or ADT proteins across all single cells within each identified cluster. **g**, Expression distribution of Proliferation Score on the Discovery Cohort UMAP, with an evident absence of expression observed in Cluster 2. **h**, Comparison of cell proportion in Cluster 2 between persistence groups. **i**, Experimental design schematic for assessing the impact of IL-4 supplementation on the functional profile of CAR T cells derived from short-term BCA2 patients. **j**, UMAP clustering of CAR T cells from 6 patients in BCA2 group, with and without 10ng/mL IL-4 added during in vitro CAR-specific activation. **k**, Comparison of cell proportions in specific clusters between conditions. **l**, Comparison of regulatory pathways between CAR T cells from the long-term BCA-L patients and six BCA2 patients under original and 10ng/mL IL-4 conditions.

Response: Thank you for your valuable insights! We completely agree that in silico ligand-receptor (L-R) analyses may not provide definitive conclusions. However, we have chosen to retain the key L-R regulation results to initiate Figure 3, while eliminating some redundant information. The primary rationale behind this decision is that identifying how type-2 cells regulate dysfunctional cells is a pivotal aspect that motivated us to conduct functional assays, as depicted in the original Figure 4. In these assays, we supplemented IL-4 during CAR T cell stimulation, and indeed observed a reduction in CAR T cell dysfunctions, further supporting the relevance of the L-R interactions identified. Furthermore, our analysis is grounded on a substantial single-cell dataset, and the findings are consistently upheld in both the Discovery Cohort and Validation Cohort.

In the revised version, we have strengthened this section by consolidating the functional results from the original Figure 4 into a more concise Figure 3 (Fig. R5 shown above). Additionally, we have incorporated the single-cell ATAC-seq results, as shown in Fig. R4. In response to your suggestion, we have adjusted the section title to "Type-2 CAR T cells regulate dysfunctional subpopulation" to ensure clarity and accuracy. We believe that this revised step-by-step logical flow: (1) utilizing L-R analysis to identify the regulatory roles of type-2 cells; (2) conducting functional evaluations on both donor and patient samples by incorporating type-2 cytokines into the activation process; and (3) performing ATAC assays to unveil the epigenetic alterations upon IL-4 inclusion, provides robust support for our new section title. All these results are presented in our revised Fig. 3 and Extended Data Fig. 7-10.

6. Figure 4: Cluster 2 (Type-1) is shown as being downregulated (Figure 4c), but the z score of the Type-1 pathway in Figure 4h is upregulated. Is this contradictory? Additionally, is it surprising that there is no z score for IL-4 signaling? Can you please add the label for cluster 7 in Figure 4b?

Response: In the original Figure 4c (now Extended Data Fig. 8b-e), which was based on healthy donor samples, the proportion of type-1 enriched Cluster 2 indeed showed a negligible tiny reduction upon IL-

4 inclusion. However, the scenario changes when considering the 6 patient samples (Fig. R5j, k). Upon inclusion of 10ng/mL IL-4 (optimized using donor samples), the proportion of type-1 clusters (now Cluster 0/2/5/9) increased in 5 out of the 6 patients, although no significant changes were observed. This could be attributed to intrinsic differences between healthy donor and patient CAR T cells.

The elevation of the type-1 pathway, alongside other functional pathways observed in the patient dataset (original Figure 4h, now Extended Data Fig. 9e/Fig. R6 shown here) represents an overall shift in the CAR T cell population's profile, rather than within any specific cluster. Upon the exclusion of dysfunctional Cluster 4 in Fig. R5j, this analysis resulted in a noteworthy decrease in the upregulation level of functional pathways, alongside a complete loss of the downregulation of apoptosis (Fig. R6). This suggests that the advantageous enhancement in functional profile may be attributed to the effective regulation of the dysfunctional population by type-2 cytokines, a notion further supported by our ATAC data presented in Fig. 3m-o (also in Fig. R4 shown above).

Fig. R6 | Comparison of signaling pathways in BCA2 patient CAR T cells supplemented with 10ng/mL IL-4 relative to the original condition, with or without the exclusion of dysfunctional cytotoxic Cluster 4 cells. A statistical comparison of the activation Z Score for functional signaling pathways, including those regulating metabolism, immune function, and proliferation, was conducted.

We inadvertently omitted the z score for IL-4 signaling in the original figure, which has now been included in the revised version (Fig. R6 shown above).

Regarding adding the label for "cluster 7" in Figure 4b (now Fig.3j/Fig. R5j shown above), we presume you are inquiring about the potential impact of adding IL-4 on the type-2 cluster, which corresponds to

Cluster 8 in this particular analysis (Fig. R5j). Based on the findings depicted in Fig. R5k, no significant changes were observed in the proportion of the type-2 cluster, mirroring the outcomes seen in the analysis of donor samples (Extended Data Fig. 8b, c, e). This outcome aligns with our expectations. We anticipated that the incorporation of a moderate level of IL-4 would exert a regulatory effect on dysfunctional cells, thereby enhancing the functional fitness of the entire population, rather than polarizing cells into type-2 CAR T cells.

As regards the impact of IL-4 stimulation, it would be expected that clusters not expressing IL4R would not be affected. Is this the case? Conversely, is IL4Ra/gc (or potentially IL13Ra) expressed at high levels on cluster 2? It would be of interest to determine whether IL-4 signaling (i.e. STAT6 phosphorylation/GATA3 upregulation) is differentially induced in cells from cluster1/2 as compared to other clusters. While potentially outside the scope of the present study, does IL4 stimulation of sorted cluster 2 or cluster 4 cells directly alter their phenotype (i.e. alleviate exhaustion)? Of note, this would not necessarily be predicted by the accompanying paper where Fc-IL-4 improves function without altering the exhausted phenotype of the CAR T cell. Please comment. Does an IL-4R blocking antibody inhibit this response?

Response: Thank you for inspiring us to explore the IL4 receptor expression patterns! Indeed, in the ligand-receptor analysis depicted in the above Fig. R5, we observed that type-2 cells primarily regulate dysfunctional cells in Cluster 2, and we also noted the highest expression of corresponding receptor genes in this cluster (Fig. R5c). We then supplied 10ng/mL IL-4 during the stimulation of CAR T cells from 6 patients who only maintain a BCA duration of ~3 months (Fig. R5i). Unsupervised clustering analysis revealed a significant enhancement in proliferation (Cluster 1) and mitigation of dysfunctional cytotoxicity (Cluster 4) in patient CAR T cells after adding IL-4 (Fig. R5j, k), with notably enriched expression of corresponding receptor genes observed in these two clusters (Extended Data Fig. 9d/ Fig. R7 here). In contrast, clusters without high-level expression of these receptor genes

were minimally affected, such as clusters 0, 2, 5, and 9 enriched with type-1 signatures, as well as cluster 8 enriched with type-2 signatures.

Fig. R7 | Dot plot showing expression profile of type-2 receptor genes across all clusters identified in Fig. R5j. The size of circle represents proportion of single cells expressing the gene, and the color shade indicates normalized expression.

As you mentioned, determining whether IL-4 signaling (i.e., STAT6 phosphorylation/ GATA3 upregulation) is differentially induced in cells from clusters 1/2 (we assume you may be referring to clusters 1/4 in Fig. R5j) as compared to other clusters would be very interesting. While in silico analysis offers a promising avenue for predicting upstream regulator activities from scRNA-seq expression data, its reliability may not be robust enough for drawing definitive conclusions. Ideally, isolating these specific clusters and conducting functional assays to evaluate whether IL-4 stimulation directly alters their phenotype and whether IL-4R blocking antibody inhibits this response would provide more conclusive evidence. However, these results were obtained from analyses of 6 patient CAR T cells. We sincerely appreciate your understanding that we lack additional samples from the exact subset of six patients necessary to expand this study further. It's reassuring to know that you also recognize that such an endeavor might extend beyond the scope of our current research.

7. Figure 5d-j: The authors compare the in vivo cytotoxicity of CAR T cells from 2 patients, with Type-2 Low and High activity, respectively. However, many parameters can potentially contribute to differences in their ability to control leukemia growth. It would be more appropriate to compare CAR T activity from identical donors (even healthy donors) that were treated with or without IL-4. This is complementary to the experiments presented in the accompanying manuscript.

Response: We kindly clarify that the Type-2 Low and High CAR T cells were selected from CAR T samples obtained from **healthy donors**, not patients. Your suggestion to utilize identical donors treated with or without IL-4 for comparing CAR T activity is greatly valued! We have conducted a comprehensive in vivo study, evaluating two strategies aimed at enhancing the type-2 functionality of the Type-2 Low CAR T cells from this donor, with data shown in the above Fig. R1. Furthermore, we implemented the priming strategy on **patient** CAR T cells mediating BCA duration of only 3 months and performed in vitro repeat stimulation assays (Fig. R8a). Similarly to the in vivo results, the primed CAR T group exhibited a significant increase in CAR T cell count over the first 4 days and displayed significantly enhanced memory signatures, reduced coinhibitory markers, along with augmented expression of IL-2, IFN- γ , granzyme B, Ki67, and IL-13 on day 4 (Fig. R8b-e). By gradually reducing the E/T ratio to 1:16 on day 8, we observed a noteworthy discrepancy in tumor killing between the primed CAR T cells and the original sample (Fig. R8a). These results are presented in our revised Fig. 6 and Extended Data Fig. 13.

Fig. R8 | Enhancing functional fitness of short-term patient CAR T via type-2 cytokine priming.

a, Evaluation of tumor cell lysis efficacy and CAR T cell count in an in vitro repeat stimulation assay using patient-derived CAR T cells (DC80) with a BCA duration of 3 months, with or without 10ng/mL IL-4 priming for 12h. Nalm6 cells were introduced daily over a 9-day span at different effector-to-target (E/T) ratios as depicted in the schematic. **b–e**, Flow cytometry analysis of memory (**b**), coinhibitory (**c**), type-1 functionality/cytotoxicity (**d**), and proliferation (**e**) markers of CAR T cells at day 4.

8. Figure 6: Is the imbalance of “metabolic and functional programs” between BCA-L and RL+/- associated with a type2 response? The title of this figure refers to “metabolic and functional programs” but the data are based on transcriptional profiles. It would be helpful to validate at least some of these data by functional assays (i.e. flux, glycolysis, mTOR signaling (S6 phosphorylation), metabolomics).

Response: We sincerely value these insightful suggestions! Nevertheless, we concur with **Reviewer #3's** observation that the original Figure 6, depicting metabolic and functional programs of CAR T cells, diverges from the overarching theme of this study. Consequently, we have relocated it to the supplementary figures and briefly referenced these findings in the "Discussion" section of the revised version. Instead, to further enhance the clinical relevance of our study and evaluate the role of type-2 functionality over an extended period following CAR T infusion, we have expanded the proteomic profiling of patient serum samples. This extension encompasses up to two months post-infusion in the Discovery Cohort and up to 28 days post-infusion in the Validation Cohort (Fig. R9a). Once more, these datasets validate the presence of elevated circulating type-2 cytokines in the post-infusion sera of long-term BCA-L and BCA-O patients (Fig. R9b-e). Notably, the expanded data sets afford us the opportunity to conduct unsupervised analysis, delving into whether the serum proteomic profile might reflect intrinsic variations in immune response among different patients post-CAR T treatment. Impressively, the results indeed indicate the feasibility of this notion (Fig. R9f-h). These results are presented in our revised Fig. 4, Extended Data Fig. 11, and Supplementary Fig. 1-3. We believe the exclusion of the metabolic dataset and the expansion of patient serum profiling now more effectively highlight the central focus of this study: the role of type-2 functionality in maintaining long-term CAR T persistence.

Fig. R9 | Elevated levels of type-2 cytokines detected in post-infusion sera from long-term responders.

a, Schematic of the serial proteomic profiling to measure serum proteins in 33 patients from the Discovery Cohort and 8 patients from the Validation Cohort. Timepoints are relative to the day of first infusion of CTL019 cells (Day 0). **b, c**, Longitudinal levels of type-2 cytokines in patients from the Discovery Cohort (**b**) and the Validation Cohort (**c**). **d, e**, Comparison of the average type-2 cytokine levels at multiple time points between persistent groups in the

Discovery Cohort (d) and the Validation Cohort (e). f, Principal Component Analysis (PCA) of 345 measurements of serum samples from 33 patients in the Discovery Cohort, based on the detected values of the 30-Plex cytokines and grouped by BCA response. g, Unsupervised clustering analysis of proteomic measurements, grouped by Cluster ID or BCA response, visualized using UMAP. Cluster 2 exhibits enrichment in BCA-L patients. h, Heatmap showing the differentially expressed proteins defining each cluster. Cluster 2 exhibits a high expression level of type-2 cytokines (IL-13 and IL-4).

9. Figures 4 and 6: Is the impact of IL-4 on CAR T cell activity dependent on Th2 polarization and/or a mechanistic pathway (i.e. STAT6 signaling and/or GATA3-driven reprogramming) altering metabolic/functional programs? Addressing this point would significantly strengthen this study.

Response: Thank you for this highly valuable suggestion! We addressed this point by conducting functional assays with data shown in the above Fig. R4 and Fig. R8, excluding exploration of metabolic factors as this part has been removed from the main “Results” section entirely. Surprisingly, it appears that the influence of IL-4 on CAR T cell activity does not primarily revolve around Type-2 functionality. Both chromatin accessibility profiles (Fig. R4) and in vitro functional assays (Fig.R8) revealed that augmenting type-2 functionality led to a significant enhancement of CAR T cells derived from patients who had only achieved short-term responses, improving various aspects such as type-1 functionality, cytotoxicity, proliferation, and survival. This amelioration even induced a transcriptional reversion to states akin to BCA-L CAR T (Fig.R5I), potentially attributable to the regulation of dysfunctional cells. These findings suggest that the presence of type-2 CAR T cells maintains a homeostatic state of the entire population by suppressing hyperactive cytotoxicity in the early stage and mitigating irreversible exhaustion, thereby enhancing their functional persistence.

Minor points:

1. Page 4, 3rd paragraph, line #10: The authors refer to a “proteomic analysis” but this appears to be a “surface-omic” analysis.

Response: Thank you for bringing this to our attention! We have now updated the term to “surface proteomics” to ensure accuracy. Additionally, we have thoroughly reviewed the entire manuscript to eliminate any instances of such inaccurate term usage.

2. Extended figure 1i: The authors state that the expression of HLA-DR in clusters 6 and 8 and GRAZM in cluster 11 may be the result of tonic signaling. However, these authors have suggested that a CD19-CAR with a 4-1BB domain results in minimal tonic signaling.

Response: We believe you may be referring to our previous paper published in the Journal for ImmunoTherapy of Cancer (<https://jitc.bmj.com/content/9/5/e002328>). However, we would like to

respectfully clarify that this is a misunderstanding. In our study, we reported that basal CAR T cells exhibit tonic signaling governed by a combination of early activation, exhaustion, and cytotoxic activities. As part of this signature, we observed a significant increase in the expression of *HLA-DRB1* in CAR+ cells compared to CAR- cells, a finding that aligns with the results obtained from

this considerably larger dataset. For your convenience, we have included the corresponding figure panel here.

3. Extended figure 2c: The legends for BCA-A2 and 1 are missing.

Response: We have corrected this and reviewed the entire manuscript to prevent any similar oversights.

4. Figure 5b,c: The units of cytokines in Figure 5 b and c differ (Average type-2 cytokine values (pg/ml) in b and Normalized expression of type-2 cytokines in c). Please harmonize. It might also be helpful to harmonize the axes.

Response: While it's indeed a valuable suggestion, we are unable to harmonize the two datasets as they were obtained using different assays. In our Discovery Cohort, the measurements were conducted using a FlexMAP 3D instrument (Luminex), and data acquisition and analysis were performed using xPONENT software (Luminex). For the Validation Cohort, the Olink Explore panel (Olink Proteomics) was utilized to measure serum proteins, with all protein data reported automatically in normalized expression values on a log2 scale. Despite the use of two different methods across the two cohorts, the consistently elevated levels of type-2 cytokines circulating in long-term responders suggest that this represents genuine biology unaffected by technological variations.

5. Figure 5d-j/Extended figure 11: "High" cells expanded but show low PD-1 expression on Day 8 (Figure 5i). Furthermore, the MFI of PD1 staining varies markedly between days 8, 12, and 16 (for "Low," MFIs of 1e4, 5e2, and 5e3, respectively, are presented). Is this correct? Other activation/ proliferation markers (CD25, CD71) can be assessed.

Response: After double-checking all the raw data, we confirm the accuracy of the presented data. It's important to note that comparing fluorescence intensity across experiments can be challenging due to various experimental conditions affecting intensity, such as antibody dilution and laser fluctuations [1]. Furthermore, we must disclose that we encountered an issue with our 637nm laser between day 8 and day 12 of the experiment, prompting its replacement and subsequent compensation, further complicating inter-day comparisons. Despite this, we presented the data transparently, believing that the observed variation does not undermine the intrinsic differences between the two groups (Type-2 Low and Type-2 High). While incorporating activation/proliferation markers is indeed a valid suggestion, we hope for your understanding that repeating these in vivo experiments presents considerable challenges.

Reference [1]: Herzenberg, L., Tung, J., Moore, W. et al. Interpreting flow cytometry data: a guide for the perplexed. *Nat Immunol* 7, 681–685 (2006). <https://doi.org/10.1038/ni0706-681>.

6. P. 10; The authors indicate that “The metabolic program of T cells shifts from oxidative phosphorylation (OXPHOS) to aerobic glycolysis upon activation to meet their energy demands⁴². Surprisingly, we observed that CD19-3T3 activated BCA-L cells maintained a significantly higher level of OXPHOS than cells from the other three groups (Fig. 6b and Extended Data Fig. 12d)”. However, T cell activation is known to result in a significant induction of OXPHOS; activation does not lead to a shift from OXPHOS to glycolysis but rather to a higher relative induction of the latter resulting in an increased ECAR/OCR ratio.

Response: Although these contents have been completely removed, we greatly appreciate this suggestion that has directed us to consider further exploration in the near future!

7. Figure 6/ Extended data 12: Were these experiments performed with non-stimulated or stimulated CAR T cells?

Response: The data initially presented in Figure 6 and Extended Figure 12 were obtained from stimulated CAR T cells. Although analysis of non-stimulated basal CAR T cells would provide valuable insights, we have chosen not to include discussion of these findings in the main "Results" section of the revised version.

8. It would be helpful to add line numbers to be able to refer to specific questions.

Response: The line numbers were added in this revised version.

Referee #2 (Remarks to the Author):

Bai and colleagues report an exciting large single-cell multi-omics dataset analyzing the pre-infusion CAR T cells from 82 pediatric B-ALL patients. These patients have been followed up for 5 to 10 years, providing the unique opportunity to understand the biological features associated long-term CAR T cell persistence (5-8 years). Remarkably, the analysis uncovered an elevated type-2 cytokine population in long term responder. Receptor-Ligand analysis suggested that this Type-2 cytokine CAR T cell population was maintaining a more functional type-1 function. Serum analysis from patient pre- and post-infusion confirmed higher levels of type-2 cytokines in long term responders. Using sc-RNA seq to select a healthy donor with high or low type 2 T cell population, they showed that high type 2 CAR T cells performed better in a Nalm6 xenograft rechallenge model. Finally, they show that CAR T cells from non-responders or early-relapse patients displayed a transcriptional program associated with dysfunctional mTOR signaling and lower metabolic activity. Together, the deep analysis of CAR T cells from a unique patient cohort uncovered novel insights in CAR T cell function and dysfunction, and open new ways to improve response to CAR T cell therapy. This study is likely to have a broad impact to the field.

Response: Thank you sincerely for your encouraging affirmation of this study!

Comments:

1. The authors do not mention the CAR expression. It has been reported by multiple teams that CAR expression levels can impact tonic signaling, T cell differentiation and function. Was it detected and was the level associated with certain clusters? Especially the one the authors suggested was associated with tonic signaling. If possible, such analysis should be included.

Response: In this study, we conducted multi-omics profiling of both basal unstimulated CAR T cells and activated CAR+ cells. To ensure consistent capture of authentic CAR-specific immune synapses, we engineered a murine NIH3T3 cell line expressing human CD19 (CD19-3T3) as antigen presenting cells (APCs), exclusively activating CAR T products through their CAR. Following coculture with CD19-3T3 for 12 hours, **CAR+ cells** that were successfully transduced were magnetically **sorted using a monoclonal antibody specific for CAR19 before single cell isolation and sequencing** (Fig. R1, also Extended Data Fig. 1a). Consequently, all the data presented in our main figures are based on purely sorted CAR+ cells.

Fig. R1 | Experimental pipeline of this study.

While the basal unstimulated condition yielded results reflecting a mixed expression of both transduced CAR+ cells and untransduced CAR- cells, we fully acknowledge the potential informativeness of a separate analysis, particularly regarding CAR tonic signaling. However, it's important to note that such an analysis is not the primary focus of this study. Consequently, all findings from unstimulated CAR T were just briefly presented in our Extended Data Fig. 2. We also sincerely invite you to explore our previous paper published in the Journal for ImmunoTherapy of Cancer (<https://jitc.bmj.com/content/9/5/e002328>), wherein we elucidated a detailed molecular signature of tonic signaling by comparing CAR+ to CAR- cells at the basal state.

2. The Ligand-Receptor (L-R) analysis is really interesting. Does the L-R analysis take in consideration the percentage of cells expressing each receptor/ligand? From figure 2a, it looks like cluster 1 and 7 each represent approximately 5% of the total CAR T cell population.

Fig. R2 | Ligand-receptor (L-R) analysis across identified CAR T populations.

a, Identification of L-R interactions originating from type-2 enriched Cluster 7 cells, predominantly interacting with Cluster 2 cells through L-R pairs involving type-2 cytokines. The thickness of edges is proportional to correlation weights, and edge color corresponds to the Cluster ID. **b**, Identification of L-R interactions targeting Cluster 2 cells, revealing that cells from the majority of other clusters predominantly regulate these cells through L-R pairs involving type-2 cytokines. **c**, Dot plot showing expression profile of type-2 receptor genes across all clusters, with notable high expression observed in Cluster 2. The size of circle represents proportion of single cells expressing the gene, and the color shade indicates normalized expression level.

Response: We performed ligand-receptor (L-R) analysis and found that, within the intricate network of communication pathways emanating from Type-2 cells enriched in Cluster 7 (**3.14%** of the entire population), the highest level of interactions explicitly converging on Cluster 2 cells (**Fig. R2a**), accounting for **13.9%** of the whole population. We further found that, in most identified clusters, L-R interactions toward Cluster 2 cells prevalently engage type-2 cytokines (**Fig. R2b**). In **Fig. R2c**, a dot plot was employed to illustrate both the normalized expression level and **expression percentage** of receptor genes sensing type-2 ligands, highlighting the highest levels in Cluster 2 compared to other clusters. Notably, these percentages, such as ~95% for *IL2RG* and ~45% for *IL4R*, represent the proportion **within Cluster 2** rather than the entire CAR T cell population. The analysis process underlying this algorithm indeed takes into consideration both ligand and receptor expression percentages, with increased percentages yielding higher statistical confidence. In these figures, we selected most top-ranked interaction pairs for visualization. In the revised manuscript, we have added a new section in the Methods to provide a detailed explanation of this L-R analysis methodology.

3. In Extended Data Fig 8: The difference in serum IL-4 seems mostly non significant in the serum, especially in the validation cohort. The authors should highlight this result and provide a hypothesis why it contradicts the pre-infusion CAR T cells data. Also, as mentioned by the authors, IL-13 in the contrary, is highly elevated in both BCA cohorts. How does IL-13 addition affect CAR T cells? In vitro experiments would help understanding the functional impacts of elevated IL-13 on CAR T cell function.

Response: Thank you for your insightful suggestion! As you pointed out, our analysis revealed that IL-13, rather than IL-4, significantly influenced the differences observed in the averaged expression level of type-2 cytokines. In this revised version, to further enhance the clinical relevance of our study and evaluate the role of type-2 functionality over an extended period following CAR T infusion, we have expanded the proteomic profiling of patient serum samples. This extension encompasses up to two months post-infusion in the Discovery Cohort and up to 28 days post-infusion in the Validation Cohort (**Fig. R3a**). In both cohorts, it was consistently observed that IL-13 exhibited higher overall expression

levels compared to IL-4 or IL-5 across all patients and longitudinal time points, as indicated by the y-axis values.

Fig. R3 | Elevated levels of type-2 cytokines detected in post-infusion sera from long-term responders.

a, Schematic of the serial proteomic profiling to measure serum proteins in 33 patients from the Discovery Cohort and 8 patients from the Validation Cohort. Timepoints are relative to the day of first infusion of CTL019 cells (Day 0). **b, c**, Longitudinal levels of type-2 cytokines in patients from the Discovery Cohort (**b**) and the Validation Cohort (**c**).

However, this observation is not in conflict with our scRNA-seq data from pre-infusion CAR T cells. Previously, we presented a module expression analysis that integrated four type-2 genes (*IL4*, *IL5*, *IL13*, and *GATA3*) (Fig. R4a, b). In the revised version, we included the expression distribution of each gene individually and observed a notably higher expression of *IL13* compared to the others (Fig. R4c).

Fig. R4 | Expression distribution of type-2 gene module and defining markers.

a, UMAP clustering of CAR-specific stimulated CAR T cells from the Discovery Cohort patients and healthy

donors. **b**, Expression distribution of Type-2 Score on the UMAP. **c**, Expression distribution of each gene defining the Type-2 Score.

This promoted us to perform in vitro functional studies to assess the impact of incorporating IL-13 during CAR-specific activation (Fig. R5). In a similar experiment using IL-4, it is found that the addition of 10 ng/mL IL-4 resulted in a significant increase in cell proportion within the proliferative cluster, a decrease in the dysfunctional cytotoxic cluster, and had a negligible impact on the type-1 CAR T enriched cluster (Extended Data Fig. 8a–e). Here, UMAP clustering analysis revealed Cluster 4 as dysfunctional cells exhibiting overactivation of cytotoxicity, diminished cytokine production, and upregulation of exhaustion and apoptosis pathways. Intriguingly, despite observing a beneficial increase in the proliferative cluster, the introduction of 10 ng/mL IL-13 failed to reduce the proportion of this dysfunctional cluster. We also conducted the same analysis using IL-5 and obtained results similar to those observed with IL-13 (Extended Data Fig. 8f–h). While further dosage optimization and alternative assays are needed to draw definitive conclusions and explore differences with IL-4, we believe that this falls outside the scope of the current study and warrants a systematic evaluation in a separate work.

Fig. R5 | Clustering analysis of donor CAR T cell population response to IL-13.

a, UMAP clustering of CAR T cells from healthy donors, with and without 10ng/mL of IL-13 added during in vitro CAR-specific activation. Characteristic clusters enriched for high proliferative (Cluster 1), Type-1 (Cluster 5), Type-2 (Cluster 7), and dysfunctional cytotoxic (Cluster 4) CAR T are indicated. **b**, Expression distribution of Proliferation Score, Type-1 Score, Type-2 Score, and Cytotoxic Score on the UMAP. **c**, Comparison of cell proportions in specific clusters between conditions. “Original” denotes no IL-13 added during CAR-specific activation.

4. Fig5: In the in vivo experiment comparing type 2 high and low, while selecting two different donors by scRNA seq is interesting, the difference in CD4/CD8 ratio can potentially have a major impact in CAR T cells function, beyond the sole Type-2 cytokine effect. For example, it has been reported by Stan Riddell’s group that CD4 CAR T cells alone are more functional than CD8 CAR T cells alone, and the ratio

of each is critical for optimal antitumor response in a Nalm6 model. Also, in Extended Data Figure 10c, it is clear that ND463 has a much lower representation of cluster 1 and 4 T cell. We understand it might be impossible to find two donors that can only be differentiated by their Type 2 signature, but the current experiment is not conclusive. The authors should at least perform an experiment with similar CD4/CD8 ratio. Or better, they should pick a Type-2 low donor and pre-treat with IL-4 since they claim it can rescue the phenotype.

Response: Thank you for your invaluable suggestion! In this revised version, we have opted for the "better" solution to address this concern. We investigated whether enhancing type-2 functionality could elevate the performance of the **Type-2 Low CAR T sample (ND585)**, exploring two strategies: priming CAR T products with type-2 cytokine before infusion and incorporating type-2 cytokine into the manufacturing process, starting from apheresis T cells (Fig. R6a). Consistent with all the previous studies, 10ng/mL IL-4 was utilized to prime ND585 CAR T products, resulting in "Primed CAR T". Two different IL-4 doses were assessed during the new manufacturing of ND585 T cells, namely 10ng/mL or 50ng/mL, generating Enhanced Type-2 CAR T referred to as "ET2-L CAR T" or "ET2-H CAR T". All newly generated CAR T cells were administered to Nalm6 cell-bearing NSG mice, and their efficacy was **compared to the original Type-2 Low CAR T manufactured from the identical donor**. While all treatment groups successfully cleared the tumor burden within one week, only Primed CAR T and ET2-L/H CAR T demonstrated the ability to completely reject the same amount of tumor cell rechallenge (Fig. R6b), highlighting the therapeutic potency of enhanced type-2 functionality. Flow cytometry analysis at various time points post-CAR T infusion revealed significant expansion of CAR+ cells in the peripheral blood for the new products (Fig. R6c), coupled with significantly reduced expression of coinhibitory markers and increased IFN- γ production at the peak on day 8 (Extended Data Fig. 13g, h). This could be linked to their superior capacity contributing to a significantly prolonged survival (Fig. R6d). To rigorously assess the resilience of the two strategies, we initiated a second tumor rechallenge on day 42, simulating a relatively late-stage relapse. Following this third tumor cell injection, both the ET2-L and ET2-H CAR T showcased robust tumor control capabilities, outperforming the group treated with Primed CAR T (Fig. R6b, d).

We also implemented the priming strategy on BCA2 patient (Patient ID: DC80) CAR T cells mediating BCA duration of 3 months before developing a CD19pos relapse. During the initial 4 days of a repeat stimulation assay, cocultured with Nalm6 cells at an E/T ratio of 1:2, both the original and 10ng/mL IL-4 primed DC80 CAR T demonstrated potent tumor killing capability, achieving nearly 100% efficiency (Fig. R6e). Similarly to the in vivo results, the primed CAR T group exhibited a significant increase in CAR T cell count over this period and displayed significantly enhanced memory signatures, reduced coinhibitory markers, along with augmented expression of IL-2, IFN- γ , granzyme B, Ki67, and IL-13 on day 4 (Fig. R6e–i). We gradually decreased the E/T ratio to 1:16 on day 8, resulting in a near-total loss of tumor killing for the original DC80 CAR T cells. However, the IL-4 primed cells retained approximately 50% activity, likely attributed to their preserved memory state, significantly heightened production of functional cytokines,

and increased proliferation on day 9 (Extended Data Fig. 13k–n). These results are presented in our revised Fig. 6 and Extended Data Fig. 13.

Fig. R6 | Revitalizing Type-2 Low CAR T via enhanced type-2 functionality boost.

a, Schematic representation depicting two strategies employed to enhance the type-2 functionality of Type-2 Low CAR T derived from the donor ND585. **b**, Tumor burden measured by bioluminescence at indicated days since CAR T cell infusion (n=5 mice for each group). **c**, CAR T cell expansion in the peripheral blood of Nalm6-bearing mice measured at different timepoints after infusion (n=5 mice for each group). **d**, Kaplan–Meyer curves showing mouse survival. **e**, Evaluation of tumor cell lysis efficacy and CAR T cell count in an in vitro repeat stimulation assay using patient-derived CAR T cells (DC80) with a BCA duration of 3 months, with or without 10ng/mL IL-4 priming for 12h. Nalm6 cells were introduced daily over a 9-day span at different effector-to-target (E/T) ratios as depicted in the schematic. Flow cytometry analysis was conducted on day 4 and day 9. Significance levels on specific days are denoted due to space constraints. **f–i**, Flow cytometry analysis of memory (**f**), coinhibitory (**g**), type-1 functionality/cytotoxicity (**h**), and proliferation (**i**) markers of CAR T cells at day 4.

5. The relationship between mTor signaling and CD19⁻ relapse is interesting. The definition of CD19⁻ relapse depends on CD19 detection and it appears that many are actually CD19 very low and not CD19⁻. Could the differences observed have a link to CAR T cell sensitivity to low antigen densities?

Response: We greatly appreciate this insightful direction, which motivates us to delve deeper into this unique and fascinating discovery! Nevertheless, we concur with **Reviewer #3**'s observation that the original Figure 6, depicting metabolic and functional programs of CAR T cells, diverges from the overarching theme of this study. Consequently, we have relocated it to the supplementary figures and briefly referenced these findings in the "Discussion" section of the revised version. Instead, we have conducted an expanded proteomic profiling of patient serum samples spanning up to two months, as depicted in part in Fig. R3. Notably, the expanded data sets afford us the opportunity to conduct unsupervised analysis, delving into whether the serum proteomic profile might reflect intrinsic variations in immune response among different patients post-CAR T treatment. Impressively, the results indeed indicate the feasibility of this notion. These results are presented in our revised Fig. 4, Extended Data Fig. 11, and Supplementary Fig. 1-3. We believe the exclusion of the metabolic dataset and the expansion of patient serum profiling now more effectively highlight the central focus of this study: the role of type-2 functionality in maintaining long-term CAR T persistence.

6. In Ext Fig2 D: incomplete legend

Response: We have corrected this and reviewed the entire manuscript to prevent any similar oversights.

7. Page 7 : “these data revealed that type 2 marked cluster 7 cells mainly regulate a subpopulation”. These are correlation studies which at best suggest an interaction but does not prove by it any mechanistic study. Thus, I think at this point the author should say “suggest that cluster 7 regulates”.

Response: Thank you for bringing this to our attention! We have now updated the term to "implying a potential regulatory role of type-2 CAR T cells" to ensure accuracy. Additionally, in the revised version, we have strengthened this section by consolidating the functional results from the original Figure 4 into a more concise Figure 3. We further assessed the chromatin accessibility alterations following the inclusion of 10 ng/mL IL-4 in stimulating CAR T cells from three BCA2/1 patients. These results are presented in our revised Fig. 3 and Extended Data Fig. 7-10.

8. Schematic representation in Fig 5a: the discovery cohort is a bit confusing. Does the circle in between number suggest that the harvest was performed at some point in between these days? If so it should be clarified in the legend.

Response: You are totally correct! In the Discovery Cohort, serum collections may have been performed on different days for various patients within a given time frame. In the Validation Cohort, serum collections were consistently conducted for all patients on specified days. We have incorporated this information into the revised caption for Fig. 4a.

Referee #3 (Remarks to the Author):

In the paper entitled "Single-cell multi-omics reveals type-2 function and metabolic fitness in maintaining CAR T cell longevity associated with 8-year leukemia remission" Zhiliang and colleagues determined the molecular factors that contribute to the ultra-long persistence of CAR T cells by conducting a thorough single-cell RNA sequencing analysis of pre-infusion CAR T cells. Additionally, they correlated the transcriptional signatures of these cells with the corresponding 5-year or 8-year clinical outcomes. Remarkably, their findings indicated a notable elevation of type-2 signatures, rather than type-1 signatures, in the pre-infusion CAR T products, which was significantly associated with long-term persistence in patients. Through an examination of ligand-receptor interactions, they also discovered that type-2 cytokines contribute to long-term CAR T persistence through regulating the dysfunctional signatures of TIM3⁺ terminal effectors. Furthermore, they demonstrated that culturing BCA2 cells (CAR T cells with an average of ~4 months persistence) with the type-2 cytokine IL-4 resulted in a transcriptional and metabolic reprogramming, transforming them into BCA-L cells (CAR T cells with an average of ~101 months persistence). Notably, in a leukemic mouse model, the researchers showed that high type-2 CAR T cell products exhibited superior expansion and demonstrated increased antitumor activity upon with leukemia rechallenge.

Overall, the authors present an original and provocative study, employing rigorous analysis of pre-infusion CAR T cells and tracking their clinical responses over 8 years. Importantly, they establish, for the first time, a correlation between the type-2 response and ultra-long CAR T persistence with relatively large clinical datasets, which holds significant implications for understanding the molecular mechanisms

of remarkable persistence of CAR T cells over an extended period. These findings will undoubtedly captivate a wide audience.

Response: Thank you sincerely for your encouraging affirmation of our study!

Nonetheless, it is crucial to conduct further analysis of the scRNA-Seq and functional data to substantiate the bold conclusions drawn in the paper. In general, the authors have a clear hypothesis that they are proposing, but their experiments and analysis do not fully support the hypothesis and the experimental approach could be refined to more directly test their ideas to develop more solid conclusions. It is known that CD4 T cell help is critical to avoid Cd8 T cell exhaustion and to promote memory and so if the authors are suggesting that Th2-like cells provide the best help, then they should develop experiments to more directly examine this. With more definite tests of the hypothesis and better delineation of the subsets and their functional roles, the authors may have quite a novel clinical finding that would be of high impact to the field.

Suggested improvements:

1. Cite-seq analysis has been conducted to aid in the identification of CD4 and CD8 subsets within the scRNA-Seq analysis of pre-infusion CAR T cells. However, the subsequent analysis of clusters and cells reveals a mixture of CD4 and CD8 T cells. It is advisable to segregate the CD4 and CD8 cells and perform separate downstream analyses to strengthen their point. For instance, refer to Figure 2a and Figure 2c for illustrations. For instance, they need to better define the Th2-like cells...are they canonical CD4+ Th2 cells? Are there CD8 T cells in the Th2 cluster and if so, do they exhibit canonical Th2 properties. Talk about other surface markers typically used to define Th2 cells (chemokine receptors, CRTH2, Transcription factors, etc).

Response: Thank you very much for these constructive suggestions! They have led us to realize that we may not have fully utilized our CITE-seq datasets. In this revised version, we provided a comprehensive global correlation analysis across the identified subpopulations, unveiling substantial connections and differences among distinct transcriptomic and proteomic states within each category of cells (Fig. 1d). Remarkably, this analysis identified a correlation pattern between active immune functions and cytotoxicity with the expression of genes *CTLA4*, *LAG3*, and protein LAG-3, rather than *PDCD1* or PD-1, suggesting potential strategies for combining immunotherapy with checkpoint inhibitors.

Based on your suggestion, we performed sub-clustering analyses on CD4+ or CD8+ cells in both cohorts, separated by CITE-seq surface proteins, and identified the increased type-2 signatures in long-term

responders, despite a slightly lower percentage in CD8+ cells across all patient groups (Fig. R1). These results are presented in our revised Extended Data Fig. 4a-d.

Fig. R1 | Transcriptomic clustering analysis of CD4+ or CD8+ activated CAR T cells.

a, b, UMAP clustering of CD4+ (**a**) or CD8+ (**b**) CAR-specific stimulated CAR T cells from the Discovery Cohort patients and healthy donors, along with the expression distribution of Type-2 Score and the comparison of cell proportions in the type-2 cell-enriched cluster. **c, d**, UMAP clustering of CD4+ (**c**) or CD8+ (**d**) CAR-specific stimulated CAR T cells from the Validation Cohort patients and healthy donors, along with the expression distribution of Type-2 Score and the comparison of cell proportions in the type-2 cell-enriched cluster.

The aforementioned findings, coupled with a notable increase in BCA-L cells expressing type-2 cytokines in both CD4+ and CD8+ cells as confirmed by intracellular flow cytometry analysis, suggest that the elevated type-2 signature associated with long-term persistence was consistently present in both subtypes. Therefore, for downstream ligand-receptor analysis, we considered CD4 and CD8 CAR T cells as a single population and only conducted separate analyses based on their cohort identity (Discovery or Validation). Doing otherwise would yield redundant results.

In reference to the term "type-2" we utilized, it's important to note that both CD4+ and CD8+ subtypes exhibit canonical "Th2" properties. Signature makers defining the type-2 identity, including cytokines, transcription factors, chemokine receptors, and *PTGDR2* encoding CRTH2, collectively exhibited elevated

expression in both CD4+ and CD8+ CAR T cells from HD, BCA-L and BCA-O patients (Fig. R2). Here, we endeavored to offer a comprehensive expression profile of type-2 related genes, and certain other markers with extremely low expression were not included. These results are presented in our revised Extended Data Fig. 4e.

Fig. R2 | Dot plot showing expression profile of type-2 related genes across patient groups.

2. The authors need to better delineate the function of the Th2-like cells from the other clusters. Although the researchers observed an elevation of type-2 signatures in persistent CAR T products, it comprised less than 5% of the total population. On the other hand, the memory subsets, which account for over 30% of the total population, showed elevated levels in the persistent CAR T cells too. These memory subsets have previously been implicated in the long-term persistence of CAR T cells. Therefore, the authors must ascertain the relative contribution of the elevated presence of memory cells vs. the elevation of type 2 response in determining the ultra-long survival of CAR T cells by purifying, separating and mixing different populations of cells together.

Response: We fully acknowledge the significance of this direction and recognize its importance for in-depth investigation. However, we respectfully choose not to pursue it at this time. While the memory subset undeniably constitutes a substantial population, as you noted, numerous studies have already elucidated the associations between memory signature and CAR T persistence. Several recent publications discussing these associations were included in our "Introduction" section. Therefore, we have decided not to emphasize these findings and have just presented the results in extended figures. These results are considered "expected" and do not constitute new findings.

Instead, the primary focus of our study is to investigate the role of type-2 functionality in mediating long-term CAR T persistence. In the revised version, we have introduced multiple new datasets to bolster this argument, including single-cell ATAC profiling of patient CAR T cells to uncover upstream regulations, expanded proteomic profiling of patient serum samples spanning up to two months, and innovative manufacturing strategies aimed at rescuing type-2 low products. We sincerely hope you agree that all these datasets collectively represent our efforts to contribute unique insights into CAR T biology.

We also invite you to explore our previous work

(<https://www.science.org/doi/full/10.1126/sciadv.abj2820>) published in Science Advances (shown in Figure 7), where we demonstrated that the combination of Th2 and memory signatures predicts long-term remission or relapse (Fig. R3 included here for your convenience). While we can certainly conduct a similar analysis to determine the relative contribution of memory or type-2 functionality by adjusting their contribution ratio index to maximize prediction power, as explained earlier, this falls outside the focus of our study.

Fig. R3 | Integrated model analysis demonstrates that the combination of TH2 strength and early memory potential is predictive of patient response durability.

(A) Response predictive index of each patient in the initial discovery cohort (n = 10) and the

comparison between complete remission (CR) and relapse (RL) groups. (B) ROC curve for response prediction based on an integrative biomarker consisting of CAR+TH2+ frequency, T_{CM} frequency, and the (T_{EM} + T_{EF}) frequency in the validation cohort (n = 49). A binomial logistic regression was used to fit the model with CR or RL as the response variable, and a stratified fivefold cross-validation was implemented to compute the ROC and AUC.

3. In figure 4, the authors demonstrated that the addition of IL4 could transcriptionally revert BCA2 cells to BCA-L states. However, it is crucial to present functional data indicating that the inclusion of IL4 can also improve the functionality of BCA2 cells. This is especially important because the experiment was conducted in the regular culture system, and the exhaustion marker may simply indicate the activation state of the cells.

Response: Thank you for this invaluable suggestion! In this revised version, we first investigated whether enhancing type-2 functionality could elevate the performance of the Type-2 Low CAR T **healthy donor sample** (ND585), exploring two strategies: priming CAR T products with type-2 cytokine before infusion and incorporating type-2 cytokine into the manufacturing process, starting from apheresis T cells (Fig. R4a). Consistent with all the previous studies, 10ng/mL IL-4 was utilized to prime ND585 CAR T products, resulting in "Primed CAR T". Two different IL-4 doses were assessed during the new manufacturing of ND585 T cells, namely 10ng/mL or 50ng/mL, generating Enhanced Type-2 CAR T referred to as "ET2-L CAR T" or "ET2-H CAR T". All newly generated CAR T cells were administered to Nalm6 cell-bearing NSG mice, and their efficacy was **compared to the original Type-2 Low CAR T manufactured from the identical donor**. While all treatment groups successfully cleared the tumor burden within one week, only Primed CAR T and ET2-L/H CAR T demonstrated the ability to completely reject the same amount of tumor cell rechallenge (Fig. R4b), highlighting the therapeutic potency of enhanced type-2 functionality. Flow cytometry analysis at various time points post-CAR T infusion revealed significant expansion of CAR+ cells in the peripheral blood for the new products (Fig. R4c), coupled with significantly reduced expression of coinhibitory markers and increased IFN- γ production at the peak on day 8 (Extended Data Fig. 13g, h). This could be linked to their superior capacity contributing to a significantly prolonged survival (Fig. R4d). To rigorously assess the resilience of the two strategies, we initiated a second tumor rechallenge on day 42, simulating a relatively late-stage relapse. Following this third tumor cell injection, both the ET2-L and ET2-H CAR T showcased robust tumor control capabilities, outperforming the group treated with Primed CAR T (Fig. R4b, d).

Fig. R4 | Revitalizing Type-2 Low CAR T via enhanced type-2 functionality boost.

a, Schematic representation depicting two strategies employed to enhance the type-2 functionality of Type-2 Low CAR T derived from the donor ND585. **b**, Tumor burden measured by bioluminescence at indicated days since CAR T cell infusion (n=5 mice for each group). **c**, CAR T cell expansion in the peripheral blood of Nalm6-bearing mice measured at different timepoints after infusion (n=5 mice for each group). **d**, Kaplan–Meyer curves showing mouse survival. **e**, Evaluation of tumor cell lysis efficacy and CAR T cell count in an in vitro repeat stimulation assay using patient-derived CAR T cells (DC80) with a BCA duration of 3 months, with or without 10ng/mL IL-4 priming for 12h.

Nalm6 cells were introduced daily over a 9-day span at different effector-to-target (E/T) ratios as depicted in the schematic. Flow cytometry analysis was conducted on day 4 and day 9. Significance levels on specific days are denoted due to space constraints. **f–i**, Flow cytometry analysis of memory (**f**), coinhibitory (**g**), type-1 functionality/cytotoxicity (**h**), and proliferation (**i**) markers of CAR T cells at day 4.

We also implemented the priming strategy on **BCA2 patient** (Patient ID: DC80) CAR T cells mediating BCA duration of 3 months before developing a CD19pos relapse. During the initial 4 days of a repeat stimulation assay, cocultured with Nalm6 cells at an E/T ratio of 1:2, both the original and 10ng/mL IL-4 primed DC80 CAR T demonstrated potent tumor killing capability, achieving nearly 100% efficiency (**Fig. R4e**). Similarly to the in vivo results, the primed CAR T group exhibited a significant increase in CAR T cell count over this period and displayed significantly enhanced memory signatures, reduced coinhibitory markers, along with augmented expression of IL-2, IFN- γ , granzyme B, Ki67, and IL-13 on day 4 (**Fig. R4e–i**). We gradually decreased the E/T ratio to 1:16 on day 8, resulting in a near-total loss of tumor killing for the original DC80 CAR T cells. However, the IL-4 primed cells retained approximately 50% activity, likely attributed to their preserved memory state, significantly heightened production of functional cytokines, and increased proliferation on day 9 (Extended Data Fig. 13k–n). These results are presented in our revised Fig. 6 and Extended Data Fig. 13.

4. In figure 5, the mouse leukemia model utilizing type-2 high cells failed to adequately address their point, mainly due to the limited representation of the type-2 high cells among the total transferred cells. To overcome this limitation, the author should consider (a) sorting out the type-2 cells and conducting the experiments anew. A potential surrogate for the sorting strategy could involve utilizing CXCR3-CCR6-CCR4+ as a distinguishing marker (<https://pubmed.ncbi.nlm.nih.gov/20042588/>). (b) blocking type-2 signaling through neutralizing antibodies. This is essential to draw more solid conclusions since it is experimentally tractable.

Response: Thank you for your comment and your constructive suggestions! We have chosen to implement your first suggestion and conducted in vitro assays to further confirm the functional role of the type-2 population in the observed superior response. We sorted out type-2 cells from the Type 2 High CAR T based on the surface marker expression of CCR3 and CCR4, conducting in vitro coculture with Nalm6 cells over a 4-day period at an effector/target (E/T) ratio of 1:4 (**Fig. R5a**). The exclusion of this population markedly compromised the anti-tumor capacity, resulting in a reduction of the absolute CAR T cell number at the end of the killing assay (**Fig. R5b**). Phenotypic analysis revealed heightened coinhibitory signatures, diminished type-1 functionalities, and decreased memory states in the type-2 sorted group (**Fig. R5c**). This vulnerability was partially alleviated by supplementing with 10ng/mL IL-4 after type-2 cell removal; intriguingly, the addition of IL-4 further augmented the tumor cell lysis capability and CAR T cell count of the original Type-2 High CAR T. These results are presented in our revised Fig. 5h–j. We sincerely appreciate your understanding that conducting new in vivo experiments with sorted type-2 cells poses challenges, especially as we prioritize other experiments that may hold

greater significance within the constraints of a limited revision period. We believe that the addition of all the new datasets listed collectively on the first page of this response file, particularly the in vivo results presented in the above Fig. R4 with follow-up exceeding 100 days, provides valuable insights to enhance this work.

Fig. R5 | Type-2 cells are essential to support the superior performance of Type-2 High CAR T.

h, Schematic of an in vitro repeat stimulation assay using Type-2 High CAR T products, with type-2 cells sorted out based on the surface markers CCR3 and CCR4. Nalm6 cells were added daily over a 4-day period at an effector-to-target (E/T) ratio of 1:4. **i**, Evaluation of tumor cell lysis efficacy and CAR T cell count at the endpoint under different conditions. **j**, Flow cytometry analysis of coinhibitory, type-1 functionality, and memory markers of CAR T cells at the endpoint.

5. In figure 6, it is crucial to comprehend the significant role of elevated type-2 responses in contributing to the metabolic and functional reprogramming associated with non-response, CD19-positive and CD19-negative relapse. However, the comparison of metabolic differences between the responders vs. non-response dose not add noteworthy significance or impact, and it seems to diverge from the overall theme of the paper.

Response: We deeply appreciate these insightful suggestions and have decided to relocate these results to the supplementary figures, while also providing a brief discussion in the "Discussion" section of the revised version. Instead, to further enhance the clinical relevance of our study and evaluate the role of type-2 functionality over an extended period following CAR T infusion, we have expanded the proteomic profiling of patient serum samples. This extension encompasses up to two months post-infusion in the Discovery Cohort and up to 28 days post-infusion in the Validation Cohort (Fig. R6a). Once more, these datasets validate the presence of elevated circulating type-2 cytokines in the post-infusion sera of long-term BCA-L and BCA-O patients (Fig. R6b-e). Notably, the expanded data sets afford us the opportunity

to conduct unsupervised analysis, delving into whether the serum proteomic profile might reflect intrinsic variations in immune response among different patients post-CAR T treatment. Impressively, the results indeed indicate the feasibility of this notion (Fig. R6f-h). These results are presented in our revised Fig. 4, Extended Data Fig. 11, and Supplementary Fig. 1-3. We believe the exclusion of the metabolic dataset and the expansion of patient serum profiling now more effectively highlight the central focus of this study: the role of type-2 functionality in maintaining long-term CAR T persistence.

Fig. R6 | Elevated levels of type-2 cytokines detected in post-infusion sera from long-term responders.

a, Schematic of the serial proteomic profiling to measure serum proteins in 33 patients from the Discovery Cohort and 8 patients from the Validation Cohort. Timepoints are relative to the day of first infusion of CTL019 cells (Day 0). **b**, **c**, Longitudinal levels of type-2 cytokines in patients from the Discovery Cohort (**b**) and the Validation Cohort (**c**). **d**, **e**, Comparison of the average type-2 cytokine levels at multiple time points between persistent groups in the

Discovery Cohort (d) and the Validation Cohort (e). f, Principal Component Analysis (PCA) of 345 measurements of serum samples from 33 patients in the Discovery Cohort, based on the detected values of the 30-Plex cytokines and grouped by BCA response. g, Unsupervised clustering analysis of proteomic measurements, grouped by Cluster ID or BCA response, visualized using UMAP. Cluster 2 exhibits enrichment in BCA-L patients. h, Heatmap showing the differentially expressed proteins defining each cluster. Cluster 2 exhibits a high expression level of type-2 cytokines (IL-13 and IL-4).

Additional comments:

1. Exhaustion signature was calculated only based on 4 genes; more exhaustion genes should be included since these are also “activation markers” and exhaustion is truly defined by loss of cytokine production like IL-2, TNFa and IFNg.

Response: We highly appreciate this valuable suggestion and acknowledge that including more exhaustion genes could potentially enhance the accuracy of the signature annotation. However, as a multi-omics study encompassing surface proteomic profiling, we have chosen to maintain consistency with our CITE-seq panel, which includes canonical markers including PD-1, CTLA-4, TIM-3, LAG-3, and TIGIT. In the revised version, we have changed the term "exhaustion" to "co-inhibitory" to describe both gene and protein markers throughout the manuscript, aiming to prevent any misleading information. Regarding the definition of dysfunctional clusters of cells (primarily used in Fig. 3 and Extended Data Fig. 7-9), we strictly adhere to the widely accepted definition, which involves increased expression of these co-inhibitory markers, loss of cytokine production, limited proliferative activity, and activation of exhaustion or apoptosis-related signaling pathways.

2. It would be beneficial to include IL10 and IL21 production in single cell secretome assay.

Response: Thank you for bringing this to our attention! In the revised version, we have expanded our analysis to include three additional type-2 related cytokines: IL-9, IL-10, and IL-21. The index of CAR T cells secreting type-2 cytokine IL-4, IL-5, IL-9, IL-13, and IL-21 was significantly higher in BCA-L and BCA-O patients (Fig. R7), while IL-10 secretion showed indiscernible levels between persistence groups. These results are presented in our revised Fig. 2f.

Fig. R7 | Comparison of type-2 cytokine secretion levels between persistence groups, utilizing multiplexed secretomic assay on a cohort of 32 patients. The 'Secretion Index' reflects the frequency of cells secreting a specific cytokine multiplied by the average signal intensity of that cytokine.

3. Figure 2b and Figure 2d showed expression score of type-1 and type-2 signatures. Additionally, it would be beneficial to include the expression scores of individual type-1 and type-2 genes.

Response: Thank you for this valuable suggestion! Previously, we presented a module expression analysis that integrated four type-1 genes (*IFNG*, *TNF*, *CSF2*, and *TBX21*) and four type-2 genes (*IL4*, *IL5*, *IL13*, and *GATA3*) (Fig. R8a, b). In the revised version, we included the expression distribution of each gene individually and observed that genes defining Type-1 or Type-2 Score have variable expression distribution and level (Fig. R8c), highlighting the necessity of using a group of genes to define a phenotypic signature. This observation holds true across the Validation Cohort UMAPs as well. The individual expression plots are presented in our revised Extended Data Fig. 3d, g.

Fig. R8 | Expression distribution of type-2 gene module and defining markers.

a, UMAP clustering of CAR-specific stimulated CAR T cells from the Discovery Cohort patients and healthy donors.

b, Expression distribution of Type-2 Score on the UMAP. **c**, Expression distribution of each gene defining the Type-1 or Type-2 Score.

Referee #4 (Remarks to the Author):

In this manuscript, Bai et al. conducted single-cell analysis on approximately 700k pre-infusion CAR T cells obtained from 82 ALL patients and 6 healthy donors. They investigated the relationship between the type-2 function of CAR T cells and the outcome of CAR T cell therapy based on long-term clinical follow-up data. In addition, the authors performed in vitro experiments and utilized a leukemic mouse model to validate their findings, with a particular focus on the importance of type-2 functionality in persistent CAR T cells. This study presents a valuable dataset for assessing CAR T cell persistency. The manuscript is well-written and easily understandable. The robustness of the results and the overall integrity of the manuscript maybe further enhanced by addressing the following issues.

Response: Thank you for acknowledging the value of our large-scale single-cell dataset!

Major comments:

1. In lines 122-123, the authors state that “basal state CAR T cells are partitioned into 8 sub-clusters...”, but the corresponding figures do not show or indicate these eight clusters. It should be noted that Extended Data Fig. 1 identifies a total of 17 clusters.

Response: We apologize for the oversight in providing insufficient information. In our global analysis of the entire dataset, encompassing both basal unstimulated and CD19-3T3 stimulated CAR T cells, unsupervised clustering identified a total of 17 clusters (Fig. R1a). This profile exhibited clear separation based on stimulation conditions, with basal CAR T cells predominately located in the right half portion (Fig. R1b), constituting **8 clusters** in total. In the revised version, we have rectified this by clearly indicating this information, listing the cluster IDs in the modified description: "The basal state CAR T cells were partitioned into 8 sub-clusters (Cluster ID: 0, 2, 5, 6, 8, 11, 15, 16), primarily distinguished by their memory or cell cycle state."

Fig. R1 | Global clustering analysis of the integrated dataset.

a, UMAP visualization of 695,819 high-quality single CAR T cells across all patients and donors. Unsupervised clustering

identifies 17 distinct clusters. **b**, UMAP distribution of all the single cells grouped by in vitro stimulation condition (upper panel) or cell cycle (lower panel).

2. In lines 215-216 and lines 218-240, while the authors demonstrated elevated levels of IL-4 and IL-25 in Cluster 7 as a type-2 functionality cluster, there is a lack of analysis or validation regarding STAT6 and GATA3 activation. For Fig 3e-3g, expression profile of all Type-2 related genes need to be analyzed and presented. Similarly in Fig. 2k, the key molecules in this cascade model should be validated in patient samples before drawing conclusions about the upstream regulation of type-2 function in the present study.

Response: Thank you for these invaluable suggestions! To reveal the upstream regulatory factors linked to CAR T persistence, we conducted a single-cell ATAC and gene co-profiling of CD19-specific stimulated CAR T cells (Fig. R2a), comparing 3 BCA-L patients with another 3 patients from the BCA2 or BCA1 groups, who maintained BCA duration of less than 3 months. Pseudo-bulk chromatin accessibility signals in the genomic regions of *GATA3* were notably higher in each of the BCA-L patients (Fig. R2b). Interestingly, comparable signals were observed for another key type-2 regulator, *STAT6*, across the six patients (Fig. R2c). Within the realm of significantly enhanced motif binding activities observed in BCA-L CAR T cells compared to the BCA2/1 cells, *GATA3* emerged as the foremost differentially enriched site, along with several other GATA family members (Fig. R2d). Delving into the per-cell motif activity profile for each patient, *TBX21* displayed consistent levels across all patients, whereas *GATA3* showcased

notably enriched activities in single cells from the three BCA-L patients. It's noteworthy that STAT6 did not emerge as a differential motif in this analysis. These results are presented in our revised Fig. 2g-j and Extended Data Fig. 6.

Fig. R2 | Single-cell ATAC+Gene co-profiling of CD19-specific activated patient CAR T cells.

a, Integrated ATAC-Gene UMAP clustering of CAR-specific stimulated CAR T cells derived from 6 patients. Unsupervised clustering reveals 10 distinct clusters, with an enrichment of type-2 cells in Cluster A3. Comparison of cell proportions in Cluster A3 is presented between patient groups. **b, c**, Pseudo-bulk chromatin accessibility tracks in the genomic region of *GATA3* (**b**) and *STAT6* (**c**), depicted separately for each patient. The enhancer elements predicted by ENCODE within this region are highlighted in a light-yellow shade. **d**, Volcano plot showing differential motif activities in BCA-L vs. BCA2/1 CAR T cells, with *GATA3* identified as the most enhanced motif in the BCA-L group. Dot plot showing expression profile of type-2 motif MA0037.3 (*GATA3*) and type-1 motif MA0690.1 (*TBX21*) across each patient.

To further gauge the relative importance of *STAT6* and *GATA3* in orchestrating type-2 functionality, an in vitro repeat stimulation assay was conducted using CAR T cells with knockdown of *STAT6* or *GATA3* (**Fig. R3a**). While both knockdowns significantly attenuated tumor killing efficacy and CAR T cells count, the *GATA3* knockdown demonstrated a notably more pronounced compromise in the CAR T population

number at the assay's endpoint (Fig. R3b), suggesting its central role in sustaining functional type-2 immunity in long-term persistent CAR T cells. These results are presented in our revised Fig. 2k.

Fig. R3 | Loss-of-function experiment to evaluate the relative importance of GATA3 and STAT6.

a, b, Evaluation of tumor cell lysis efficacy and CAR T cell count in an in vitro repeat stimulation assay using CAR T cells with knockdown of STAT6 or GATA3. Nalm6 cells were added daily over a 4-day period at an effector-to-target (E/T) ratio of 1:4.

Following your suggestion, we have expanded the expression profile to include additional Type-2 related genes. In reference to the term "type-2" we utilized, it's important to note that both CD4+ and CD8+ subtypes exhibit canonical "Th2" properties. Signature makers defining the type-2 identity, including cytokines, transcription factors, chemokine receptors, and

PTGDR2 encoding CRTH2, collectively exhibited elevated expression in both CD4+ and CD8+ CAR T cells from HD, BCA-L and BCA-O patients (Fig. R4). Here, we endeavored to offer a comprehensive expression profile of type-2 related genes, and certain other markers with extremely low expression were not included. These results are presented in our revised Extended Data Fig. 4e.

Fig. R4 | Dot plot showing expression profile of type-2 related genes across patient groups.

We also expanded our analysis to include three additional type-2 related cytokines in our secretomic dataset: IL-9, IL-10, and IL-21. The index of CAR T cells secreting type-2 cytokine IL-4, IL-5, IL-9, IL-13, and

IL-21 was significantly higher in BCA-L and BCA-O patients (Fig. R5), while IL-10 secretion showed indiscernible levels between persistence groups. These results are presented in our revised Fig. 2f.

Fig. R5 | Comparison of type-2 cytokine secretion levels between persistence groups, utilizing multiplexed secretomic assay on a cohort of 32 patients. The 'Secretion Index' reflects the frequency of cells secreting a specific cytokine multiplied by the average signal intensity of that cytokine.

Regarding the molecular cascade originally depicted in Fig. 2k, we have recognized its limitation in providing conclusive information. Therefore, we have relocated this portion to Extended Data Figure 4f and adjusted the manuscript accordingly by removing detailed descriptions.

3. The detailed approaches applied for ligand-receptor analysis and communication pathway analysis in Figs. 3a and 3b should be stated in Methods section.

Response: Thank you for this suggestion! In the revised Methods section, we have included a concise description of this analysis and referenced the R package we utilized, directing readers to access more detailed information. The content of this section is as follows:

The R toolkit Connectome V1.0.0 (with reference listed below) was employed to investigate cell-cell connectivity patterns using ligand and receptor expressions from our scRNA-seq datasets with default parameters. The normalized Seurat object served as input, and cluster identities were utilized to define nodes in the interaction networks, resulting in an edge list connecting pairs of nodes through specific ligand-receptor mechanisms. We selected top-ranked interaction pairs for visualization, prioritizing those more likely to be biologically and statistically significant based on the scaled weights of each pair. The thickness of edges is directly proportional to correlation weights, with wider edges indicating a

higher level of interaction. The "sources.include" and "targets.include" parameters were applied to specify the source cluster emitting ligand signals and the target cluster expressing receptor genes that sense the ligands.

Raredon, M. S. B. *et al.* Single-cell connectomic analysis of adult mammalian lungs. *Sci Adv* 5, eaaw3851 (2019). <https://doi.org/10.1126/sciadv.aaw3851>

4. The definition of "Type-2 L-R interactions" also needs to be clarified. Are any L-R interactions that includes type-2-related proteins can only be categorized into Type-2 L-R interactions?

Response: Thank you for bringing this to our attention! We recognize that "Type-2 L-R interactions" may be arbitrary and have revised it to "L-R interactions involving type-2 cytokines".

5. In Fig. 3b, what does the term "top-ranking level of interactions" specifically refer to? The reviewer suggests that the methodology used to determine Cluster 7 to 2 communication should be clarified.

Response: Thank you for your comment and your constructive suggestion! In these L-R plots (Fig. 3a, b in our revised manuscript), the thickness of edges is directly proportional to correlation weights, with wider edges indicating a higher level of interaction (also included in the Methods section describing the L-R analysis). Specifically, the ligand signals originating from type-2 cells in Cluster 7 (bottom half circle in Fig. R6, orange color) were received by all other clusters (top half circle in Fig. R6). The edge thickness of the Cluster 2 was the largest, representing the highest level of interactions.

In the revised main manuscript, we adjusted the description to better guide the reader in

understanding this message. The content of our new description is as follows: Within the intricate network of L-R communication pathways emanating from type-2 cells in Cluster 7, a notable hierarchy emerges, with the highest echelon of interactions explicitly converging on Cluster 2 cells, accounting for 13.9% of the whole population.

Fig. R6 | Identification L-R interactions originating from type-2 enriched Cluster 7 cells, predominantly interacting with Cluster 2 cells through L-R pairs involving type-2 cytokines.

6. In lines 277-288, The observation of a lower Exhaustion Score and Exhaustion ADT Score in Cluster 2 of BCA-L cells does not provide sufficient evidence to conclude that type-2 marked Cluster 7 cells primarily regulate Tim-3+ dysfunctional CAR T cells.

Response: Thank you for bringing this to our attention! In the revised version, we have strengthened this section by consolidating the functional results from the original Figure 4 into a more concise Figure 3 (Fig. R7 shown below). Additionally, we assessed the chromatin accessibility alterations following the inclusion of 10 ng/mL IL-4 in stimulating CAR T cells from short-term patients using single-cell ATAC-seq. In response to your suggestion, we have adjusted the section title to "Type-2 CAR T cells regulate dysfunctional subpopulation" to ensure clarity and accuracy. We believe that this revised step-by-step logical flow: (1) utilizing L-R analysis to identify the regulatory roles of type-2 cells; (2) conducting functional evaluations on both donor and patient samples by incorporating type-2 cytokines into the activation process; and (3) performing ATAC assays to unveil the epigenetic alterations upon IL-4 inclusion, provides robust support for our new section title. All these results are presented in our revised Fig. 3 and Extended Data Fig. 7-10.

Fig. R7 | Type-2 CAR T cells regulate dysfunctional subpopulation.

a, Identification of ligand-receptor (L-R) interactions originating from type-2 enriched Cluster 7 cells. **b**, Identification of L-R interactions targeting Cluster 2 cells. **c**, Dot plot showing expression profile of type-2 receptor genes across all clusters, with notable high expression observed in Cluster 2. **d**, Differentially expressed genes (DEGs) specific to Cluster 2 in comparison to all other clusters, along with the expression distribution of the

Cytotoxic Score. **e**, Corresponding signaling pathways regulated by the DEGs identified in Cluster 2. **f**, Heatmap showing the average expression levels of coinhibitory-related genes or ADT proteins across all single cells within each identified cluster. **g**, Expression distribution of Proliferation Score on the Discovery Cohort UMAP, with an evident absence of expression observed in Cluster 2. **h**, Comparison of cell proportion in Cluster 2 between persistence groups. **i**, Experimental design schematic for assessing the impact of IL-4 supplementation on the functional profile of CAR T cells derived from short-term BCA2 patients. **j**, UMAP clustering of CAR T cells from 6 patients in BCA2 group, with and without 10ng/mL IL-4 added during in vitro CAR-specific activation. **k**, Comparison of cell proportions in specific clusters 1 between conditions. **l**, Comparison of regulatory pathways between CAR T cells from the long-term BCA-L patients and six BCA2 patients under original and 10ng/mL IL-4 conditions. **m**, Volcano plot showing differential accessible peak activities in BCA2/1 CAR T cells with and without the addition of 10ng/mL IL-4. **n**, Pseudo-bulk chromatin accessibility tracks in the genomic region of *CSF2* and *IL32*, depicted separately for each patient with and without the addition of IL-4. The enhancer elements predicted by ENCODE within this region are highlighted in a light-grey shade. **o**, Dot plot showing expression profile of type-2 motif MA0037.3 (GATA3) and type-1 motif MA0690.1 (STAT1) across each patient with and without the addition of IL-4.

7. In Fig. 3c, which Cluster is the top-ranking cluster that communicate with Cluster 2, is it also Cluster 7?

Response: From the Fig. R7 above, clusters 6, 7, and 9 exhibit similarly high levels of communication with cells in Cluster 2. We have chosen not to explore the other two clusters further, as they do not show statistically significant proportional differences between CAR T persistent groups, and therefore fall outside the focus of this study.

8. In Figs. 4h and 4i, is it possible to calculate the fold change that quantifies the improvement in functional fitness of CAR T cells upon the addition of IL-4? This would help in assessing the extent of the observed change.

Response: As additional datasets from functional assays have been included, we have relocated the figures you are referring to Extended Data Fig. 9e. Using differentially expressed genes (DEGs) analysis comparing CAR T cells treated with 10ng/mL IL-4 to the original condition, we generated the signaling pathway profile regulated by these identified DEGs. A z score was used to reflect the activation level ($z > 0$, activated/upregulated; $z < 0$, inhibited/downregulated; $z \geq 2$ or $z \leq -2$ can be considered significant). For the signaling programs presented in this figure, significant differences are observed in most of them. However, quantifying the fold change in such analyses may not be reliable. An alternative approach would be to calculate the expression fold change of selected marker genes, but this may not fully capture the intricate pathway activities. Therefore, we sincerely hope you can understand our decision to rely on the current pathway analysis, which better reflects the global functional changes upon the addition of IL-4.

Instead, we conducted in vitro repeat stimulation assays using an IL-4 priming strategy on patient CAR T cells with a BCA duration of only 3 months (Fig. R8a). This approach allowed for a more accurate determination of both the fold change in CAR T cell count and the expression of functional molecules through flow cytometry data. The IL-4 primed CAR T group exhibited a significant increase in CAR T cell count and displayed significantly enhanced memory signatures, reduced coinhibitory markers, along with augmented expression of IL-2, IFN- γ , granzyme B, Ki67, and IL-13 on day 4 (Fig. R8b-e). These results are presented in our revised Fig. 6e-i and Extended Data Fig. 13j-o.

Fig. R8 | Enhancing functional fitness of short-term patient CAR T via type-2 cytokine priming.

a, Evaluation of tumor cell lysis efficacy and CAR T cell count in an in vitro repeat stimulation assay using patient-derived CAR T cells (DC80) with a BCA duration of 3 months, with or without 10ng/mL IL-4 priming for 12h. Nalm6 cells were introduced daily over a 9-day span at different effector-to-target (E/T) ratios as depicted in the schematic. **b–e**, Flow cytometry analysis of memory (**b**), coinhibitory (**c**), type-1 functionality/cytotoxicity (**d**), and proliferation (**e**) markers of CAR T cells at day 4.

9. In leukemic mouse model experiments, how are the type-2 -low and -high CAR T cells produced prior to infusion? How to define the type-2 functionality in the future potential manufactural application?

Response: We performed clustering analysis of scRNA-seq data from the 6 healthy donors and identified ND463 as Type-2 High and ND585 as Type-2 Low, determined by the proportion of cells within the type-2 enriched Cluster 6 (Fig. R9, Extended Data Fig. 12a–c in the revised version). There was no significant variance observed in Type-1 Score gene expression between these two groups (Extended Data Fig. 12e). Despite the anticipated higher ratio of CD4+ subtype in Type-2 High CAR T cells, flow cytometry analysis revealed comparable CAR transduction efficiency, expression of memory markers, and coinhibitory markers (Extended Data Fig. 12f–h).

Fig. R9 | Selection of type-2 low and type-2 high CAR T for in vivo leukemia model study.

a, UMAP clustering of CAR-specific stimulated CAR T cells from six healthy donors, along with the expression distribution of surface proteins ADT-CD4 and ADT-CD8. **b**, Expression distribution of Type-2 Score on the UMAP in (a), with the gene module found to be enriched in Cluster 6. **c**, Comparison of cell proportions in each identified cluster among different donors.

We demonstrated that the Type-2 High CAR T products (ND463) exhibited superior expansion and antitumor activity particularly upon leukemia rechallenge. To further confirm the functional role of the type-2 population in the observed beneficial response and provide experimentally tractable data, we sorted out type-2 cells from the Type 2 High CAR T based on the surface marker expression of CCR3 and CCR4, conducting in vitro coculture with Nalm6 cells over a 4-day period at an effector/target (E/T) ratio of 1:4 (Fig. R10a). The exclusion of this population markedly compromised the anti-tumor capacity, resulting in a reduction of the absolute CAR T cell number at the end of the killing assay (Fig. R10b). Phenotypic analysis revealed heightened coinhibitory signatures, diminished type-1 functionalities, and decreased memory states in the type-2 sorted group (Fig. R10c). This vulnerability was partially alleviated by supplementing with 10ng/mL IL-4 after type-2 cell removal; intriguingly, the addition of IL-4 further augmented the tumor cell lysis capability and CAR T cell count of the original Type-2 High CAR T. These results are presented in our revised Fig. 5h-j.

Fig. R10 | Type-2 cells are essential to support the superior performance of Type-2 High CAR T.

h, Schematic of an in vitro repeat stimulation assay using Type-2 High CAR T products, with type-2 cells sorted out based on the surface markers CCR3 and CCR4. Nalm6 cells were added daily over a 4-day period at an effector-to-target (E/T) ratio of 1:4. **i**, Evaluation of tumor cell lysis efficacy and CAR T cell count at the endpoint under different conditions. **j**, Flow cytometry analysis of coinhibitory, type-1 functionality, and memory markers of CAR T cells at the endpoint.

We further investigated whether enhancing type-2 functionality could elevate the performance of the **Type-2 Low CAR T sample (ND585)**, exploring two strategies: priming CAR T products with type-2 cytokine before infusion and incorporating type-2 cytokine into the manufacturing process, starting from apheresis T cells (Fig. R11a). Consistent with all the previous studies, 10ng/mL IL-4 was utilized to prime ND585 CAR T products, resulting in "Primed CAR T". Two different IL-4 doses were assessed during the new manufacturing of ND585 T cells, namely 10ng/mL or 50ng/mL, generating Enhanced Type-2 CAR T referred to as "ET2-L CAR T" or "ET2-H CAR T". All newly generated CAR T cells were administered to Nalm6 cell-bearing NSG mice, and their efficacy was **compared to the original Type-2 Low CAR T manufactured from the identical donor**. While all treatment groups successfully cleared the tumor burden within one week, only Primed CAR T and ET2-L/H CAR T demonstrated the ability to completely reject the same amount of tumor cell rechallenge (Fig. R11b), highlighting the therapeutic potency of enhanced type-2 functionality. Flow cytometry analysis at various time points post-CAR T infusion revealed significant expansion of CAR+ cells in the peripheral blood for the new products (Fig. R11c), coupled with significantly reduced expression of coinhibitory markers and increased IFN-γ production at the peak on day 8 (Extended Data Fig. 13g, h). This could be linked to their superior capacity contributing to a significantly prolonged survival (Fig. R11d). To rigorously assess the resilience of the two strategies, we initiated a second tumor rechallenge on day 42, simulating a relatively late-stage relapse. Following this third tumor cell injection, both the ET2-L and ET2-H CAR T showcased robust tumor control capabilities, outperforming the group treated with Primed CAR T (Fig. R11b, d). These results are presented in our revised Fig. 6a-d and Extended Data Fig. 13a-i.

Fig. R6 | Revitalizing Type-2 Low CAR T via enhanced type-2 functionality boost.

a, Schematic representation depicting two strategies employed to enhance the type-2 functionality of Type-2 Low CAR T derived from the donor ND585. **b**, Tumor burden measured by bioluminescence at indicated days since CAR T cell infusion (n=5 mice for each group). **c**, CAR T cell expansion in the peripheral blood of Nalm6-bearing mice measured at different timepoints after infusion (n=5 mice for each group). **d**, Kaplan–Meyer curves showing mouse survival.

We sincerely hope you agree that all these additional experiments collectively generate valuable pre-clinical datasets that could be useful in future development efforts. However, determining the optimal level of type-2 functionality for future applications remains unclear. Our scRNA-seq data indicate an average of 8% type-2 CAR T cells in 8-year cancer-free patients, compared to less than 2% in short-term

counterparts. While these findings could serve as potential thresholds, establishing clinical standards requires thorough evaluations. We have incorporated this information into our "Discussion" section.

10. Please provide the GEO dataset accession number.

Response: We have prepared all the raw and processed sequencing data for deposition into the GEO database. However, our large-scale dataset exceeds 10TB and requires special allocation of space for our project. We have been actively working with the Sequence Read Archive team from the National Library of Medicine on this matter, and we will ensure that the accession number is accessible before publication.

Minor points:

1. In Extended Data Fig.2c, labels for groups are not all indicated.

Response: We have corrected this and reviewed the entire manuscript to prevent any similar oversights.

2. In line 330-331, the expression "Baseline measurements (-2 to 0 days before CTL019 infusion)" is confusing. Maybe more appropriate with "0 to 2 days before CTL019 infusion".

Response: Thank you very much for bringing this to our attention! We have made the necessary modification in accordance with your suggestion.

Reviewer Reports on the First Revision:

Referees' comments:

Referee #1 (Remarks to the Author):

Bai et al. should be commended for their extensive and impressive responses to reviewers. They have invested significantly in responding to the reviewers' comments and the study is an important addition to our understanding of CAR T cell biology and patient outcome.

I would like to raise one comment as regard the response to reviewers (Fig. R3/ Fig. 2K); the authors show significant differences in tumor cell lysis following both STAT6-KD and GATA3-KD with triplicate samples. However, it is not clear whether these experiments reflect technical triplicates or 3 distinct donors. This should be indicated. Additionally, for the mouse experiments with n=5, these data most likely represent CAR T products from one donor. It would be helpful to indicate if these experiments are representative of data performed with multiple donors as results can vary significantly as a function of donors.

Referee #2 (Remarks to the Author):

We thank the authors for addressing the reviewers comments. The manuscript is greatly improved and will likely have a broad impact in the cancer immunotherapy field.

Referee #4 (Remarks to the Author):

In the revised version, the authors have appropriately addressed all my concerns. Specifically, they conducted additional analyses of single-cell ATAC-seq data and in vitro functional assays to reinforce the mechanistic evidence of upstream regulation of STAT6 and GATA3 signaling in type-2 differentiation. Additionally, the expansion of the patient sera proteomic dataset post-CAR T infusion provides further clinical insight, supporting the sustained tumor remission associated with the type-2 phenotype.

Taken together, the present study addresses the unmet need for understanding CAR T longevity by leveraging clinical trials with ultra-long-term responses. Through single-cell multi-omics analysis of CAR T products, coupled with in vivo and in vitro assays, this study provides mechanistic insights into enhancing CAR T persistence by augmenting type-2 functionality. These findings offer a potential avenue for improving CAR T therapy outcomes.

One minor suggestion to enhance the precision of the manuscript:

In Introduction line 71 “...one million single CAR T cells...”, it might be more precise to use the exact cell number (695,819) which is the retained cells used for the present analysis.

Author Rebuttals to First Revision:

Response to Reviewers

We are very appreciative of all the constructive feedback from the reviewers and have properly addressed all the comments. All the major changes are highlighted in yellow in the manuscript attached.

Point-by-point replies to Referees' comments:

Referee #1 (Remarks to the Author):

Bai et al. should be commended for their extensive and impressive responses to reviewers. They have invested significantly in responding to the reviewers' comments and the study is an important addition to our understanding of CAR T cell biology and patient outcome.

Response: Thank you very much for acknowledging the significance of our study and recognizing our efforts during the last revision!

I would like to raise one comment as regard the response to reviewers (Fig. R3/ Fig. 2K); the authors show significant differences in tumor cell lysis following both STAT6-KD and GATA3-KD with triplicate samples. However, it is not clear whether these experiments reflect technical triplicates or 3 distinct donors. This should be indicated. Additionally, for the mouse experiments with n=5, these data most likely represent CAR T products from one donor. It would be helpful to indicate if these experiments are representative of data performed with multiple donors as results can vary significantly as a function of donors.

Response: The experiments and results shown in Fig. 2k reflect 3 technical replicates rather than 3 distinct donors. This information has been indicated in the figure legend in the revised version (Page 15, Line 28). Regarding the mouse in vivo experiments with n=5, these data represent CAR T products from a single donor rather than representative from multiple donors. The purpose of this section is to evaluate whether CAR T products with a higher type-2 composition exhibit enhanced antitumor efficacy compared to those with lower type-2 composition. Therefore, we selected one donor with the highest type-2 cell content and another with the lowest type-2 cell content among six different donors to

perform the in vivo experiments. We believe that all the results shown in Fig. 5 and Fig. 6, generated from a systematic and rigorous experimental set, are adequate to support our conclusions.

Referee #2 (Remarks to the Author):

We thank the authors for addressing the reviewers comments. The manuscript is greatly improved and will likely have a broad impact in the cancer immunotherapy field.

Response: Thank you very much for acknowledging the significance of our work!

Referee #4 (Remarks to the Author):

In the revised version, the authors have appropriately addressed all my concerns. Specifically, they conducted additional analyses of single-cell ATAC-seq data and in vitro functional assays to reinforce the mechanistic evidence of upstream regulation of STAT6 and GATA3 signaling in type-2 differentiation. Additionally, the expansion of the patient sera proteomic dataset post-CAR T infusion provides further clinical insight, supporting the sustained tumor remission associated with the type-2 phenotype.

Taken together, the present study addresses the unmet need for understanding CAR T longevity by leveraging clinical trials with ultra-long-term responses. Through single-cell multi-omics analysis of CAR T products, coupled with in vivo and in vitro assays, this study provides mechanistic insights into enhancing CAR T persistence by augmenting type-2 functionality. These findings offer a potential avenue for improving CAR T therapy outcomes.

Response: Thank you very much for affirming our efforts to revise the manuscript and highlighting the potential impact of our work!

One minor suggestion to enhance the precision of the manuscript:

In Introduction line 71 "...one million single CAR T cells...", it might be more precise to use the exact cell number (695,819) which is the retained cells used for the present analysis.

Response: Thank you very much for bringing this to our attention! We have made the necessary modification in accordance with your suggestion (Page 2, Line 18).